# ZNF334 truncation mutation drives cold-induced autoinflammation

Joung-Liang Lan[1,2,3], Shih-Hsin Chang [ID] [3,4], Yi-Hua Lai[1,2,3], Ju-Pi Li [ID] [2,5], Guan-Jun Chen[6], Yen-Ju Lin[7,8], Ya-Ling Huang[6], Bor-Luen Chiang [ID] [9,10], Jan-Gowth Chang[1,11], Guan-Yun Lu[6], Tung-Lin Tsai[12], Chien-Yu Lin[6], John Wang[13], Yi-Chuan Li [ID] [14,15], Mien-Chie Hung [ID] [16] & Chin-An Yang [ID] [1,6,12,17,18 ✉]

## Abstract

**The role of zinc finger protein 334 (ZNF334) in immunological processes remains unknown. We identified a ZNF334 truncation mutation (p.Thr399fs) in a rare case of late-onset cold-induced autoinflammatory disease with elevated TNF-α, IL-1β, IL-6, and extracellular heat shock protein 90 (eHsp90) plasma levels and progressive sensorineural hearing loss. Using patient-derived monocytes and CRISPR/Cas9-edited THP-1 monocytes with a ZNF334 truncation mutation, we discovered that the mutation reduced the interaction between ZNF334 and Hsp90, diminished the endogenous levels of the cold stress regulators Hsp90 and transient receptor potential melastatin 8 (TRPM8), disrupted ER protein folding response and redox homeostasis, and increased cold-induced NF-κB activation and secretion of the proinflammatory TRPM8+ mitochondria-containing extracellular vesicles in monocytes. Long-term cold avoidance alleviated the patient's cold-induced symptoms. In addition, treatment of ZNF334-truncated THP-1 cells with an Hsp90 inhibitor prevented cold-induced *TNF* and *NLRP3* upregulations. Our findings suggest ZNF334 as an essential regulator of cold-induced inflammation and oxidative stress, and Hsp90 ATPase inhibitors might be effective in the treatment of autoinflammatory diseases induced by repeated mild cold exposure.**

**Keywords** Autoinflammation; Cold; Mutation; Sensorineural Hearing Loss; ZNF334

**Subject Categories** Evolution & Ecology; Genetics, Gene Therapy & Genetic Disease; Immunology

## Introduction

Cold exposure alters cellular homeostasis and modulates immune responses (Jansky et al, 1996; Liu et al, 2022; Spiljar et al, 2021). Cold adaptation is an evolutionary mechanism used by different species to cope with dynamic environmental changes. It is reported that repeated cold water immersions increased the percentages of monocytes, activated lymphocytes, and plasma TNF-αlevels in athletic young men (Jansky et al, 1996). Dysregulation of cold stress response could lead to human inflammatory disorders, such as autoinflammatory diseases (AIDs) triggered by hours of general environmental cold exposure (Moltrasio et al, 2022), and cold urticaria, which could be induced in minutes after direct local cold contact (Maltseva et al, 2021). To date, the regulatory mechanism linking cold sensing and prevention of cold stress-induced hyperinflammation in humans is still unclear.

Cryopyrin-associated periodic syndromes (CAPS) encompass a spectrum of autoinflammatory disorders that result from dominant mutations in *NLRP3*, leading to hyperactivation of the NLRP3 inflammasome in monocytes and macrophages (Moltrasio et al, 2022; Romano et al, 2022). Familial cold autoinflammatory syndrome (FCAS) in the spectrum of CAPS is the prototypical disease due to cold exposure, which typically develops urticarial rash, fever, joint pain, and conjunctivitis at least 1–2 h after general cold exposure (Hoffman et al, 2001; Moltrasio et al, 2022). The symptoms of patients with severe forms of CAPS, such as Muckle–Wells syndrome (MWS) and neonatal-onset multisystem inflammatory disease (NOMID) could include progressive sensorineural hearing loss (SNHL), but were not associated with cold exposure (Kuemmerle-Deschner et al, 2015; Ma et al, 2022; Nakanishi et al, 2017). According to the literature, cold urticaria is typically associated with a positive ice cube test, whereas FCAS and other AIDs are not (Haas et al, 2004; Maltseva et al, 2021; Romano et al, 2022). Therefore, inflammasome hyperactivation

[1]College of Medicine, China Medical University, Taichung, Taiwan. [2]Rheumatic Diseases Research Center, China Medical University Hospital, Taichung, Taiwan. [3]Rheumatology and Immunology Center, China Medical University Hospital, Taichung, Taiwan. [4]Ph.D. Program in Translational Medicine and Rong Hsing Research Center for Translational Medicine, National Chung Hsing University, Taichung, Taiwan. [5]Department of Pathology, School of Medicine, Chung Shan Medical University and Chung Shan Medical University Hospital, Taichung, Taiwan. [6]Integrated Precision Health and Immunodiagnostic Center, Department of Laboratory Medicine, China Medical University Hsinchu Hospital, Zhubei City, Taiwan. [7]Biomedical Technology and Device Research Laboratories, Industrial Technology Research Institute, Hsinchu, Taiwan. [8]Institute of Biotechnology, National Tsing Hua University, Hsinchu, Taiwan. [9]Department of Pediatrics, National Taiwan University Hospital, College of Medicine, National Taiwan University, Taipei, Taiwan. [10]Genome and Systems Biology Degree Program, College of Life Science, National Taiwan University, Taipei, Taiwan. [11]Department of Research and Development, Show Chwan Healthcare System, Changhua, Taiwan. [12]Department of Laboratory Medicine, Chang Gung Memorial Hospital, Linkou, Taiwan. [13]Department of Pathology, China Medical University Hospital, Taichung, Taiwan. [14]Department of Biological Science and Technology, China Medical University, Taichung, Taiwan. [15]Cancer Biology and Precision Therapeutics Center, China Medical University, Taichung, Taiwan. [16]Graduate Institute of Biomedical Sciences, Research Center for Cancer Biology and Center for Molecular Medicine, China Medical University, Taichung, Taiwan. [17]Department of Pediatrics, China Medical University Hsinchu Hospital, Zhubei City, Taiwan. [18]Department of Biomedical Engineering and Environmental Sciences, National Tsing Hua University, Hsinchu, Taiwan. ✉E-mail: yangca@cgmh.org.tw

alone may be insufficient to explain the direct pathophysiology underlying cold-induced hyperinflammation.

Herein, we report an unusual case of a female patient who presented with middle-aged-onset symptoms triggered by repeated exposure to mild cold (air-conditioning at 25–27 °C). These symptoms included non-itchy urticaria-like skin rash, fever, arthralgia, and lymphadenopathy. The patient also presented with positive ice cube test results, acrocyanosis induced by local ice cube application, and progressive SNHL, which substantially affected her quality of life. Her symptoms overlapped with those of CAPS, acquired cold urticaria, Raynaud's disease, cold-induced acrocyanosis, and low-grade inflammation with cold-induced flare-ups. Through whole-exome sequencing and functional studies, we identified a *ZNF334* truncation mutation in monocytes as the novel genetic cause for this previously unreported case of a cold-induced AID with SNHL.

Zinc finger protein 334 (ZNF334) belongs to the C2H2-type zinc finger (ZnF) protein family. This family contains tandem repeats of C2H2-type ZnF motifs, which could interact with various molecules in multiple biological processes (Cassandri et al, 2017; Krishna et al, 2003). C2H2-type ZnF proteins have been reported to primarily function as transcription factors, and *ZNF334* has been described as a tumor-suppressor gene targeting signaling pathways involving tumor growth and metastasis (Cheng et al, 2022; Sun et al, 2022; Varnamkhasti et al, 2024; Yang et al, 2023). Downregulation of *ZNF334* expression has been reported in various cancer tissues and in the CD4 + T cells of patients with rheumatoid arthritis (Cheng et al, 2022; Li et al, 2024; Soroczynska-Cybula et al, 2011). Although treatment with tumor necrosis factor-α (TNF-α) was reported to potentially influence the expression of *ZNF334*, the immunological function of ZNF334 itself remains unknown (Henc et al, 2015). Therefore, in this study, we examined the effect of ZNF334 truncation mutation, which results in the loss of multiple C2H2-type ZnF motifs, on the regulation of monocyte homeostasis upon cold-induced oxidative stress and inflammation.

Our results indicate that ZNF334 truncation mutation reduced the interaction between ZNF334 and the cold stress regulator heat shock protein 90 (Hsp90), and diminished the endogenous intracellular levels of Hsp90 and transient receptor potential melastatin 8 (TRPM8). TRPM8 is the most established cold sensor involving the process of cold adaptation across vertebrates (Iftinca and Altier, 2020; Yang et al, 2020), and is known as a regulator of endoplasmic reticulum (ER)-mitochondria $Ca^{2+}$ shuttling related to cellular reactive oxygen species (ROS) homeostasis (Bautista et al, 2007; Bidaux et al, 2015; Bidaux et al, 2016). In this study, we found that human monocytic THP-1 cell lines expressing truncated ZNF334 lacking C2H2-type ZnF domains exhibited not only enhanced cold-induced *TNF/NFKB1/NLRP3/STAT3* signaling and necroptosis, but also excessive oxidative stress, as evidenced by the increased production of ROS and cold-triggered TRPM8+ mitochondria-containing extracellular vesicles (EVs). These results highlight a novel role of ZNF334 in linking cold sensing and regulation of cold-induced oxidative stress and inflammation in the human innate immune system. Our findings also suggest that ZNF334 truncation mutation results in priming and loss of negative feedback to cold stress in monocytes, which in turn lead to autoinflammation. We name this AID "ZNF334-associated cold-induced autoinflammatory syndrome (ZACAS)".

# Results

## Clinical phenotype and genetic analysis

A 55-year-old Han Chinese woman presented with recurrent fever; a non-itchy, migratory urticaria-like skin rash; and joint pain occurring 1–6 h after exposure to temperatures of 25–27 °C maintained by an air-conditioning system at her workplace. This pattern had persisted since she was 40 years old. The patient had been treated with 10 mg of levocetirizine, 400 mg of hydroxychloroquine, and 400 mg of celecoxib daily since the age of 46 years. However, her symptoms persisted. Skin biopsy of the patient's urticaria-like rash, performed at the age of 47 years, revealed mild perivascular noncuffing lymphohistiocytic cell infiltration involving the superficial vascular plexus and small periadnexal capillaries (Fig. 1A). Short-term administration of immunosuppressive agents, including steroids and methotrexate, was required to alleviate her fever and joint pain after cold exposure. When the patient reached 50 years of age, inflammatory lymphadenopathies were detected through positron emission tomography, and she started developing progressive bilateral SNHL. Therefore, long-term immunosuppressant therapy, including 10 mg of steroids daily and 15 mg of methotrexate weekly, was initiated. To determine the likelihood of urticarial vasculitis, another skin biopsy of the same type of urticaria-like rash was performed at the age of 51 years. It revealed neutrophilic and lymphocytic infiltrates in the walls of the dermal vessels (Fig. 1B), without the immune deposits of IgG, IgM, IgA, or C3. Masson's trichrome staining of the skin biopsy revealed no intravascular fibrin deposition (Fig. 1C). Immunofluorescence staining revealed perivascular-infiltrating cells, which were double-positive for phosphorylated-NF-κB p65 (p-p65) and nuclear STAT3, suggesting an ongoing inflammatory process (Fig. 1D). Laboratory examinations consistently revealed high erythrocyte sedimentation rates (ESR) and C-reactive protein (CRP) levels throughout the follow-up period (Fig. 1E). In June 2015, steroid and methotrexate (immunosuppressants) were discontinued for a short period as a trial. In addition, colchicine was added, and the dosage of celecoxib was increased, which resulted in a mild reduction in the patient's CRP level, but an abrupt increase in her ESR ("is − " period; Fig. 1E). At the 4th year of follow-up (at the age of 59 years), the patient's periodic cold-induced fever and arthralgia subsided after a long period of cold avoidance (by taking time off and predominantly staying at home) and continual immunosuppressive therapy. However, her urticaria-like skin rash recurred after short periods of exposure to low temperatures, including by staying for more than 15 min in a supermarket. She noticed that her rash was particularly prominent when she stood close to the dairy section. Considering that the patient's urticaria could be induced within less than 30 min by exposure to temperatures below 25 °C, we conducted an ice cube test to determine whether her clinical features were similar to those of cold urticaria. Our ice cube test was positive because the extent of urticaria induced by exposure to the hospital's air-conditioning system was exacerbated 5 min after the application of an ice bag. Acrocyanosis, circumoral cyanosis, and chillness developed 10 min after the application of the ice bag, but these symptoms resolved after the patient spent 30 min outdoors at 30 °C (Fig. 1F). To avoid recurrence, the patient began adopting a more stringent cold avoidance strategy, which helped alleviate her intermittent urticaria-like skin rash. At her last visit at the age of 60 years, she had no fever, arthralgia or urticarial-like skin rash because of her retirement and long periods of cold avoidance. However, her bilateral SNHL had

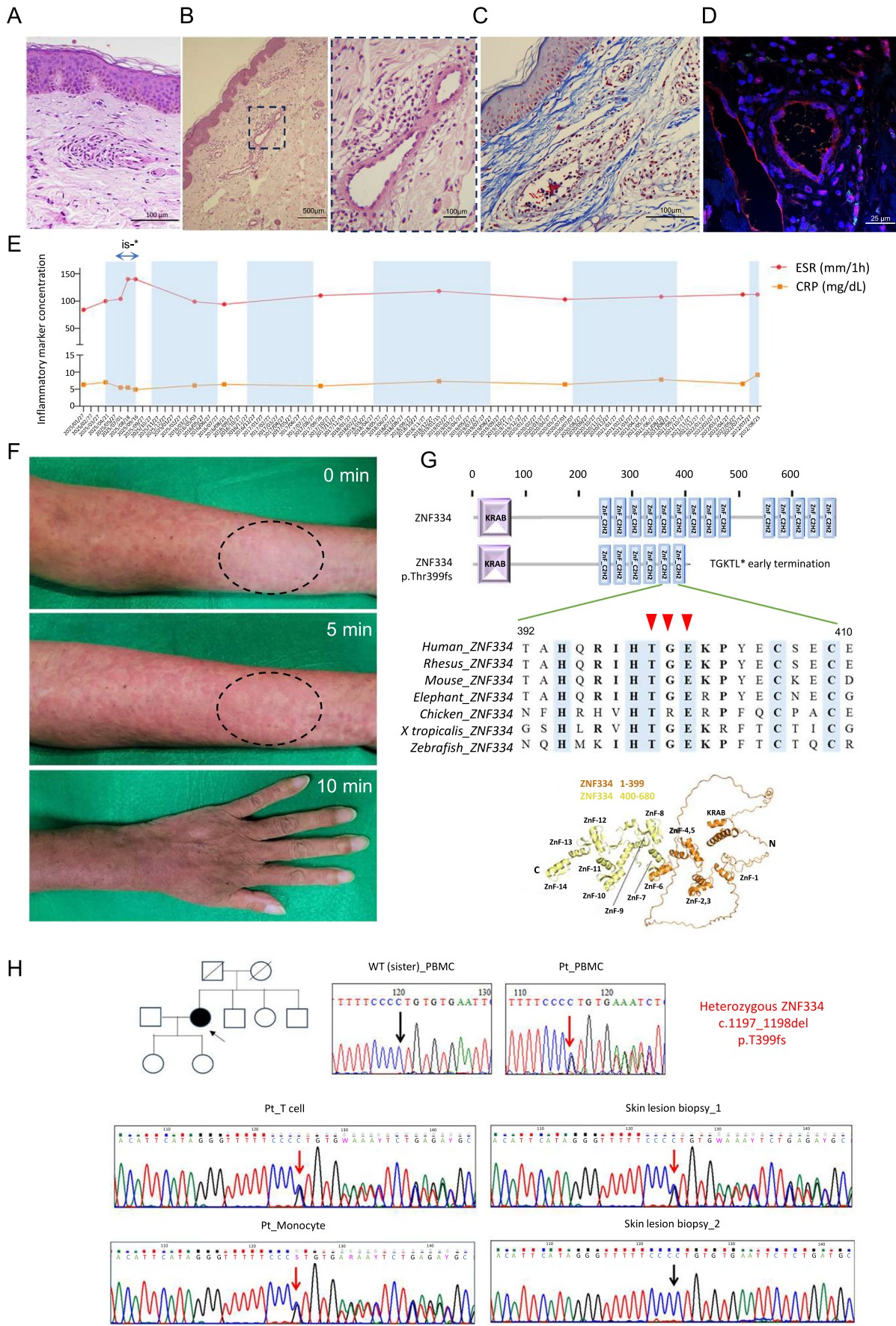

◄ **Figure 1.   Heterozygous *ZNF334* p.Thr399fs mutation was associated with a novel late-onset autoinflammatory disease and sensorineural hearing loss after repeated cold exposure.**

(A) Hematoxylin and eosin staining of an urticarial skin biopsy obtained after the patient had been treated with 10 mg levocetirizine, 400 mg hydroxychloroquine, and 400 mg celecoxib daily for 1 year and was exposed to cold air. Scale bar, 100 μm. (B) Hematoxylin and eosin staining of another skin biopsy of the urticaria-like rash performed at the age of 51 years. Right graph is the magnification of the selected area in the left graph. (C) Masson's trichrome staining of the skin biopsy taken at the age of 51 years. (D) A confocal micrograph showing the immunofluorescence signals of phospho-p65 (green) and STAT3 (red) staining in perivascular infiltrating leukocytes of another cold-induced urticarial skin biopsy obtained from the patient when she was receiving 10 mg steroids daily and 15 mg methotrexate (MTX) weekly for 1 year. Scale bar, 20 μm. (E) Levels of erythrocyte sedimentation rate (ESR) and C-reactive protein (CRP) detected in the patient's serum over a 7-year follow-up. The background blue color indicates the "cold + +++" periods: daily 8-h cold exposure at work or off work for <3 days, * is − : period without the use of immunosuppressant. Periods without color coding represent the time during cold avoidance (off work for >1 month with minimal cold exposure, such as going to the supermarket occasionally, with immunosuppressant). (F) Positive ice cube test. Ice cubes packed in an ice bag were placed on the forearm marked with an oval dash line for 5 min; the changes were observed at 5 and 10 min after the removal of the ice bag. (G) Upper graph: schematic of ZNF334 protein domains annotated using SMART (smart.embl-hidelberg.de). Alignment illustrating the degree of conservation of sites affected by the frameshift mutation between orthologs of ZNF334. Lower graph: structure of ZNF334 predicted using UniProt (uniprot.org). The *ZNF334* variant was predicted to cause a frameshift mutation, resulting in loss of the C-terminal 400–680 amino acid fragment, indicated in yellow. (H) Upper graph: pedigree showing the proband (indicated with an arrow) as the only affected individual within the family. Symbols with the diagonal line: deceased. Lower graph: Sanger sequencing of the *ZNF334* variant in the patient and in the healthy sister's PBMCs, in the patient's T cells (CD3 + T cells) and monocytes (CD14+ cells) isolated via FACS sorting, and in two skin biopsy samples of the patient (red arrow: heterozygous *ZNF334* frameshift mutation; black arrow: wild-type *ZNF334*).

severely progressed (85–90 dB higher than normal on an audiogram), and she still had an elevated CRP level (5.91 mg/dL).

Whole-exome sequencing of the patient's genome was performed to elucidate the genetic mechanisms underlying this unusual case of autoinflammation and progressive SNHL, induced by mild cold exposure. Although no known pathogenic variant was detected in either the germline or the mosaic analysis pipelines, two candidate variants with rare population frequencies and high predicted protein impacts were identified (Appendix Table S1). Of these two variants, the *ZNF334* frameshift mutation, which has a low frequency in the Taiwanese population (0.067%), was detected in the patient's blood cells, not in her healthy siblings', with this confirmed through Sanger sequencing (Fig. 1H). This heterozygous frameshift mutation (c.1197_1198del, encoding p.Thr399fs) was predicted to affect multiple C2H2-type ZnF domains of *ZNF334*. Two nucleotides were discovered to be deleted in *ZNF334*, that correspond to the mRNA sequence encoding threonine (399 T) and glycine (400 G). This deletion resulted in the conversion of glutamine (401E) to lysine, and led to the early termination of the protein, which in turn resulted in the loss of subsequent C2H2-type ZnF domains containing several highly conserved amino acids (Fig. 1G). As presented in Fig. 1G, the amino acids 401E, 406C, and 409C immediately following the frameshift mutation were conserved, whereas the flanking amino acids demonstrated variability across species. In addition to the patient's healthy siblings, one of her daughters was also tested as wild-type in *ZNF334*, with this confirmed using Sanger sequencing. However, no data were available on the patient's other child, who was working abroad, or on her parents, who were deceased (Fig. 1H). Although the sequencing depth and ratio of the *ZNF334* frameshift variant detected in the patient's blood cells were 42/104 and 40.38%, respectively, suggesting a heterozygous germline mutation pattern, the likelihood of this variant originating from acquired post-zygotic mosaicism could not be ruled out because of the late-onset nature of the disease. Therefore, *ZNF334* was further sequenced in monocytes (CD14 +) and T cells (CD3 +) isolated from the patient's peripheral blood mononuclear cells (PBMCs), and a heterozygous p.T399fs mutation was detected in both cell subsets (Fig. 1H). Notably, because no buccal swab sample was available, *ZNF334* Sanger sequencing was performed on two formalin-fixed, paraffin-embedded (FFPE) blocks of skin biopsy lesion samples. These samples were obtained from the

patient's back (at the age of 47 years), and chest (at the age of 51 years), and they primarily comprised epithelial cells and fibroblasts with a small number of infiltrating leukocytes. In contrast to expectations, a heterozygous *ZNF334* frameshift mutation was detected only in the skin biopsy sample obtained from the patient's back (Fig. 1H). Collectively, these findings suggest that the *ZNF334* frameshift mutation occurs during a postzygotic event, and that both lymphoid and myeloid blood cells carry the heterozygous *ZNF334* mutation, potentially contributing to disease pathogenesis.

## Cytokine and CITE-seq analyses revealed enhanced monocyte-mediated inflammation and dysregulated stress responses involving NF-ΚB and protein folding signaling

To investigate the effect of the *ZNF334* mutation on cold stress–associated inflammatory cytokine production, we analyzed the serum levels of TNF-α, IL-6, IL-1β, and extracellular Hsp90 (eHsp90) in samples obtained from the patient's sister (who had wild-type *ZNF334*, with samples collected at four time points when the patient's sister accompanied her to the hospital), other wild-type family members (brothers and daughter, with samples collected at a single time point), and the patient herself (samples collected under different cold exposure conditions). These conditions included daily 8-hour cold exposure at work or off work for <3 days without immunosuppressant use (S1, "cold + ++ is − "), with immunosuppressant use (S2, "cold + ++ is + "), and with cold avoidance (off work for >1 month with minimal cold exposure, such as occasionally visiting the supermarket, and with immunosuppressant use; S3, "cold+ is + "). Under all conditions, the serum levels of inflammatory cytokine were higher in the patient than in the wild-type healthy controls, with the highest levels observed during daily 8-h cold exposure without the use of immunosuppressants (Fig. 2A). Of note, in the patient's plasma, the levels of IL-1β and eHSP90 were markedly lowered during cold avoidance as compared with those during repeated cold exposure without the use of immunosuppressants (both $P = 0.028$); while the decrease in the levels of TNF-α and IL-6 by cold avoidance was less prominent.

To analyze the inflammatory cell signature, CITE-seq was performed using PBMCs collected from the healthy sister and the patient under two conditions: cold + ++ is+ (S2) and cold+ is+

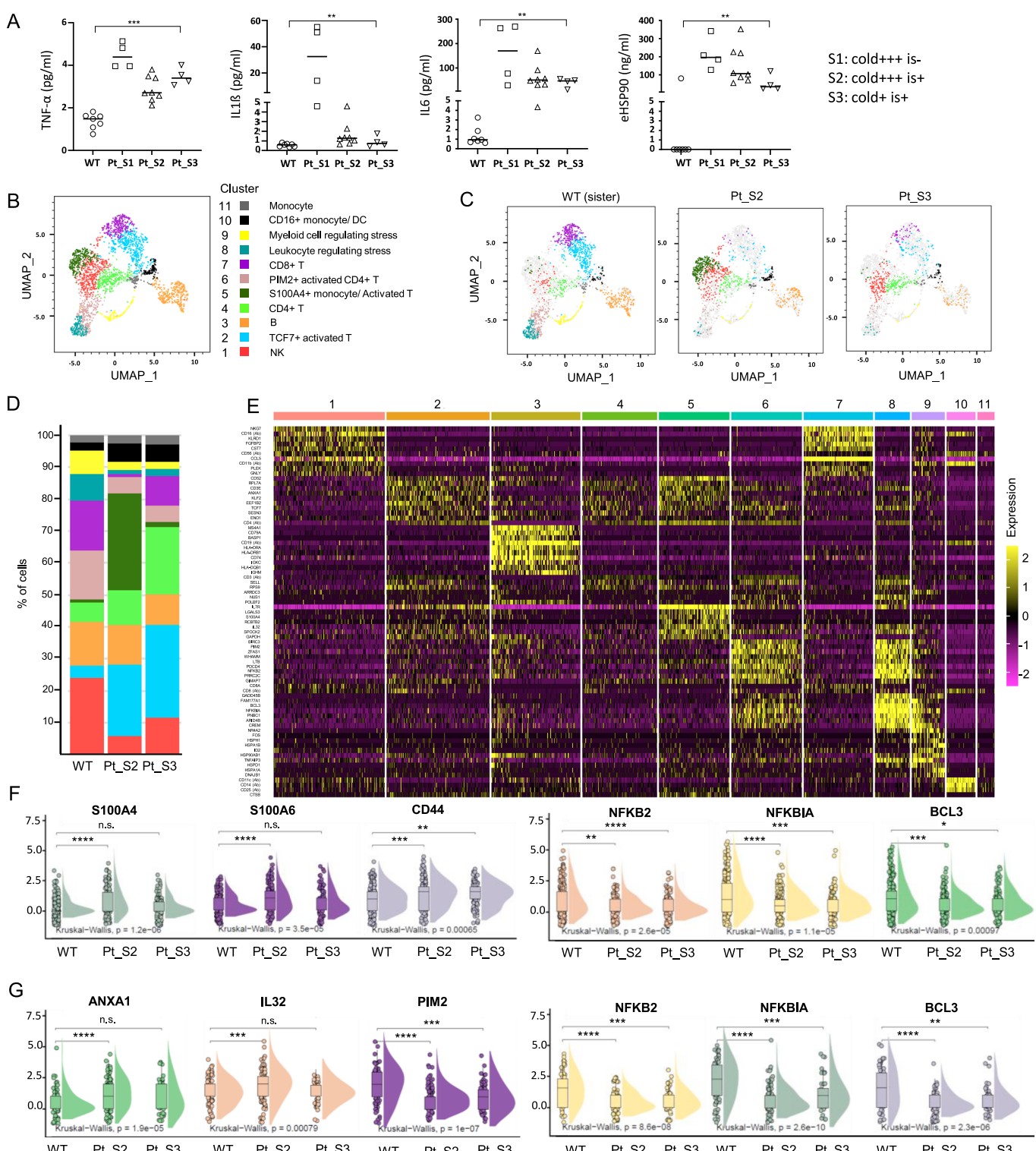

(S3). A total of 11 cell types were clustered in the uniform manifold approximation and projection plot (UMAP) (Fig. 2B). The top ten differentially expressed genes and cell surface markers (detected by 11 oligonucleotide-conjugated antibodies against cell surface markers, Fig. EV1A) per cluster are shown in the heatmap (Fig. 2E). Compared with her sister, the patient had increased proportions of

cluster 2 (TCF7+ activated T cells), cluster 4 (CD4 + T cells) and cluster 10 (CD16+ proinflammatory monocytes) in PBMCs collected under both conditions (Fig. 2C,D). Of note, when the patient's PBMCs were exposed to long periods of cold (S2), the percentage of cluster 5 (composed of S100A4 + S100A6+ mono-cytes and IL32 + ANXA1+ activated T cells) markedly increased

**Figure 2.** Serum cytokine and CITE-seq analyses of PBMCs derived from the patient (Pt) and her ZNF334 wild-type (WT) healthy family members.

(A) Serum levels of TNF-α, IL-6, L-1β, and eHsp90 in samples obtained from the patient's sister (ZNF334-WT), collected at four time points; and samples obtained from the patient's ZNF334-wild type brothers and daughter, collected at single time point; and those from the patient, collected at different time points under various cold exposure conditions ("cold + ++ is − ", n = 4; "cold + ++ is +", n = 8; and "cold+ is +", n = 4). The median value for each dataset is indicated by the horizontal line. *P < 0.05 and **P < 0.01: Kruskal–Wallis test. (B) Uniform manifold approximation and projection (UMAP) visualization and marker-based annotation of 11 distinct clusters (cell subtypes). (C) UMAP plots of PBMC cell subsets distributed in ZNF334-WT in the sister and patient under two cold exposure conditions ("cold + ++ is +" or "cold+ is +"), colored based on cluster identity. (D) Stacked bar plots of the PBMC cluster composition of each sample. Colors correspond to clusters depicted in (B). (E) Heatmap showing the normalized and scaled expression of the top ten differentially expressed genes per cluster. Adjusted P value < 0.05, calculated by the Wilcoxon rank-sum test, corrected by the B-H method. (F, G) Violin plots show the expression of S100A4, S100A6, CD44, NFKB2, NFKBIA, and BCL3 in CD45⁺ CD14⁺ monocytes, n = 592 (F); and the expression of ANAX1, IL32, PIM2, NFKB2, NFKBIA, and BCL3 in CD3 + CD3 + T cells, n = 225 (G), derived from ZNF334 wild-type cells isolated from the sister and patient under "cold + ++ is +" and "cold+ is +" conditions. In the box-and-whisker within violin plots, the horizontal lines represent the median, the box limits indicate the 25th and 75th percentiles, and the whiskers extend to 1.5× the interquartile range from the 25th and 75th percentiles. The Kruskal–Wallis test was performed to compare expression levels among the three groups, with adjusted P values shown in graphs. Two-tailed Wilcoxon tests were used to determine the difference between the two groups. *Adjusted P < 0.05; **adjusted P < 0.01; ***adjusted P < 0.001; **** adjusted P < 0.0001.

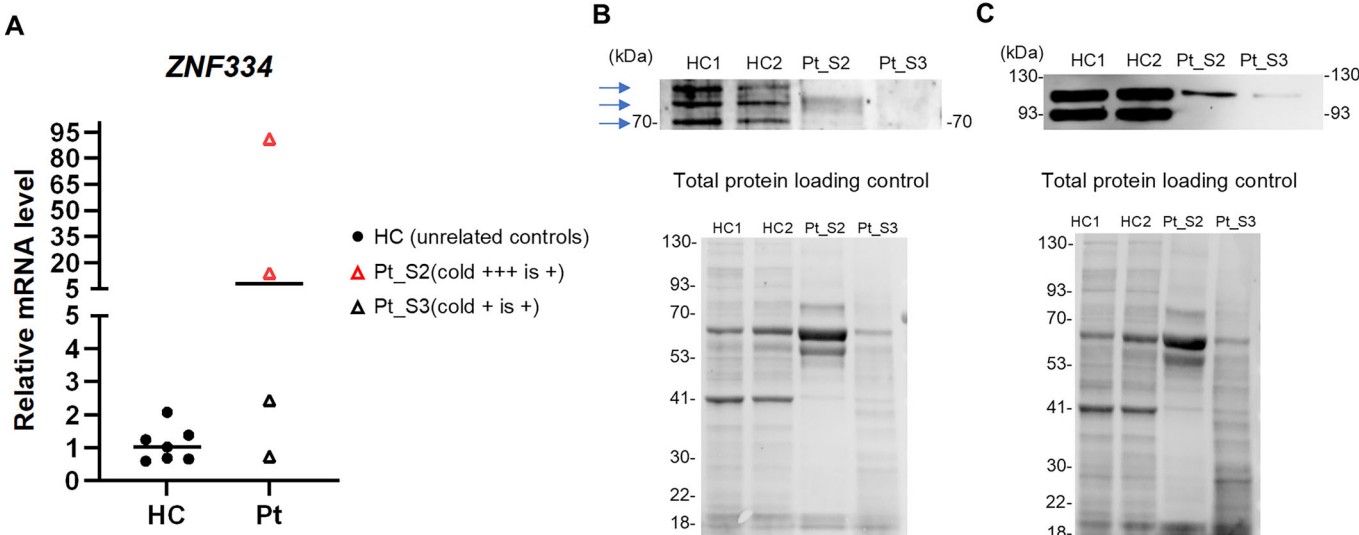

**Figure 3.** Heterozygous ZNF334 p.Thr399fs mutation affects the protein expression levels of ZNF334 and Hsp90 in monocytes.

(A) Relative mRNA expression of ZNF334 in monocytes of the patient (Pt) collected at four time points and in monocytes of seven unrelated healthy controls (HC) at baseline. Relative expression was calculated as the fold change compared with the mean basal expression level of HC. Lines represent medians. (B, C) Whole-cell lysates were prepared from monocytes separated from PBMCs derived from two unrelated healthy donors (healthy controls, HC1 and HC2) and those separated from PBMCs derived from the patient under two conditions ("cold + ++ is +" and "cold+ is +"). Immunoblotting was performed against ZNF334 (B) and Hsp90 α/β (C). Quantification of the major ZNF334 and Hsp90 bands on Western blots was performed using stain-free total protein as loading controls. Blue arrows indicate ZNF334 protein 70, 75, and 77 kDa isoforms listed in UniProt (www.uniprot.org).

compared with in the control and S3 conditions (Fig. 2D). In addition, the patient's PBMCs exhibited reduced proportions of myeloid subsets expressing genes involved in the regulation of NF-ĸB signaling, inflammatory responses, and protein folding (Figs. 2D and EV1B,C). Further gating on CD45⁺ CD14⁺ monocyte populations revealed increased expression levels of the monocyte activation markers S100A4, S100A6 and CD44 and reduced expression levels of the NF-ĸB signaling regulators NFKB2, NFKBIA, and BCL3 in the patient's samples (Fig. 2F). Gating on CD45 + CD3 + CD4 + T cells also revealed decreased expression levels of NF-ĸB signaling regulators, together with the downregulation of PIM2 and the upregulation of ANAX1 and IL32, particularly in S2 (Fig. 2G). These data suggest not only enhanced inflammation and dysregulated stress responses involving NF-ĸB and protein folding signaling in the patient's monocytes, but also enhanced Th1 activation.

## ZNF334 mutation reduced basal ZNF334 and Hsp90 protein expressions in monocytes

To evaluate the effect of the ZNF334 truncation mutation on the mRNA transcription and protein expression levels of ZNF334, we collected monocytes derived from the patient under "cold + ++ is +" and "cold+ is +" conditions and the unrelated healthy controls under baseline conditions. We observed that the ZNF334 mRNA expression was not lower in the patient's monocytes, and was highest in monocytes collected from the patient under the "cold + ++ is +" condition (Fig. 3A). Western blot analysis showed that compared with the unrelated healthy controls' samples, the patient's monocytes exhibited reduced expression of ZNF334 70, 75, and 77 kDa protein isoforms under both conditions (Fig. 3B). Consistent with the results regarding mRNA levels, the protein expression level of ZNF334 was lowest in the patient's

monocytes under the "cold+ is +" condition (Fig. 3B). Furthermore, since the patient exhibited increased eHsp90 secretion (Fig. 2A) and reduced transcription of genes involved in regulating chaperone-mediated protein folding and heat response (Fig. EV1C), we investigated the effect of the *ZNF334* mutation on Hsp90 expression in monocytes. Decreased Hsp90α/ß protein expression was observed in the patient's monocytes under the two conditions, with the lowest level detected under the "cold+ is +" condition (Fig. 3C). In addition to Hsp90, the expression of several abundant proteins commonly used as Western blot loading controls, such as β-actin and GAPDH, have been reported to be altered in proteotoxic/oxidative stress conditions (Eaton et al, 2013; Nakajima et al, 2017). Therefore, in this study, we used stain-free total protein as a loading control (Fig. 3C). These data revealed that ZNF334 p.Thr399fs mutation decreased baseline ZNF334 and Hsp90 protein expressions in the patient's monocytes.

## CRISPR/Cas9-mediated gene knockout revealed the role of ZNF334 in regulating cold-induced inflammation and cell death in THP-1 monocytes

To explore the role of *ZNF334* in regulating cold stress responses in monocytes, including inflammation mediated by TNF-α/NF-KB and IL-6/STAT3 signaling, cell proliferation, and cell death, we generated heterozygous and homozygous *ZNF334*-edited human THP-1 monocyte cell lines by using the CRISPR/Cas9 technology. These cells expressed a truncated protein (85 amino acids) for all ZNF334 isoforms that lacked all C2H2-type ZnF domains (Fig. EV2). Subsequently, we performed real-time PCR to analyze inflammation-related gene expression; determined IL-1β levels in the supernatants; and evaluated cell cycle regulation, apoptosis, and necroptosis in ZNF334-knockout and wild-type THP-1 monocytes at various time points during cold stimulation at 32 °C.

Compared with the findings in the ZNF334-wild-type THP-1 monocytes, the baseline RNA levels of *TNF*, *RIPK3*, *NFKB1*, and *STAT3* were markedly increased in the heterozygous (ZNF334$^{+/-}$) and homozygous (ZNF334$^{-/-}$) knockout cells (Fig. EV3). Upon stimulation at 32 °C, the highest expression of inflammation-related genes was observed in ZNF334$^{+/-}$ THP-1 cells exposed to cold stimulation for 6–8 h (Fig. 4A–G). We noted significant differences between the baseline expression levels of *TNF-α*, *RIPK3*, *NFKB1*, and *STAT3* in ZNF334 wild-type cells and the levels observed after 6 h of cold stimulation in ZNF334 wild-type, ZNF334$^{+/-}$, and ZNF334$^{-/-}$ cells (Fig. 4A,C,D,F). Furthermore, ZNF334$^{+/-}$ and ZNF334$^{-/-}$ THP-1 monocytes produced higher levels of IL-1β detected in the culture supernatant than did ZNF334 wild-type cells both at 37 °C ($P = 0.0007$) and after 24 h of 32 °C cold stimulation ($P = 0.0032$; Fig. 4H). To evaluate potential off-target effects in CRISPR/Cas9 editing, we tested another ZNF334$^{+/-}$ THP-1 clone and observed similar effects on NF-KB and STAT3 signaling (Appendix Fig. S1).

In cell cycle analysis, we observed higher proportions of ZNF334$^{+/-}$ and ZNF334$^{-/-}$ cells in the S and G2 phases and lower proportions of these cells in the G0/G1 phase at baseline (all $P = 0.0002$; Fig. 4I–K). Upon cold stimulation at 32 °C, the proportions of cells in the S and G2 phases increased significantly, and those of cells in the G0/G1 phase decreased in the ZNF334 wild-type, ZNF334$^{+/-}$, and ZNF334$^{-/-}$ THP-1 cells over time. The most prominent change occurred in ZNF334$^{-/-}$ cells after 7 h of

cold exposure. The median proportions of cells in ZNF334 wild-type, ZNF334$^{+/-}$, and ZNF334$^{-/-}$ THP-1 cells were 60.81%, 39.33%, and 15.62% in the G0/G1 phase ($P = 0.0002$; Fig. 4I); 33.90%, 45.02%, and 54.22% in the S phase ($P = 0.0002$; Fig. 4J); and 3.89%, 12.45%, and 26.36 in the G2 phase ($P = 0.0002$; Fig. 4K), respectively.

We examined cell death by using live cell imaging (Fig. EV4). Baseline cell membrane staining revealed that membrane integrity was impaired in ZNF334-knockout cells, particularly in the homozygous knockout THP-1 cells ($P = 0.0002$; Fig. 4L). The percentages of early apoptotic cells ($P = 0.0005$) and necroptotic cells ($P = 0.0012$) were higher in the ZNF334$^{+/-}$ and ZNF334$^{-/-}$ groups than in the wild-type group before cold stimulation (Fig. 4M,N). After 7 h of exposure to 32 °C, the percentage of early apoptotic cells increased among wild-type THP-1 monocytes. However, the highest percentages of early apoptotic cells were still detected in the ZNF334$^{+/-}$ and ZNF334$^{-/-}$ groups ($P = 0.015$; Fig. 4M).

To further evaluate the impact of ZNF334 truncation mutation on the expression of Hsp90, p-p65, and p-STAT3 (Tyr705), we performed the immunofluorescence staining of these molecules in ZNF334 wild-type and ZNF334-knockout THP-1 monocytes exposed to or not exposed to 8 h of cold stimulation at 32 °C. The expression of Hsp90 increased in viable ZNF334 wild-type, ZNF334$^{+/-}$, and ZNF334$^{-/-}$ THP-1 monocytes after 8 h of cold stimulation at 32 °C compared with their baseline levels (all $P < 0.0001$; Fig. 4O). The Hsp90 protein expression was lower in ZNF334$^{+/-}$ and ZNF334$^{-/-}$ cells than in wild-type cells both before ($P < 0.0001$) and after 8 h ($P < 0.0001$) of 32 °C treatment (Fig. 4O). Furthermore, after 8 h of cold stimulation at 32 °C, increased nuclear levels of p-p65 ($P = 0.006$; Fig. 4P) and p-STAT3 ($P < 0.0001$; Fig. 4Q) were detected in ZNF334 wild-type monocytes compared with their baseline levels. ZNF334-knockout THP-1 monocytes exhibited significantly higher nuclear levels of p-p65 (37 °C, $P < 0.0001$; 32 °C, $P = 0.0003$; Fig. 4P) and p-STAT3 (37 °C, $P < 0.0001$; 32 °C, $P < 0.0001$; Fig. 4Q) than did ZNF334 wild-type cells. At baseline, ZNF334$^{+/-}$ THP-1 monocytes had the highest level of nuclear p-p65, whereas ZNF334$^{-/-}$ cells had the highest level of nuclear p-STAT3 (Fig. 4P,Q).

The results conclude that THP-1 monocytes harboring truncated ZNF334 with the loss of all C2H2-type ZnF domains had reduced intracellular Hsp90 expression, enhanced baseline and cold-induced inflammation, an interrupted cell membrane, and a higher rate of cell death.

## RNA-seq of ZNF334-edited THP-1 monocytes revealed the involvement of ZNF334 in maintaining the regulatory protein folding machinery and suppressing cold-induced inflammatory response

To further investigate the potential molecular mechanisms of *ZNF334* truncation mutation on monocyte responses to cold stress, we performed RNA-seq analysis on ZNF334 wild-type, ZNF334$^{+/-}$, and ZNF334$^{-/-}$ THP-1 monocytes before and after 8 h of cold stimulation at 32 °C. In addition, differential gene expression analyses were conducted to compare the average normalized count per gene between ZNF334$^{+/-}$ and wild-type, and between ZNF334$^{-/-}$ and wild-type samples under both conditions. These analyses were performed using multiple *t* tests with

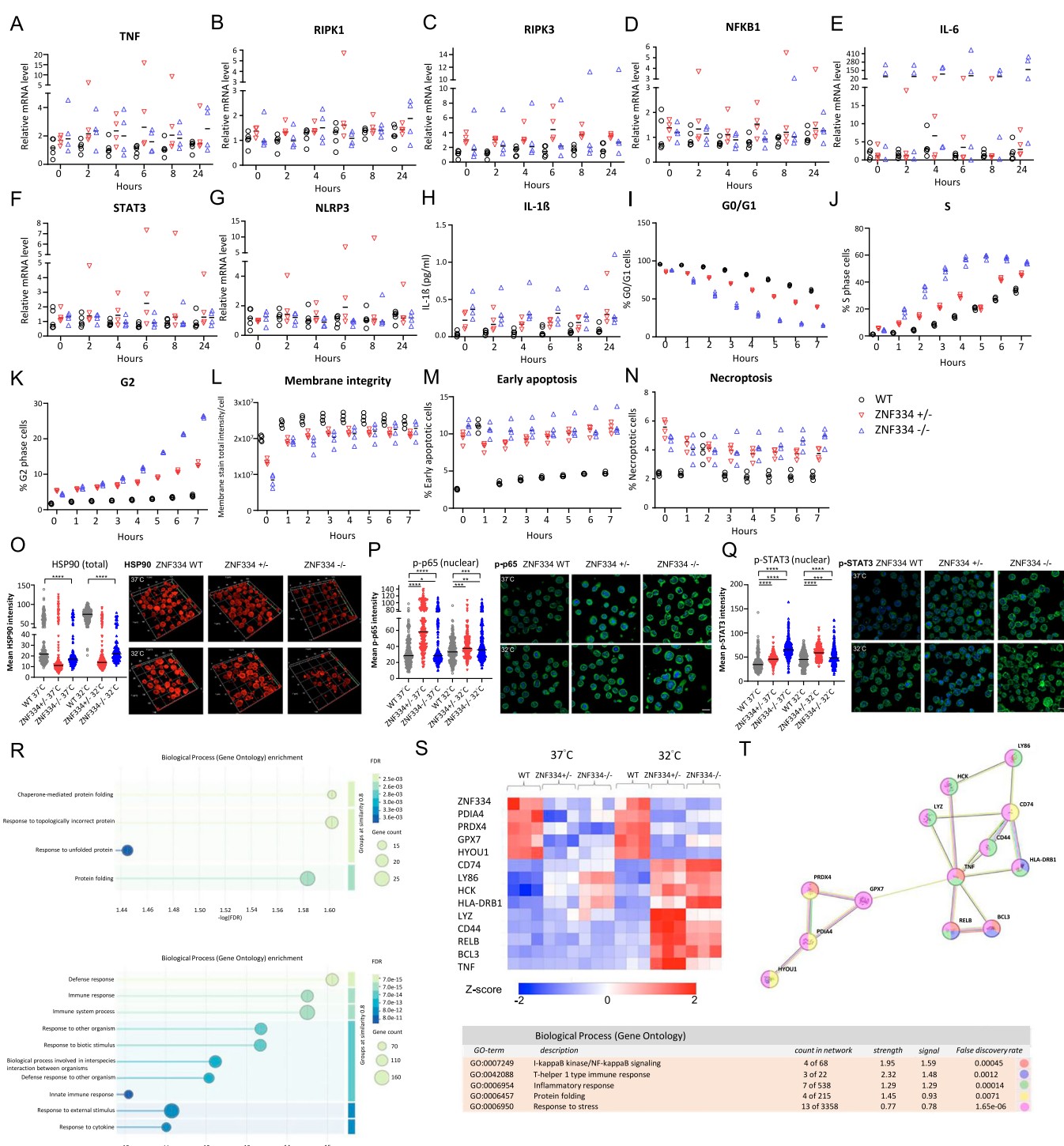

Benjamini–Hochberg (B-H) adjustments for corrected *P* values. Network and functional enrichment analyses of genes upregulated or downregulated in ZNF334-edited THP-1 monocytes (adjusted *P* value (FDR) < 0.05 and |log₂ fold change| ≥0.8 compared with ZNF334 wild-type cells) were performed using String v.12.0 (sting-db.org). We identified the top three enriched downregulated biological pathways (Gene Ontology) were chaperone-mediated protein folding, response to topologically incorrect protein, and

response to unfolded protein; whereas the top three enriched upregulated biological pathways were defense response, immune response, and immune system process (Fig. 4R). Specifically, the gene expression levels of key endoplasmic reticulum (ER)-resident enzymes responsible for antioxidative activity and oxidative protein folding (Wang et al, 2014; Zito et al, 2010), such as peroxiredoxin 4 (*PRDX4*), glutathione peroxidase 7 (*GPX7*), and protein disulfide isomerase 4 (*PDIA4*), were decreased in ZNF334⁺/⁻ and ZNF334⁻/⁻

**Figure 4.  Analyses of expression levels of inflammation-related genes, secretion of IL-1β, levels of cell death, cell cycle, and immunofluorescence staining of Hsp90, p-p65, and p-STAT3 in ZNF334-knockout THP-1 cells.**

(A–G) Relative mRNA expression levels of TNF (A), RIPK1 (B), RIPK3 (C), NFKB1 (D), IL-6 (E), STAT3 (F), and NLRP3 (G) in ZNF334 wild-type ($\circ$, $n = 5$), ZNF334 $+/-$ ($\triangledown$, $n = 4$), and ZNF334 $-/-$ ($\triangle$, $n = 4$) THP-1 monocytes at 37 °C and at 2, 4, 6, 8, and 24 h of cold stimulation at 32 °C. Relative expression was calculated as the fold change compared with the mean baseline (37 °C) expression level in ZNF334 wild-type THP-1 cells. Lines represent median values. WT wild-type. (H) IL-1β levels measured in the culture supernatant of ZNF334 wild-type, ZNF334 $+/-$, ZNF334 $-/-$ THP-1 cells at different time points of 32 °C cold stimulation. Lines represent medians, $n = 5$. (I–N) Proportions of cells in the G0/G1 phase (I), S phase (J), and G2 phase (K) and the integrity of the cell membrane (L), proportion of early apoptotic cells (M), proportion of necroptotic cells (N) were analyzed every hour upon 32 °C cold stimulation for 7 h through live cell imaging performed using a wide-field fluorescence microscope. ZNF334 wild-type ($\circ$), ZNF334 $+/-$ ($\triangledown$), and ZNF334 $-/-$ ($\triangle$); Lines represent median values, $n = 4$. (O) Quantitative analysis of Hsp90 protein levels (mean fluorescence intensity per cell) in ZNF334 wild-type, ZNF334$^{+/-}$, and ZNF334$^{-/-}$ THP-1 cells at 37 °C ($n = 173$, $n = 195$, $n = 119$ cells) or after 8 h of cold stimulation at 32 °C ($n = 194$, $n = 194$, $n = 115$ cells) and representative 3D images. Staining fluorescent intensities were analyzed for cells shown in 3–6 microscopic images acquired from slides made from each THP-1 cell line incubated at 37 °C or at 32 °C in two independent experiments. (P) Quantitative analysis of the nuclear levels of p-p65 (mean fluorescence intensity per nuclei) in ZNF334 wild-type, ZNF334$^{+/-}$, and ZNF334$^{-/-}$ THP-1 cells at 37 °C ($n = 223$, $n = 214$, $n = 222$ cells) or after 8 h of cold stimulation at 32 °C ($n = 226$, $n = 231$, $n = 231$ cells) and representative confocal microscopy images. p-p65, green; DAPI, blue. Scale bar, 10 μm. Staining fluorescent intensities were analyzed for cells shown in 3–6 microscopic images acquired from slides made from each THP-1 cell line incubated at 37 °C or at 32 °C in two independent experiments. (Q) Quantitative analysis of the nuclear levels of p-STAT3 (mean fluorescence intensity per nuclei) in ZNF334 wild-type, ZNF334$^{+/-}$, and ZNF334$^{-/-}$ THP-1 cells at 37 °C ($n = 265$, $n = 265$, $n = 242$ cells) or after 8 h of cold stimulation at 32 °C ($n = 265$, $n = 238$, $n = 252$ cells), and representative confocal microscopy images. p-STAT3, green; DAPI, blue. Scale bar, 10 μm. Staining fluorescent intensities were analyzed for cells shown in 7–9 microscopic images acquired from slides made from each THP-1 cell line incubated at 37 °C or at 32 °C in two independent experiments. $\sqcap$: P values were calculated using the Kruskal–Wallis tests; $-$: P values were calculated using the two-tailed Mann–Whitney U tests. The lines represent median values. $^*P < 0.05$; $^{**}P < 0.01$; $^{***}P < 0.001$; $^{****}P < 0.0001$. (R) Upper graph: downregulated biological pathways in ZNF334-knockout THP-1 monocytes. Lower graph: upregulated biological pathways in ZNF334-knockout THP-1 monocytes. Pathway enrichment analysis performed using String v.12.0 (sting-db.org). (S) Differentially expressed genes shown in z scores in heatmap, analyzed via Morpheus (software.broadinstitue.org). (T) Network analysis of genes depicted in (S), performed using String v.12.0 (sting-db.org).

monocytes as compared with ZNF334 wild-type at 37 °C and after 32 °C cold stimulations (Fig. 4S). The expressions of genes related to inflammatory responses and monocyte activations, including *CD74*, lymphocyte antigen 86 (*LY86*), hematopoietic cell kinase (*HCK*), and *HLA-DRB1* were already upregulated at baseline and further increased after cold stimulations in ZNF334$^{+/-}$ and ZNF334$^{-/-}$ THP-1 monocytes (Fig. 4S). Of note, genes involving NF-κB signaling pathway, like *RELB* and *TNF*, were markedly upregulated in cold-stimulated ZNF334$^{+/-}$ and ZNF334$^{-/-}$ THP-1 cells (Fig. 4S). Network analysis of the genes shown in the heatmap using String v.12.0 revealed that these molecules were all related to the pathway of response to stress, and potential interactions were identified among them (Fig. 4T). These data suggest that wild-type ZNF334 is required to maintain the ER protein folding response necessary to regulate redox homeostasis, and plays a role in suppressing cold-induced NF-κB activation in monocytes.

## ZNF334 truncation in THP-1 monocytes increased cold-induced oxidative stress, reduced the expression of endogenous TRPM8, and promoted the secretion of TRPM8+ extracellular vesicles containing mitochondria upon cold stimulation

To investigate the effect of the *ZNF334* truncation mutation on cellular oxidative stress, we analyzed the intracellular level of ROS in ZNF334 wild-type and ZNF334-knockout THP-1 monocytes before and after 2 h of cold stimulation at 32 °C. At 37 °C baseline, both ZNF334$^{+/-}$ and ZNF334$^{-/-}$ THP-1 monocytes had elevated levels of ROS as compared with ZNF334 wild-type monocytes (Fig. 5A). After cold stimulation, the intracellular ROS levels were further increased in all THP-1 monocytes with or without ZNF334 truncation, but no significant difference was detected between ZNF334 wild-type and ZNF334-knockout THP-1 monocytes after 2 h of 32 °C treatment (Fig. 5A).

Since TRPM8 is the main cold sensor and regulator of cellular ROS homeostasis (Bidaux et al, 2016; Iftinca and Altier, 2020), we then examined the expression levels of TRPM8 in ZNF334-knockout THP-1 monocytes through immunofluorescence confocal microscopy. TRPM8 expression was detected on the plasma membrane and ER in THP-1 monocytes (Fig. EV5). Overall TRPM8 levels were found to be significantly lower in ZNF334-knockout THP-1 monocytes than in ZNF334 wild-type cells before cold stimulation ($P < 0.0001$; Fig. 5B). Although the total and nuclear levels of TRPM8 increased after 8 h of 32 °C treatment in the three THP-1 cell lines (ZNF334 wild-type, ZNF334$^{+/-}$, and ZNF334$^{-/-}$ monocytes), lower levels were still detected in ZNF334-knockout cells ($P < 0.0001$; Fig. 5B).

While examining the expression of TRPM8 in THP1 cells under confocal microscopy, we noted the presence of TRPM8+ extracellular vesicles (EVs)/microvesicles ranging in size from 1 to 5 μm. It has been reported that damaged mitochondria induced by oxidative stress could be segregated into large EVs called exophers in *C. elegans* (Melentijevic et al, 2017), and activated human monocytes could also release microvesicle-embedded mitochondria, which trigger inflammatory responses in endothelial cells (Puhm et al, 2019). Therefore, we further analyzed the mitochondrial component in TRPM8 + EV using MitoTracker staining under confocal microscopy. We found that the secretion of 1–5 μm TRPM8+ mitochondria+ EVs was markedly increased in ZNF334 knockout THP-1 monocytes upon 8 h of cold stimulation, as compared with ZNF334 wild-type monocytes ($P = 0.0029$; Fig. 5C,D).

To examine the impact of the exophers on the differentiation and activation of other monocytes, we isolated microvesicles secreted by cold-stimulated ZNF334 wild-type, ZNF334$^{+/-}$ and ZNF334$^{-/-}$ THP-1 cells, ranging in size from 0.45 to 1 μm via qEV Isolation Columns (Izon Science, Christchurch, New Zealand) and membrane filters, then applied the respective isolated microvesicles to ZNF334 wild-type THP-1 monocytes and collected the supernatant after 48 h of incubation at 37 °C for cytokine analysis. We found that as compared with cold-stimulated wild-type-derived exophers, microvesicles derived from 32 °C-treated ZNF334-

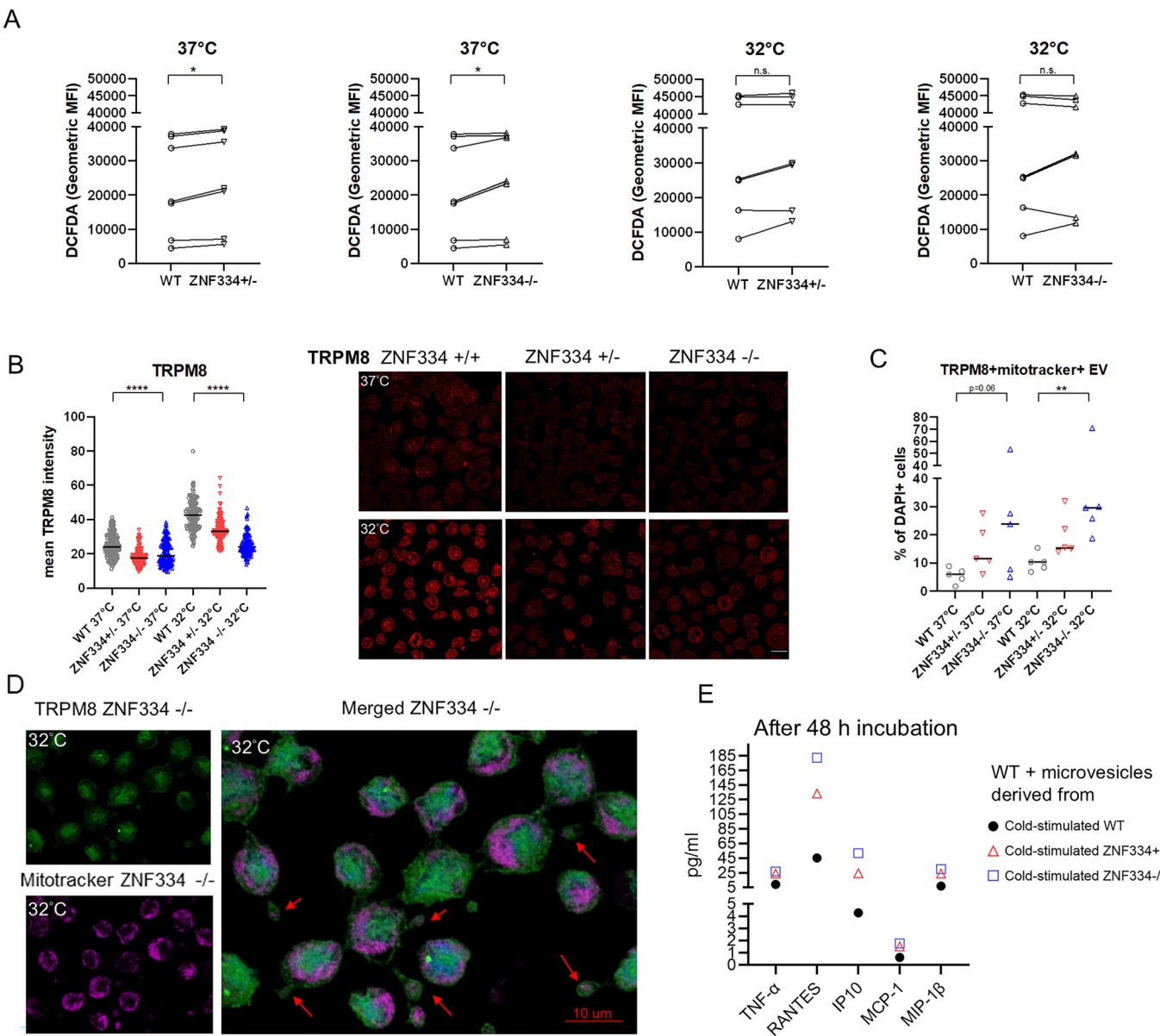

**Figure 5. ZNF334 knockout in THP-1 monocytes increased ROS production, reduced intracellular TRPM8 expression, and enhanced the secretion of TRPM8+ extracellular vesicles containing mitochondria.**

(A) Levels of ROS measured by the detection of DCFDA+ cells using flow cytometry. Geometric DCFDA mean fluorescence intensities (Geometric MFI) detected in ZNF334 wild-type, ZNF334$^{+/-}$, and ZNF334$^{-/-}$ THP-1 cells at 37 °C or at 32 °C are presented. Lines represent paired experiments. P values were calculated using the two-tailed Wilcoxon signed-rank test. *P < 0.05, n = 7 experiments (B) Quantitative analysis of the TRPM8 protein level analyzed by calculating the mean fluorescence intensity per cell in ZNF334 wild-type, ZNF334$^{+/-}$, and ZNF334$^{-/-}$ THP-1 cells at 37 °C (n = 193, n = 196, n = 167 cells) or after 8 h of cold stimulation at 32 °C (n = 162, n = 172, n = 152 cells). ****P < 0.0001: Kruskal–Wallis test. The lines represent median values. Staining fluorescent intensities were analyzed for cells shown in 4–5 microscopic images acquired from slides made from each THP-1 cell line incubated at 37 °C or at 32 °C in two independent experiments. Right: representative confocal microscopy images of TRPM8 expression in THP-1 cells. Scale bar, 10 µm. (C) THP-1 monocytes were stained with fluorescent-conjugated antibody against TRPM8, Mitotracker (mitochondria stain), and DAPI, and were examined under confocal microscopy with Z-stacking. Percentage of TRPM8+Mitotracker+ extracellular vesicles of 1–5 µm are expressed as a fraction of DAPI+ nucleated cells in each confocal image with z-stacking and 3D deconvolution. Lines represent medians, **P < 0.01 using Kruska-Wallis test, n = 5 experiments (D) Representative images of TRPM8+ extracellular vesicles containing mitochondria (red arrows). TRPM8, green; Mitotracker, magenta; DAPI, blue. Scale bar, 10 µm. (E) Microvesicles secreted by cold-stimulated (8 h at 32 °C)-ZNF334 wild-type, ZNF334$^{+/-}$ and ZNF334$^{-/-}$ THP-1 cells, ranging in size from 0.45 to 1 µm were isolated and transferred to ZNF334 wild-type THP-1 monocytes, and the supernatants were collected 48 h after incubation at 37 °C for cytokine analysis.

knockout monocytes stimulated the ZNF334 wild-type THP-1 cells to produce more TNF-α and chemokines IP-10 (CXCL10), RANTES (CCL5), MCP-1, and MIP1-ß in this single experiment (Fig. 5E).

These data indicate that ZNF334 truncation in THP-1 monocytes reduces the cellular expression of TRPM8, which is further associated with excessive oxidative stress and cold-induced secretion of TRPM8+ mitochondria-containing EVs with proinflammatory potentials and chemotactic activities for monocytes and lymphocytes.

## ZNF334 p.Thr399fs mutation in primary monocytes upregulated cold-induced expressions of TNF and NLRP3, and reduced the binding affinity of ZNF334 to Hsp90

In order to investigate whether the exact ZNF334 p.Thr399fs mutation has similar impact on cold-induced inflammatory gene expressions as those observed in THP-1 monocytes with CRISPR/Cas9-mediated ZNF334 truncation that lacked all C2H2-type zinc finger domains, we performed in vitro transcription assays producing mRNAs from ZNF334 wild-type or ZNF334 mutant (c.1197_1198del, encoding p.Thr399fs) cDNA clone in primary monocytes derived from healthy controls (Fig. 6A). We found that overexpression of the ZNF334 p.Thr399fs mutant enhanced baseline monocyte expressions of TNF, RIPK1, RIPK3, and NLRP3, as compared with the levels detected in monocytes overexpressing the wild-type ZNF334 (Fig. 6B). After 6 h of cold stimulation, the levels of TNF and NLRP3 were markedly increased in monocytes expressing the ZNF334 p.Thr399fs mutant, consistent with cold-induced hyperinflammation (Fig. 6B).

Next, since Hsp90 is known to be the major stress regulator and an abundant chaperone able to interact with other proteins to maintain cellular homeostasis (Bhattacharya et al, 2022), and we have detected dysregulations in Hsp90 expression, chaperone-mediated signaling and cold stress responses in monocytes harboring the truncated ZNF334, we further studied if ZNF334 could interact with Hsp90, and if ZNF334 p.Thr399fs truncation mutation would impair this interaction, contributing to the observed cellular immunophenotype. Prediction of the protein-ligand complex binding affinity between ZNF334-wild type and Hsp90α or Hsp90ß homodimer using AlphaFold-Multimer revealed potential binding of ZNF334 with Hsp90α homodimer. Yet, when forming a complex with Hsp90α homodimer, ZNF334 mutant was predicted to have smaller interface and less hydrogen bonds than ZNF334 wild-type, suggesting decreased binding affinity (Fig. 6C). Of note, the ZNF334 p.Thr399fs mutant was predicted to lose its interaction with the N-terminal ATPase domain of Hsp90 (Fig. 6C). We then evaluated the impact of ZNF334 p.Thr399fs on the interaction with Hsp90 via immunoprecipitation (IP)/Western analyses on monocytes derived from the patient and from healthy controls. Although the Immunoprecipitation using anti-Hsp90α/ß antibody detected interactions of Hsp90α/ß with ZNF334 in both patient's and healthy control's monocytes, immunoprecipitation using anti-ZNF334 antibody only detected Hsp90α/ß in healthy controls, not in patient's monocytes (Fig. 6D,E).

To further characterize the physiological changes in the interactions of Hsp90 with ZNF334 and other cold stress regulators identified in our monocyte model of cold stimulation, we performed IP/Western analyses on monocytes derived from healthy controls with or without in vitro 8 h of 32 °C cold exposure. Hsp90 interacted with ZNF334, TRPM8 (83 kDa and 41 kDa isoforms) and IKKα at 37 °C and after 32 °C cold stimulations (Fig. 6F–H). The binding between ZNF334 and Hsp90 decreased (Fig. 6F), while the interaction between Hsp90 and IKKα increased after cold exposure (Fig. 6G). These results identified ZNF334 and TRPM8 as novel proteins interacting with Hsp90, and suggest that the reduced interaction between ZNF334 and Hsp90 might contribute to the enhanced inflammatory responses.

Together, our data of monocytes support a model on how ZNF334 wild-type maintain cellular homeostasis upon cold stress (Fig. 7A), and how ZNF334 truncation mutation alters its interaction with Hsp90, disrupts basal homeostasis, and leads to excessive oxidative stress and inflammation after cold exposures (Fig. 7B).

## Treatment of ZNF334⁺/⁻ THP-1 monocytes with inhibitor against TNF-α, Hsp90, or TRPM8 revealed differential effects on the attenuation of cold-induced proinflammatory cytokine productions

Based on our proposed model that ZNF334 truncation mutation results in dysfunction of ZNF334 and dysregulation of Hsp90 and TRPM8, contributing to cold-induced hypersecretion of inflammatory cytokines, such as TNF-α, IL-1ß, IL-6, and extracellular Hsp90 (eHsp90), together with hyperproduction of potentially proinflammatory TRPM8+mitochondria+ microvesicles (Fig. 7), we further evaluated the effects of inhibitors against TNF-α (adalimumab), against Hsp90/eHsp90 (geldanamycin, an inhibitor of the Hsp90 N-terminal ATP binding site), or against TRPM8 (AMTB) on the attenuation of cold-stimulated upregulation of TNF, NLRP3 (upstream of IL-1ß) and IL-6 in THP-1 monocytes harboring wild-type ZNF334 or having heterozygous ZNF334 truncation mutation (ZNF334⁺/⁻). We did not test the effect of IL-6 inhibitors in our study due to the previous patient's history of severe allergic reaction/adverse effect to anti-IL-6 receptor antibody treatment. Furthermore, considering that the patient's serum IL-1ß level could be markedly decreased using immunosuppressants and avoiding cold exposures (Fig. 2A), and the high cost of anti-IL-1ß antibody treatment which created barriers to accessibility, we did not test the effect of IL-1ß inhibitor, either. As shown in Fig. 8, pretreatment of ZNF334 wild-type THP-1 monocytes with 10 μg/ml adalimumab had a trend of attenuation in 2 h cold stimulation-induced increase in TNF mRNA production as compared with those treated with heat-inactivated adalimumab control (P = 0.06; Fig. 8A); while pretreatment of ZNF334 wild-type monocytes with 2 μM geldanamycin or 10 μM AMTB significantly attenuated the 24 h cold stimulation-induced TNF upregulation as compared with those treated with DMSO only (P = 0.031; Fig. 8B; and P = 0.015; Fig. 8C). Significant attenuation of cold-induced hyperproduction of TNF was observed in ZNF334⁺/⁻ monocytes pretreated with 2 μM geldanamycin or 10 μM AMTB (P = 0.031; Fig. 8B; and P = 0.015; Fig. 8C). Of note, cold-induced NLRP3 hyperactivation in ZNF334⁺/⁻ monocytes could only be markedly attenuated by pretreatment with geldanamycin (P = 0.031; Fig. 8B). Pretreatment with AMTB, however, enhanced the cold-induced NLRP3 hyperactivation in ZNF334⁺/⁻ monocytes (P = 0.015; Fig. 8C). None of the three inhibitors showed significant effect on

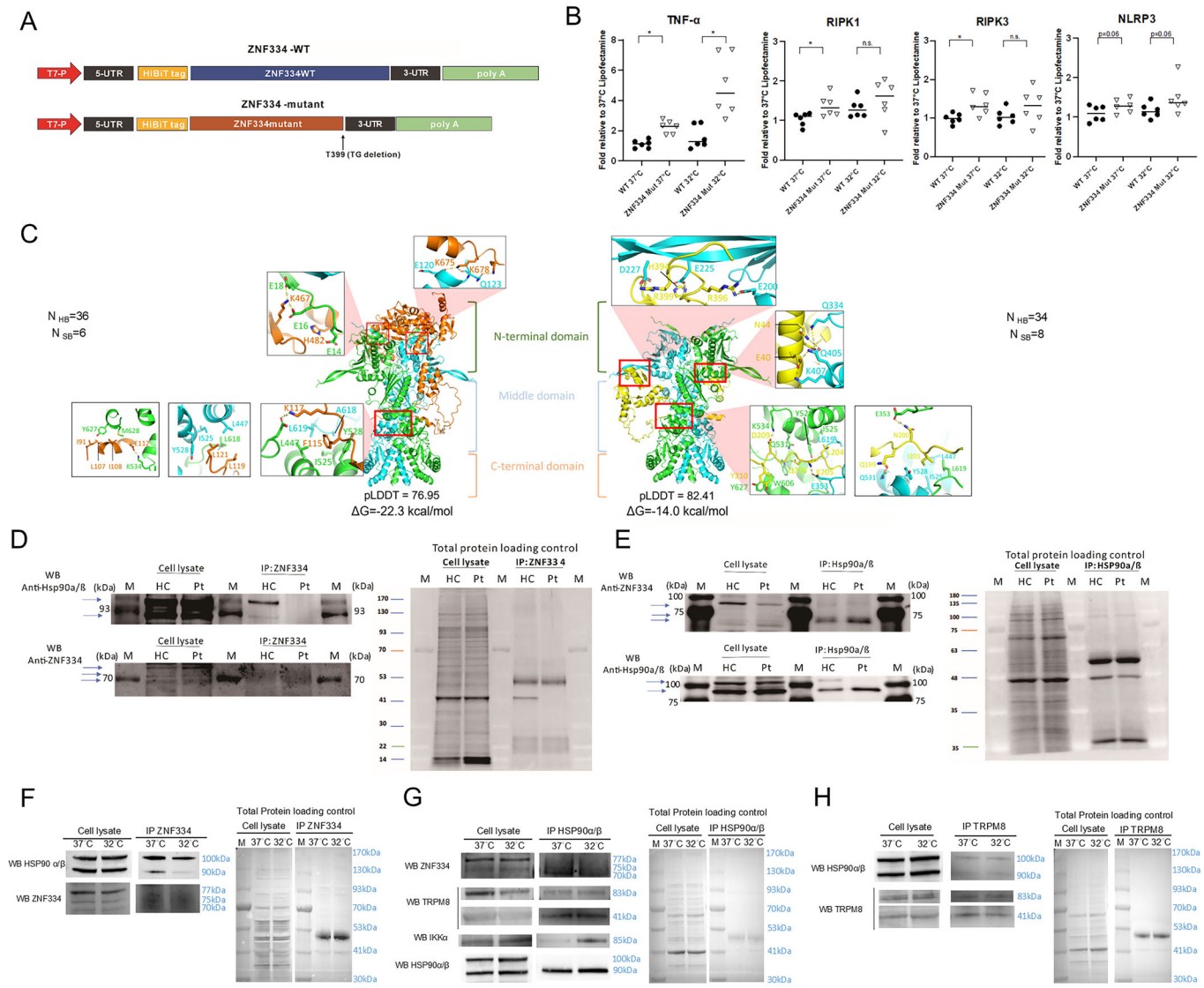

**Figure 6. ZNF334 p.Thr399fs mutant in primary monocytes enhanced cold-induced expression of inflammation-related genes, and reduced the binding affinity of ZNF334 to Hsp90.**

(A) structures of IVT-mRNA. (B) Relative mRNA expressions of *TNF, RIPK1, RIPK3,* and *NLRP3* in monocytes (derived from healthy controls) transfected with ZNF334-WT or ZNF334-mutant IVT-mRNA. Monocytes were further stimulated at 32 °C incubator for 6 h after 2 h of transfection. Relative mRNA expression was calculated as a fold increase relative to the mean expression level in monocytes treated with lipofectamine only at 37 °C. Lines represent medians. *$P < 0.05$ by two-tailed Wilcoxon signed-rank test. (C) Predicted binding affinity of ZNF334 wild-type (ZNF334$^{WT}$, orange) or ZNF334 mutant (ZNF334$^{mut}$, yellow) with Hsp90α homodimer. As shown in the figure, when forming a complex with Hsp90α homodimer, ZNF334$^{mut}$ has a smaller interface and less affinity than ZNF334$^{WT}$. The interface area of ZNF334$^{WT}$ and Hsp90α homodimer is predicted to be 3540.1 Å$^2$; the interface area of ZNF334$^{mut}$ and Hsp90α homodimer is predicted to be 2724.8 Å$^2$. Interface residues forming hydrophobic cores, hydrogen bonds and salt bridges are shown as sticks. Δ$^i$G, solvation free energy gain upon formation of the interface; Δ$^i$G *P* value, *P* value of the observed solvation free energy gain; N$_{HB}$, number of potential hydrogen bonds across the interface; N$_{SB}$, number of potential salt bridges across the interface. (D, E) Interaction of ZNF334 with Hsp90 in primary monocytes derived from PBMCs of healthy control (HC) or patient (Pt) carrying heterozygous ZNF334 p.Thr399fs mutation. (D) ZNF334 was immunoprecipitated in the whole-cell lysate of primary monocytes and blotted against anti-Hsp90α/ß, or anti-ZNF334 antibody. (E) Hsp90α/ß was immunoprecipitated in the whole-cell lysate of primary monocytes and blotted against anti-ZNF334, or anti-Hsp90α/ß antibody. M: marker; HC: pooled monocytes from two healthy controls; Pt: pooled monocytes from the patient collected at "cold + + + is + " and "cold+ is + " stages. Blue arrows at the left side of the blot of cell lysate indicate ZNF334 70 kDa, 75 kDa and 77 kDa isoforms; Hsp90α/β 90 kDa, 100 kDa isoforms. (F–H) ZNF334 (F), Hsp90α/β (G), or TRPM8 (H) was immunoprecipitated in the whole-cell lysate of monocytes derived from healthy controls before or after the 8-h cold stimulation at 32 °C and blotted against anti-Hsp90α/β, anti-ZNF334, or anti-TRPM8 antibody. A representative graph of three experiments is shown.

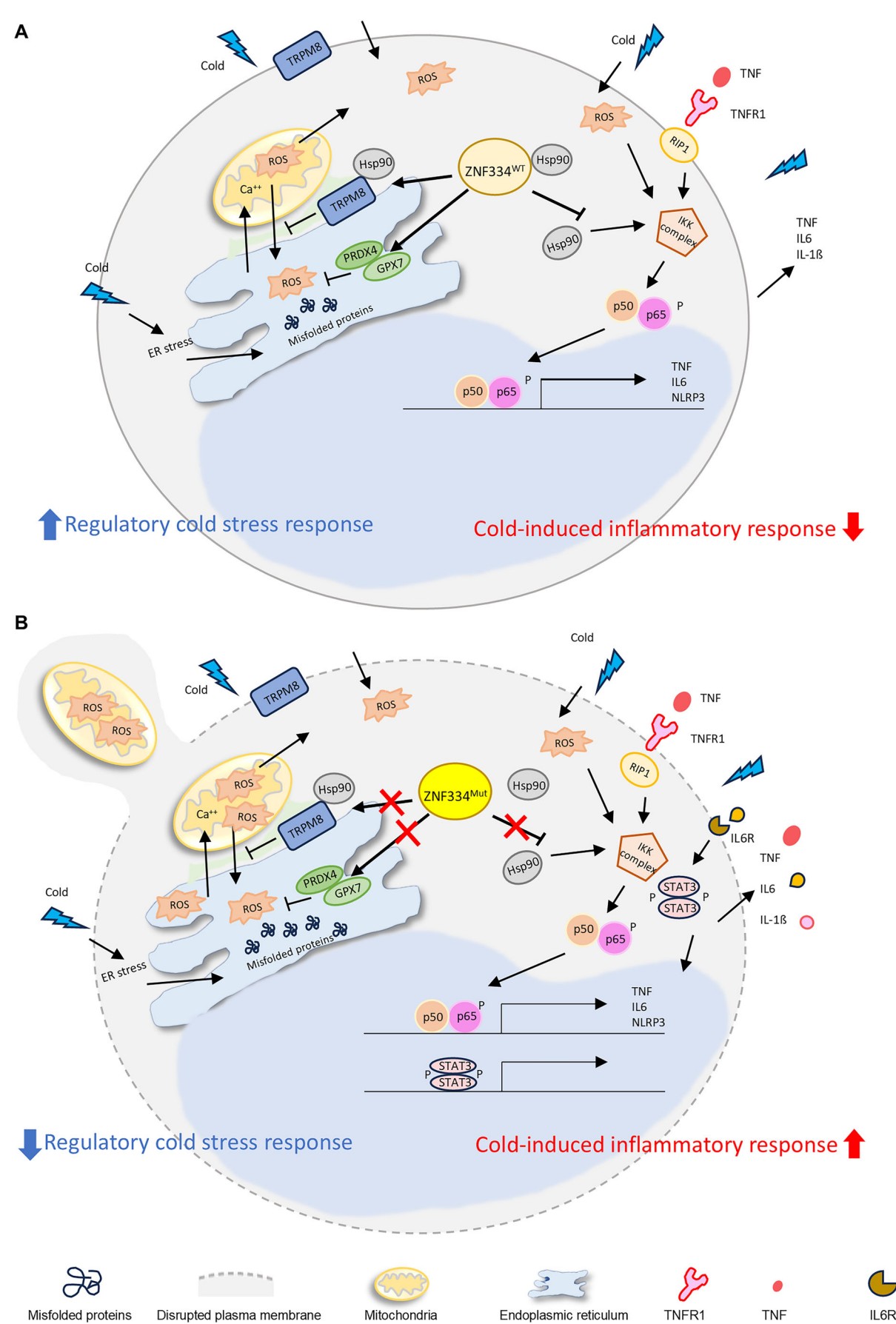

**Figure 7.   Our model of ZNF334 truncation mutation mediated cold-induced hyperinflammation and excessive oxidative stress in human monocytes, underlying the pathogenesis of ZACAS.**

(A) ZNF334 wild-type keeps the homeostasis in monocytes by maintaining the cell membrane integrity, maintaining the ER protein folding response necessary to regulate redox homeostasis, stabilizing the endogenous expression of the stress regulators Hsp90 and TRPM8, and plays a role in suppressing cold-induced NF-κB activation in monocytes by interacting with Hsp90, preventing the accessibility of IKK complex to the ATP binding site in the N-domain of Hsp90 upon cold stimulation. (B) ZNF334 truncation mutation disrupted the plasma membrane integrity, leading to exposure of endogenous danger signals and increased intracellular ROS. ZNF334 truncation mutation downregulated the transcription of genes involving ER protein folding response necessary to regulate redox homeostasis in response to stress, and reduced the baseline protein levels of Hsp90 and TRPM8, which are key molecular chaperones that regulate stress adaptation. The ZNF334 truncation mutation decreased the binding affinity of ZNF334 to Hsp90 protein dimers, which potentially increased the accessibility of the IKK protein complex to Hsp90 upon cold exposure, thereby augmenting cold stress-induced NF-κB activation. With the deficient ER protein folding and antioxidative activities, repeated cold exposure formed a positive feedback loop that resulted in excessive production of proinflammatory cytokines TNF-α, IL-6, and IL-1ß. The cold-induced excessive oxidative and proteotoxic stress also led to mitochondrial damage and increased the secretion of TRPM8+ microvesicles containing mitochondria and proinflammatory chemokines, further contributing to systemic autoinflammation.

the attenuation of cold-induced *IL-6* upregulation in ZNF334$^{+/-}$ monocytes (Fig. 8A–C). Since Janus kinase (JAK) inhibitors have been reported to reduce IL-6-induced inflammation (Yeleswaram et al, 2020), we further performed pretreatment of THP-1 monocytes with 1 μM ruxolitinib, a JAK1/2 inhibitor, which however, did not significantly attenuate cold-induced *TNF*, *NLRP3* and *IL6* (Fig. 8D). These results conclude that the Hsp90 inhibitor geldanamycin had effects on the attenuation of cold-induced hyperactivation of *TNF* and *NLRP3*, but not *IL-6*, in THP-1 monocytes with ZNF334 truncation mutation.

## Analyses of public genetic and RNA-seq datasets revealed an association between downregulated monocyte *ZNF334* expression and other rheumatic disease

In this study, we have examined the functional impact of monocyte *ZNF334* truncation mutation in a patient with an AID induced by repeated cold exposure, which we referred to as "ZACAS". We searched gnomAD v4.1, a large population-based genomics database that contains the genotypic data of more than 1.6 million alleles, for other *ZNF334* truncation mutations nearby. We noticed several *ZNF334* variants predicted to cause frameshift or stop-gain effects around the 300th to 400th amino acids of ZNF334, with low total population frequencies (Appendix Table S2). However, these variants were not reported in ClinVar, and data of Sanger sequencing validations, monocyte functional assays, and environmental cold exposures were unavailable, making it difficult to interpret their clinical significance. Next, we examined whether the ZNF334 loss-of-function expression signature identified through RNA-seq analysis is present in monocytes derived from patients with other chronic inflammatory diseases, such as juvenile idiopathic arthritis (JIA). Analysis of the RNA-seq data of monocytes derived from patients with polyarticular JIA (E-MTAB-14035) (Hounkpe et al, 2024) revealed the downregulated expression of *ZNF334* and the oxidative protein folding enzyme *PRDX4*, along with upregulated expression of genes related to monocyte activation and pro-inflammatory cytokines (Appendix Fig. S2).

## Deep sequencing revealed co-existence of a low-level NLRP3 mutant in PBMCs derived from the patient

In order to investigate whether other low-level acquired genetic variants related to autoinflammation contributed to the patient's late-onset phenotype, we further deep-sequenced DNAs extracted

from PBMCs of the patient and her *ZNF334*-wild type sister for *NLRP3, NLRP12, NLRC4, PLCG2, TNFRSF1A*, and *ZNF334*. Similar to the results of whole exome sequencing, ZNF334 p.Thr399fs with a variant allele frequency of 53.1% was detected in the patient, but not in the sister. Moreover, NLRP3 p.Thr347Ile with variant allele frequency of 14% (sequencing depth 16544) was also detected only in the patient; while a highly prevalent NLRP3 synonymous single-nucleotide polymorphism was detected in both samples with variant allele frequency of 99.8%.

To further evaluate the potential impact of the low-level NLRP3 p.Thr347IIe on inflammatory cytokine production in monocytes, we reanalyzed the CITE-seq single-cell data by gating on CD14+monocytes derived from PBMCs of the healthy sister (HC) and from PBMCs of the patient collected at two conditions: after a period of frequent cold exposure (S2) and after a period of cold avoidance (S3), both were under the use of immunosuppressant. A total of 592 monocytes were analyzed in each sample: the number of *TNF*+ monocytes were 7, 25, 17 in HC, S2, and S3, respectively; the number of *NLRP3*+ monocytes were 2, 12, 5 in HC, S2, and S3, respectively; the number of *IL-1B*+ monocytes were 9, 8, 9 in HC, S2, and S3, respectively. Taken together, different levels of somatic mutations (53.1% ZNF334 p.Thr399fs and 14% NLRP3 p.Thr347Ile) were detected in the patient's blood cells. The numbers of *TNF*+ monocytes and *NLRP3*+ monocytes were highest in S2, while the proportions of *IL-1B*+ monocytes were similar among the three samples.

## Discussion

In this paper, we report a heterozygous *ZNF334* frameshift mutation, p.Thr399fs, in a patient with a previously unreported AID. This patient presented with middle-aged-onset, non-itchy urticaria-like skin rash, fever, arthralgia, and lymphadenopathy induced by repeated exposures to temperatures of 25 °C to 27 °C at her workplace. She also presented with positive ice cube test results, acrocyanosis induced by local ice cube application, and progressive SNHL.

We demonstrated via RNA-seq analysis of *ZNF334*-edited THP-1 monocytes that ZNF334 truncation decreased chaperone-mediated protein folding signaling, disrupted the ER protein folding response necessary to regulate redox homeostasis in response to stress, and enhanced cold-induced NF-κB activation in these monocytes. At the protein level, ZNF334 truncation mutation reduced the baseline levels of Hsp90 and TRPM8, which

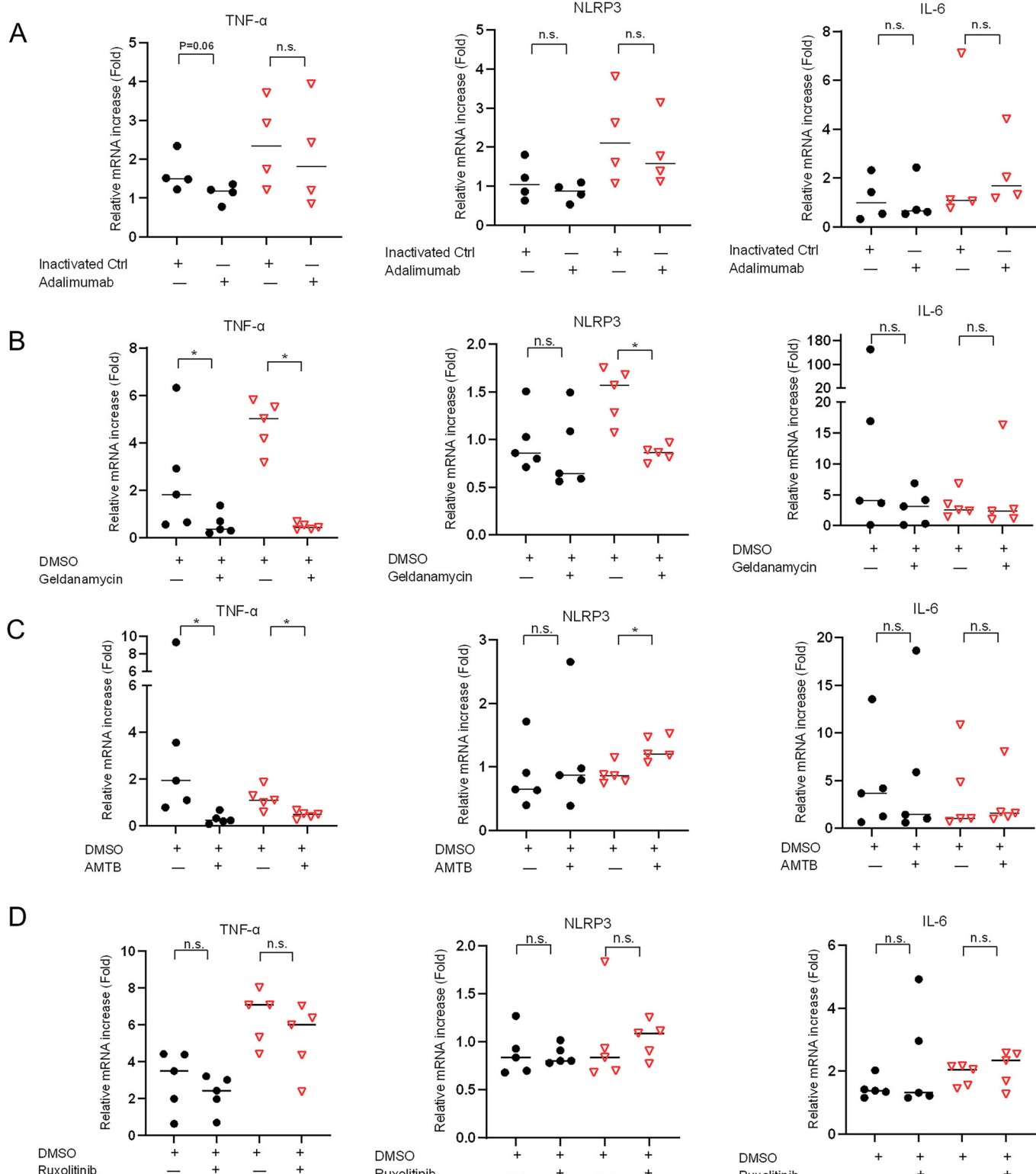

are key molecular chaperones that regulate stress adaptation and enable mild cold sensation and oxidative stress regulation, respectively. Through IP-western analysis, we found that the ZNF334 truncation mutation decreased the binding affinity of ZNF334 to Hsp90 protein dimers, which potentially increased the accessibility of the IKK protein complex to Hsp90, thereby augmenting cold stress-induced NF-κB activation. Our results further indicate that cold exposure formed a positive feedback loop that resulted in hyper-inflammation in ZNF334-mutated monocytes, which are already deficient in major stress regulators at

◀ **Figure 8.   Differential effects of inhibitor against TNF-α, Hsp90, or TRPM8 on the attenuation of cold-induced upregulation of *TNF, NLRP3,* and *IL-6* mRNA levels in ZNF334-WT and in ZNF334$^{+/-}$ THP-1 monocytes.**

(A) THP-1 monocytes (ZNF334 wild-type, $n = 4$; ZNF334 $+/-$, $n = 4$) were incubated with medium only, 10 µg/ml heat-inactivated adalimumab control (Ctrl), or 10 µg/ml adalimumab for 2 h, and then placed in the 37 °C or 32 °C incubator for another 2 h. Relative fold change of *TNF, NLRP3,* or *IL-6* mRNA expression was calculated as compared with the mean expression level in cells cultured at 37 °C with medium only. (B) THP-1 monocytes (ZNF334 wild-type, $n = 5$; ZNF334 $+/-$, $n = 5$) were pretreated with 2 µM geldanamycin or DMSO only for 1 h, and then placed in the 37 °C or 32 °C incubator for another 24 h. Relative fold change of *TNF, NLRP3,* or *IL-6* mRNA expression was calculated as compared with the mean expression level in cells cultured at 37 °C with DMSO only. (C) THP-1 monocytes (ZNF334 wild-type, $n = 5$; ZNF334 $+/-$, $n = 5$) were pretreated with 10 µM AMTB or DMSO only for 24 h, and then placed in the 37 °C or 32 °C incubator for another 24 h. Relative fold change of *TNF, NLRP3,* or *IL-6* mRNA expression was calculated as compared with the mean expression level in cells cultured at 37 °C with DMSO only. (D) THP-1 monocytes (ZNF334 wild-type, $n = 5$; ZNF334 $+/-$, $n = 5$) were pretreated with 1 µM ruxolitinib or DMSO only for 1 h, and then placed in the 37 °C or 32 °C incubator for another 24 h. Relative fold change of *TNF, NLRP3* or *IL-6* mRNA expression was calculated as compared with the mean expression level in cells cultured at 37 °C with DMSO only. Lines represent medians, n.s.: not significant; *$P < 0.05$ by one-tailed Wilcoxon signed-rank test. Black dots: ZNF334 wild-type THP-1; red triangles: ZNF334$^{+/-}$ THP-1 monocytes.

baseline. Confocal microscopy and EV isolation experiments indicated that excessive cold-induced oxidative and proteotoxic stress led to mitochondrial damage and increased the secretion of TRPM8+ microvesicles containing mitochondria and proinflammatory chemokines. Therefore, we proposed a model in which repeated cold exposure increases the levels of inflammatory cytokines and promotes the activation of ZNF334-mutated monocytes (Fig. 7). These monocytes release TNF- and chemokine-containing microvesicles that attract additional proinflammatory monocytes and lymphocytes to perivascular areas in cold-stimulated skin, as observed in our patient's lesional skin biopsy. This model suggests the use of an Hsp90 ATPase inhibitor as a potential treatment option that can be used in combination with cold avoidance to alleviate cold-induced autoinflammation in patients with "ZACAS".

Zinc finger proteins are involved in various molecular functions through their diverse zinc finger domains, which enable interactions with DNA, RNA, poly-ADP-ribose, and other proteins (Cassandri et al, 2017). Previous studies described reduced *ZNF334* transcription levels in cancer tissues and in CD4 + T cells of rheumatoid arthritis (Cheng et al, 2022; Li et al, 2024; Soroczynska-Cybula et al, 2011). In this study, we characterized the undescribed functions of ZNF334 in the regulation of cold stress, via zinc finger domain interactions with essential regulators of cellular homeostasis, and suppression of cold-induced nuclear translocation of p-p65 and p-STAT3 in monocytes. We named the newly identified cold-induced AID caused by ZNF334 truncation mutation "ZACAS" due to its distinct cellular impact on cold stress regulators Hsp90 and TRPM8, and the distinct cold exposure-induced autoinflammatory symptoms which overlaps with different categories of AIDs, reflecting the important role of ZNF334 acting as the hub of cold stress regulation, upstream of signaling pathways related to cold-induced inflammation, oxidative stress and cell death. For example, the patient's cold-induced urticaria-like rash, fever, arthralgia, and progressive SNHL, accompanied by enhanced monocyte *NLRP3* signaling and IL-1β production, corresponded with the features of disorders caused by autosomal dominant *NRLP3* gain-of-function mutations (Moltrasio et al, 2022). Periodic fever and lymphadenopathy—in addition to the increased monocyte expression of *RIPK1*, *RIPK3*, *IL6*, and *TNF-α* in the inflammatory response; elevated levels of p-STAT3; and increased sensitivity to apoptosis and necroptosis—overlapped with the features of disorders caused by the non-cleavable variants of *RIPK1* (Tao et al, 2020). In addition to these features, our patient exhibited ice bag–induced acrocyanosis

and circumoral cyanosis, which suggests vasospastic dysregulation upon exposure to cold. The unique mutation in *ZNF334*, the broad clinical and immunological features, and the strong correlation with cold exposures make ZACAS distinct from those AIDs listed in the 2022 International Union of Immunological Societies' classification of human inborn errors of immunity (Bousfiha et al, 2022; Tangye et al, 2022), suggesting that *ZNF334* truncation mutation results in a novel entity of AID.

We demonstrated that the *ZNF334* truncation mutation not only reduced the intracellular levels of stress regulators Hsp90α/β and TRPM8 in monocytes, but also weakened the binding affinity of ZNF334 with Hsp90 dimer complex, rendering monocytes more susceptible to cold stress-induced imbalance in cellular homeostasis. Intracellular Hsp90 is an essential molecular chaperone and key component of multiprotein complexes; it enhances protein stability and establishes a protein-folding reservoir in response to environmental stress (Bhattacharya et al, 2022; Somogyvari et al, 2022). Hsp90 can regulate cell death and inflammation through its interaction with client proteins, such as receptor interacting protein kinase 1 (RIPK1), RIPK3, NLRP3, and STAT3, which are key molecules involving TNF receptor 1 mediated NF-κB activation, necroptosis, NLRP3 inflammasome activation, or IL-6 signaling (Lewis et al, 2000; Mayor et al, 2007; Piippo et al, 2018; Prinsloo et al, 2012; Sato et al, 2003; Yang and He, 2016). In addition, Hsp90 can further modulate the NF-κB pathway by forming a heterocomplex with another signaling effector, the IKK complex (Chen et al, 2002). We found that in monocytes derived from healthy controls, Hsp90 interacts with ZNF334, TRPM8, and IKKα at 37 °C. Interestingly, the interaction between Hsp90 and ZNF334 decreased, while the interaction between Hsp90 and IKKα increased upon 32 °C cold stimulation. It is likely that the weakened ZNF334 and Hsp90 interaction contributes to increased accessibility of Hsp90 to key effectors in inflammation, triggering downstream NF-κB and NLRP3 activations. Indeed, we observed enhanced *TNF/RIPK1/RIPK3* signaling, together with elevated IL-1ß secretion in ZNF334 wild-type THP-1 monocytes after hours of cold exposure, although the degree of increase in the expression levels of these proinflammatory molecules were much less than those observed in monocytes harboring ZNF334 truncation mutation. In ZNF334 truncated monocytes, the interaction of ZNF334 and Hsp90 already decreased at baseline, corresponding to the elevated basal levels of *TNF/RIPK1/RIPK3/NLRP3/STAT3* detected in ZNF334$^{+/-}$ monocytes. In our IP-western analyses, the binding of mutant ZNF334 to Hsp90 was almost absent in ZNF334 pull-down assays but present in Hsp90α/β antibody-immunoprecipitated samples. These findings

may be attributable to the predicted weakening, rather than complete loss, in the interaction between mutant ZNF334 and Hsp90 and the potential masking of reduced binding affinity during ZNF334 blotting in immunoprecipitates of Hsp90, which is an abundant cellular protein. Upon cold exposure, the interaction between ZNF334 and Hsp90 was further weakened in the ZNF334$^{+/-}$ monocytes. Unlike wild-type ZNF334, the released ZNF334 truncation mutant was dysfunctional and did not facilitate regulatory stress responses, such as ER protein folding or antioxidation, leading to excessive ER and oxidative stress and triggering the release of inflammatory cytokines. In our patient, of note, the heterozygous *ZNF334* p. Thr399fs mutation was associated with a low level of intracellular Hsp90 in monocytes, but with markedly elevated levels of eHsp90 in the plasma. This suggests that the impaired interaction between ZNF334 truncation-mutant and Hsp90 might also affect the secretion of Hsp90α, which is known to be proinflammatory (Bohonowych et al, 2014; Ding et al, 2022). In addition, pretreatment of monocytes harboring ZNF334 truncation mutation with geldanamycin, an Hsp90 inhibitor reported to block both the intracellular chaperone function of Hsp90 and the secretion of eHSP90 (Gooljarsingh et al, 2006; Li et al, 2013), markedly attenuated the cold-induced hyperactivation of *TNF* and *NLRP3*. However, pretreatment with geldanamycin did not significantly reduce the cold-induced hyperproduction of IL-6 in ZNF334$^{+/-}$ THP-1 cells. Since we are not sure whether the adverse reaction to anti-IL6R antibody treatment observed in our patient with the *ZNF334* truncation mutation was due to allergic reaction or the decrease in serum IL-6 level, our findings that geldanamycin could reduce the cold-induced upregulation of *TNF, NLRP3*, but not *IL-6* in ZNF334$^{+/-}$ cells suggest that the Hsp90 inhibitor could be a potential treatment option for the patient.

In addition to Hsp90, transient receptor potential (TRP) channels maintain cellular homeostasis under environmental stress conditions. Dysregulation of TRP channels is associated with impaired vasorelaxation and hyperinflammatory responses (Nilius, 2007). Among the TRP channels, TRPM8 is the established cold sensor that is expressed on the plasma membrane of not only sensory neurons but also monocytes/macrophages (Hornsby et al, 2022a; Khalil et al, 2016; Trusiano et al, 2022). In mammals, TRPM8 senses cold at temperatures below 33 °C (Plaza-Cayon et al, 2022). Besides serving as a thermosensor, it has been reported that endogenous TRPM8 activity in monocytes regulates monocyte/macrophage differentiation and modulates the balance between TNF-α and IL-10 levels (Hornsby et al, 2022a; Khalil et al, 2016; Trusiano et al, 2022). However, these previous reports on monocytes/macrophages mainly discussed about the function of TRPM8 in regulating inflammation in the context of colitis model or LPS stimulation, which might be different from the context of cold stress. In fact, short isoforms of TRPM8 which localized to the ER have been identified in several cell types frequently contacting environmental coldness, such as keratinocytes and airway epithelial cells (Bidaux et al, 2015; Sabnis et al, 2008b). The short isoform of TRPM8 was found to regulate cold-induced ER Ca$^{2+}$ release, and activation of the TRPM8 variant increased the production of inflammatory cytokines, e.g., IL-1ß, IL-6, TNF-α in bronchial epithelial cells after cold treatment (Sabnis et al, 2008a). Furthermore, the 39–40 kDa TRPM8 isoform was reported to

control mitochondrial Ca$^{2+}$ concentration, which facilitates the synthesis and accumulation of superoxide when activated under mild-cold conditions in keratinocytes (Bidaux et al, 2015). In our study, we showed that in human monocytes, TPRM8 is expressed at the plasma membrane, ER, and even cold-stimulated EVs. We have also detected a short TRPM8 isoform of 41 kDa in monocytes, and discovered significantly decreased baseline intracellular TRPM8 expression, together with markedly increased cold-induced ROS production and secretion of TRPM8+mitochondria+ EVs in monocytes harboring the truncated ZNF334, indicating that the ZNF334 mutation might affect the extracellular translocation of TRPM8 and alter the TRPM8 function in regulating the ER-mitochondria Ca$^{2+}$/ROS homeostasis upon cold stress. Moreover, we detected interactions between Hsp90 and the 83- and 41-kDa isoforms of TRPM8 in monocytes at baseline and after cold stimulation at 32 °C. However, whether the stability and activation of TRPM8 isoforms require assistance from Hsp90 or the ZNF334-Hsp90 complex needs further investigation. Of note, pretreatment of ZNF334$^{+/-}$ THP-1 monocytes with a TRPM8 inhibitor, AMTB, could attenuate the cold-induced hyperproduction of *TNF*, but not the upregulations of *NLRP3* and *IL-6*. Unlike the effects observed in Hsp90 inhibitor treatment, AMTB increased the RNA levels of *NLRP3* in cold-stimulated ZNF334 + /− THP1 monocytes, suggesting different roles of TRPM8 and Hsp90 in regulating cold-induced inflammatory effector signaling, and ZNF334 could be the hub responsible for fine-tuning the expression and function of TRPM8 and Hsp90 during cold stress.

Together, our data of monocytes support a model deciphering the role of wild-type ZNF334 in keeping the homeostasis in monocytes by maintaining the cell membrane integrity, stabilizing the endogenous expression of stress regulators Hsp90 and TRPM8, and negatively regulate the nuclear translocations of p-p65 and p-STAT3 upon cold stimulation (Fig. 7A). Truncation mutation of ZNF334 disrupts the homeostatic function of ZNF334, and decreases the binding affinity between ZNF334 and Hsp90, resulting in priming of monocytes for enhanced cold stress inflammatory response (Fig. 7B). It is reported that the activated monocytes could secrete microvesicles containing mitochondria, which mediate inflammation in endothelial cells (Puhm et al, 2019). To the best of our knowledge, this is the first study to report the release of TRPM8+ microvesicles in human monocytes, which corresponds to cold stimulation. The increased secretion of TRPM8+mitochondria+EVs detected in monocytes with ZNF334 truncation mutation might also exert intercellular proinflammatory signals, further amplifying systemic autoinflammation, chemotaxis of activated monocytes and lymphocytes to cold-stimulated perivascular areas, and contributing to the cold-induced fever, urticaria-like skin rashes, and excessive vasospastic symptoms seen in our patient.

During the course of the disease, our patient developed progressive SNHL. It has been reported that both cochlear infiltrating macrophages and microglial cells, which are central nervous system-resident macrophages, play crucial roles in maintaining cochlear homeostasis, and their dysregulation contributes to the pathogenesis of inflammatory inner ear disease and SNHL (Nakanishi et al, 2017; O'Malley et al, 2015). Moreover, a reduced Hsp90 level has been found to be associated with ototoxic drug-induced cochlear hair cell death (Lai et al, 2018). In this study, the observation of reduced baseline expression of Hsp90 in monocytes might be detected as well in cells in the nervous system with germline *ZNF334* truncation mutation, contributing to susceptibility to

cell death upon stress. Our finding of increased cold-induced inflammation in the patient's monocyte also suggests the possibility of cochlear infiltration of hyper-inflammatory macrophages, which further aggravates the disturbed cochlear homeostasis upon repeated exposure to cold stress, leading to progressive SNHL.

Our findings suggest that ZNF334 p.Thr399fs truncation mutation arises from an early postzygotic event as a somatic mosaicism that affects both myeloid and lymphoid hematopoietic cells and demonstrates partial skin distribution. These findings are consistent with our patient's late-onset phenotype and the requirement for repeated cold stimulation to amplify the dominant proinflammatory responses necessary to activate and expand mutant-carrying immune cells. Multiple studies have indicated that somatic mutations in immune cells could contribute to AIDs, including CAPS (caused by *NLRP3* variants) and TNF receptor-associated periodic syndrome (caused by *TNFRSF1A* variants) (Cooper, 2025; Rowczenio et al, 2017). However, the pathogenicity of low-level somatic mutations requires rigorous functional studies and extensive genetic investigations of co-existing germline or somatic variants that might be the actual driver mutation or exerting synergistic effects (Schmitz et al, 2025). Our model of the pathogenic role of ZNF334 p.Thr399fs truncation mutation in ZACAS was strengthened by the hyperinflammatory cellular phenotype detected in primary monocytes transiently transfected with in vitro-transcribed mRNA overexpressing this exact mutation. Of note, our whole exome sequencing analysis with the available mosaic pipeline did not reveal other autoinflammation-related somatic variants in our patient; nevertheless, further amplicon-based deep sequencing covering exons of *NLRP3, NLRP12, NLRC4, PLCG2, TNFRSF1A*, and *ZNF334* detected the co-existence of a low-level (14%) NLRP3 p.Thr347Ile variant in the patient's PBMCs. Analysis of our CITE-seq single-cell data revealed that as compared with healthy ZNF334-wild type control, the number of *TNF*+ monocytes was 3.57-fold higher in the patient's PBMCs collected after a period of frequent cold exposure, and was 2.43-fold higher in those collected at cold avoidance; while the numbers of *IL-1B*-expressing monocytes were similar among the three samples. Similarly, our analyses of plasma proinflammatory cytokine levels showed that the patient's IL-1β levels were markedly lowered during cold avoidance as compared with those during repeated cold exposure without the use of immunosuppressants; while the decrease in the levels of TNF-α and IL-6 by cold avoidance was less prominent. These data suggest that unlike the reported phenotype of cells carrying NLRP3 p.Thr347Ile that showed constitutive NLRP3 inflammasome activation in in vitro assay (Feng et al, 2025), the majority of our patient's monocytes did not secrete excessive IL-1β. Furthermore, in the same paper analyzing the functional impacts of NLRP3 variants from the INFEVERS registry and the ClinVar database, NLRP3 p.Thr347Ile was not listed in the category of disease-associated variants conferring cold exposure-triggered NLRP3 hyperactivation (Feng et al, 2025). The percentage of NLRP3 p.Thr347Ile variant detected in the patient was not reported in the reference publication listed in the INFEVERS registry (Suri et al, 2021). Moreover, unlike our patient, all the CAPS/MWS/NOMID cases described in this reference had a very young age-of-onset, with severe consequences or accompanied with amyloidosis, and none of these patients had the record of cold exposure-induced symptoms (Suri et al, 2021). Therefore, it is less likely that the low-level NLRP3 somatic variant alone contributed to the systemic AID seen in

this patient. Instead, our data derived from cold stimulation assays using CRISPR/Cas9-edited ZNF334-mutant THP-1 monocytes and in vitro-transcribed mRNA overexpressing ZNF334 p.Thr399fs suggest that the heterozygous ZNF334 p.Thr399fs variant in monocytes is the main driver contributing to the adult-onset cold-induced AID in our case. Interestingly, while we observed that ZNF334 p.Thr399fs decreased baseline Hsp90 levels in monocytes, it has been reported that NLRP3 gain-of-function variant-mediated NLRP3 autoactivations were hampered by Hsp90 deficiency (Spel et al, 2024). Whether the co-existing low-level NLRP3 p.Thr347Ile has a minor impact on enhancing the autoinflammatory phenotype needs further investigation.

This study has several limitations. First, we were unable to "knock-in" our patient's ZNF334 frameshift mutation in THP-1 monocytes by using CRISPR/Cas9 gene editing technology without off-target effects. Therefore, we were only able to comprehensively examine molecular changes in ZNF334$^{+/-}$ and ZNF334$^{-/-}$ truncation "knockout" THP-1 monocytes. Second, in our study, we reported on this rare disease in a single patient. Although other *ZNF334* truncation mutations are reported in the gnomAD database, the genetic variants detected in ZNF proteins have been reported to be prone to inaccuracies in short-read sequencing because of potential alignment errors in sequences encoding repeated ZnF domains (Field et al, 2019). For example, significantly different frequencies are observed between the whole-exome and whole-genome data of ZNF334 p.Arg284Ter mutation in gnomAD v4.1, suggesting potential artifacts (Atkinson et al, 2023). Furthermore, the general population contributing to the total allele frequencies calculated in gnomAD does not exclude those with rheumatic diseases. Therefore, interpreting the association between *ZNF334* truncation mutation and other disease phenotypes is challenging without further Sanger and functional validation studies. Given that abrupt reductions in temperature have been reported to trigger arthralgia in patients with JIA (Tsai et al, 2006), we analyzed the gene expression signature of *ZNF334* in an RNA-seq dataset (E-MTAB-14035) derived from classical monocytes of patients with JIA and controls. We identified the potential involvement of downregulated *ZNF334* expression and decreased oxidative protein folding signaling in chronic monocyte inflammation in JIA. To further investigate the impacts of different dosages of *ZNF334* truncation mutation in various immune cell types on systemic cold-induced or cold-aggravated autoinflammatory diseases, and to better evaluate the in vivo treatment responses of inhibitors, the development of a ZNF334 gene-edited mouse model is needed.

In conclusion, we demonstrated the essential and intricate roles of ZNF334 in regulating the inflammatory response of monocytes to cold stress in humans, and delineated the pathogenesis of the repeated mild cold exposure-induced novel autoinflammatory disease, ZACAS. We characterized the impact of *ZNF334* truncation mutation on the expression and function of the main stress regulators Hsp90 and TRPM8, and the secretion of previously unidentified cold-induced TRPM8+mitochondria+ microvesicles. Our results might also shed light in the pathogenesis of other cold-induced/aggravated diseases of unclear etiologies, such as JIA (Tsai et al, 2006), acquired cold urticaria (Siebenhaar et al, 2007), and susceptibility to cold weather-induced acute myocardial infarction (Vaiciulis et al, 2021). Furthermore, given the reported associations of ZNF334, Hsp90 and TRPM8 with tumor progressions (Cheng

et al, 2022; Condelli et al, 2019; Jafari et al, 2020; Liu et al, 2016; Yang et al, 2023), and the potential of cancer hyperthermia therapies (Dewhirst et al, 2016; Kalamida et al, 2015), this study also provides hints in thermal modulation of tumor stress responses.

# Methods

### Reagents and tools table

| Reagent/resource | Reference or source | Identifier or catalog number |
|---|---|---|
| **Experimental models** | | |
| THP-1 cells (*H. sapiens*) | ATCC | TIB-202 |
| ZNF334$^{+/-}$ THP-1 cells (*H. sapiens*) | This study | C10 |
| ZNF334$^{-/-}$ THP-1 cells (*H. sapiens*) | This study | R14 |
| **Antibodies** | | |
| ZNF334 | Invitrogen | PA5-31843 |
| HSP90α/β | Santa Cruz | SC-13119 |
| TRPM8 | ECM | TM5711 |
| TRPM8 | Invitrogen | PA1-46239 |
| STAT3 | Invitrogen | MA1-13042 |
| Phospho-NF-κB p65 (Ser 536) | Cell Signaling | 3033 |
| Phospho-NF-κB p65 (Ser 536) | Santa Cruz | SC-136548 |
| Phospho-STAT3 (Tyr705) | Cell Signaling | 9145 |
| Calreticulin | Invitrogen | MA5-15382 |
| IKKα | Cell Signaling | 11930 |
| Anti-rabbit Alexa Fluor 488 | Invitrogen | A-11008 |
| Anti-mouse Alexa Fluor 594 | Invitrogen | A-11005 |
| Anti-rabbit IgG (H + L) HRP conjugated | Invitrogen | 31460 |
| Anti-mouse IgG (H + L) HRP conjugated | Invitrogen | 31430 |
| PE mouse anti-human CD3 | BD Biosciences | 345765 |
| PE-Cy7 mouse anti-human CD14 | BD Biosciences | 557742 |
| **Oligonucleotides and other sequence-based reagents** | | |
| qPCR primers | This study | Appendix Table S3 |
| CITE-seq antibody panel | BD Biosciences | Appendix Table S4 |
| Amplicon-based deep sequencing primers | This study | Appendix Table S5 |
| **Chemicals, enzymes, and other reagents** | | |
| Ficoll-Paque PLUS | GE Healthcare Biosciences | 17144002 |
| DNeasy Blood & Tissue Kit | Qiagen | 69504 |

| Reagent/resource | Reference or source | Identifier or catalog number |
|---|---|---|
| Human TNF-α Quantikine HS ELISA Kit | R&D Systems | HSTA00E |
| Human IL-6 Quantikine HS ELISA Kit | R&D Systems | HS600C |
| Human HSP-90 ELISA Kit | Elabscience | E-EL-H1864 |
| Bio-Plex Pro Human Cytokine 27-plex Assay | Bio-Rad | M500KCAF0Y |
| Calcein AM | Thermo Fisher Scientific | C1430 |
| Draq7 | BD Bioscience | 564904 |
| Human Single-Cell Multiplexing Kit | BD Bioscience | 633781 |
| BD Rhapsody Cartridge Reagent Kit | BD Bioscience | 633733 |
| BD Rhapsody cDNA Kit | BD Bioscience | 633773 |
| BD Rhapsody Whole Transcriptome Analysis Amplification Kit | BD Bioscience | 633802 |
| RPMI | Thermo Fisher Scientific | 11875093 |
| RPMI, no phenol red | Thermo Fisher Scientific | 11835030 |
| Fetal Bovine Serum | Thermo Fisher Scientific | A5256701 |
| Penicillin–Streptomycin | Thermo Fisher Scientific | 15140122 |
| geldanamycin | Enzo | BML-EI280 |
| AMTB | Sigma-Aldrich | SML0103 |
| DMSO | Sigma-Aldrich | D2650 |
| Opti-MEM Reduced Serum Medium | Thermo Fisher Scientific | 31985062 |
| T7 Endonuclease I | New England Biolabs | M0302 |
| HiScribe T7 mRNA Kit with CleanCap Reagent AG | New England Biolabs | E2080 |
| Lipofectamine MessengerMAX | Thermo Fisher Scientific | LMRNA008 |
| TRIzol | Cyrusbioscience | 2001 |
| PrimeScript RT Reagent Kit | Takara Bio USA | RR037B |
| KAPA SYBR FAST Universal Kit | Roche | KK4600 |
| IQ2 TagMan Probe qPCR Master Mix | Bio-Genesis | BB-DBU-010 |
| DCFDA / H2DCFDA - Cellular ROS Assay Kit | Abcam | ab113851 |
| Hoechst 33258 | Sigma-Aldrich | 23491-45-4 |
| Annexin V | BD Bioscience | 51-65874X |
| Propidium Iodide | BD Bioscience | 51-66211E |
| CellVue Claret Far Red Fluorescent Cell Linker Kit | Sigma-Aldrich | CB_000730 |
| MitoTracker Deep Red FM | Thermo Fisher Scientific | M22426 |
| BSA Fraction V IgG Free | Thermo Fisher Scientific | 30063481 |
| SuperSignal West Pico PLUS Chemiluminescent substrate | Thermo Fisher Scientific | 34580 |

| Reagent/resource | Reference or source | Identifier or catalog number |
|---|---|---|
| No-Stain Protein Labeling Reagent | Thermo Fisher Scientific | A44449 |
| Universal Magnetic Co-IP kit | Active Motif | 54002 |
| Ruxolitinib | MedChemExpress | HY-50856 |
| qEV | Izon Science | IC1-70 |
| DNA Sample Preparation Kit | Roche Diagnostics | 05985536190 |
| 2X KAPA2G Fast Multiplex Mix | KAPA | KK5801 |
| KAPA EvoPrep Kits | Roche | 10154039001 |
| iSeq 100 i1 Reagent | illumina | 20021533 |
| **Software** | | |
| Bcl2fastq Conversion Software | https://support.illumina.com/sequencing/sequencing_software/bcl2fastq-conversion-software.html | N/A |
| BD Rhapsody Sequence Analysis Pipeline v1.9.1 | https://velsera.com/bd-rhapsody/ | N/A |
| SeqGeq v1.7 software | https://www.flowjo.com/seqgeq/download | N/A |
| STRING v.11.5, v.12.0 | https://string-db.org/ | N/A |
| MetaXpress High-Content Image Acquisition and Analysis Software | Molecular Devices | MetaXpress |
| ImageJ Fiji v.1.53.t | https://fiji.sc/ | N/A |
| iBright Analysis software v5.1.0 | Thermo Fisher Scientific | iBright |
| GraphPad Prism software, v10.2.3 | https://www.graphpad.com/ | N/A |
| RStudio v2025.05.1 + 513 | https://dailies.rstudio.com/version/2025.05.1+513/ | N/A |
| Trimmomatic v0.39 | Bolger et al, 2014 | N/A |
| Cutadapt v4.9 | Martin, 2011 | N/A |
| Genome Analysis Toolkit Mutect2 caller v4.2 | https://gatk.broadinstitute.org/hc/en-us | N/A |
| BCFtools v1.21 | Danecek et al, 2021 | N/A |
| **Other** | | |
| ABI 3100 Genetic Analyzer | Applied Biosystems | ABI 3100 |
| BD Rhapsody Single-Cell Analysis System | BD Bioscience | BD Rhapsody |
| NovaSeq 6000 Sequencing System | Illumina | NovaSeq 6000 |
| NEPA21 electroporator | NepaGene | NEPA21 |
| BD FACSCanto II Flow Cytometer | BD Bioscience | FACSCanto II |
| Tunable Resistive Pulse Sensing (TRPS) Nanoparticle Analyzer | Izon Science | Exoid |
| Qsep100 | Bioptic | C100100 |

| Reagent/resource | Reference or source | Identifier or catalog number |
|---|---|---|
| Qubit Fluorometer | Thermo Fisher Scientific | Q33226 |
| iSeq100 sequencing system | illumina | 20021532 |

## Subjects

This study included the patient, her healthy sibling (who had wild-type *ZNF334*), and unrelated healthy controls without cold-induced disorders, other rheumatic diseases, or malignancies.

## PBMC and monocyte isolation

Venous whole blood samples were collected from the patient and healthy controls; from these samples, PBMCs were isolated through density gradient centrifugation performed using Ficoll-Paque PLUS (GE Healthcare Biosciences, IL, USA, Cat#17144002) and cryopreserved in a liquid nitrogen tank for subsequent experiments, including whole exome sequencing and CITE-seq. For RNA extraction and western blotting, monocytes were further isolated from PBMCs by using the overnight adhesion method (monocyte-enriched to 80–90%).

## Whole-exome sequencing

WES was performed on genomic DNA extracted from whole blood from the patient and healthy siblings. Briefly, genomic DNA was extracted from the PBMCs by using the DNeasy Blood & Tissue Kit (Qiagen, Manchester, UK). DNA libraries were prepared using the Agilent SureSelect Human All Exon Kit (Agilent Technologies, CA, USA) and sequenced on the Illumina HiSeq platform by using 2× 100-bp paired reads. Bcl2fastq Conversion Software was used to demultiplex data and convert BCL files into the FASTQ file format. Alignment and variant calling were based on the hg19 reference using Burrows–Wheeler Aligner (Li and Durbin, 2009). Data were analyzed as described previously (Yang et al, 2018). The following criteria were used to select potential candidates: a mutant allele frequency of ≥30%, a global minor allele frequency of <1%, and pathogenicity predicted by at least one of three software programs (SIFT, PolyPhen, and CADD_PHRED).

## Sanger sequencing

To validate candidate variants and detect the ZNF334 genotype in the PBMCs derived from the family members of the patient, Sanger sequencing was performed as described previously (Chang et al, 2020). The *ZNF334* reference sequence used for primer design and nucleotide numbering was NM_ 001270497. PCR products amplified by specific primers (whose sequences are listed in Appendix Table S3) were sequenced on an ABI 3100 Genetic Analyzer (Applied Biosystems) and analyzed using the DNAStat Lasergene software. The patient's PBMCs were further sorted by BD FACSMelody Cell Sorter using antibodies against CD3 (SK7, BD Bioscience Cat#345765) and CD14 (M5E2, BD Bioscience Cat#557742) for DNA extraction and Sanger sequencing. DNAs of the FFPE blocks of skin biopsy samples derived from the patient

were also extracted using the DNA Sample Preparation Kit (Roche Diagnostics) for Sanger sequencing. Sequencing of T cells, monocytes, and skin biopsy samples were performed on SeqStudio Genetic Analyzer (Applied Biosystems).

## Serum cytokine analysis

Sera collected from the patient and her sister at different time points were stored at $-80\,°C$, and the levels of TNF-α (R&D Systems, Cat#HSTA00E), IL-1β (R&D Systems, Cat#HSLB00D), IL-6 (R&D Systems, Cat#HS600L), and eHsp90 (Elabscience, Cat#E-EL-H1864) were measured using ELISA kits in accordance with the manufacturer's instructions.

## CITE-seq

Oligonucleotide-conjugated antibodies (Abseq antibodies, all purchased from BD Biosciences; Appendix Table S4) and BD Rhapsody Single-Cell Analysis System (BD Biosciences) were used to integrate cell surface protein immunophenotyping and transcriptome measurements into a single-cell readout, as reported previously (Geng et al, 2021). In brief, PBMCs from the patient (and her healthy sister) were collected during an episode of cold-induced urticaria and arthralgia and during a period without cold-induced inflammatory symptoms. Each sample was first stained with Calcein AM (Thermo Fisher Scientific, Cat#C1430) and Draq7 (BD Bioscience, Cat#564904) to determine cell viability. Viable cells in each sample were labeled with different sample tags by using the Human Single-Cell Multiplexing Kit (BD Bioscience, Cat#633781), stained with Abseq antibodies, and pooled together to load into a primed BD Rhapsody Cartridge following the manufacturer's protocol. Then, the cells were lysed in the microwell cartridge to hybridize mRNA molecules to barcoded capture oligos on beads, which were subsequently retrieved by using the BD Rhapsody Cartridge Reagent Kit (BD Biosciences, Cat#633733). Microbead-captured single-cell transcriptome and sample tag information were reverse-transcribed into cDNA by using the BD Rhapsody cDNA Kit. Libraries were created using the BD Rhapsody Whole Transcriptome Analysis Amplification Kit and sequenced on the NovaSeq 6000 platform (Illumina; 150-bp paired-end run).

## Single-cell data analysis

The BD Rhapsody Whole Transcriptome Analysis Pipeline (version 1.9.1) based on GRCh38 mRNA annotation was used to process and demultiplex sequencing data (fastq file). The resulting output files were analyzed and visualized using SeqGeq v1.7 software (BD Biosciences). The quality control criteria were set at cells containing between 200 and 4000 expressed genes. The built-in R Plugin Seurat Clustering Pipeline was used for dimensionality reduction, graph-based KNN clustering, and differential gene expression analysis. Uniform manifold approximation and projection (UMAP) layouts and heatmaps were generated to visualize differentially expressed cell populations among the PBMCs derived from the three samples as described previously. In brief, principal component analysis was performed on the basis of highly variable genes, and a two-dimensional projection was generated in the UMAP space by using the first 50 principal components. Cells labeled as "undetermined" or "multiplet" were excluded from the subsequent

analyses. The UMAP approach identified 11 clusters. The top 30 upregulated genes in adjacent cluster contrasts were identified on the basis of a false discovery rate of <0.01 and a log2 fold change of >0.25, sorted by log2 fold change. The presence of sequenced oligonucleotide-conjugated antibodies (cell surface markers), together with the top four enriched pathways of the top 30 genes in each cluster, was used to identify cell types. The top enriched pathways were detected using STRING v.11.5 network analysis against the following criteria: count in network >4, false discovery rate <0.01, and strength >1.0, sorted by strength.

## Monocyte cold stimulation

Incubation of monocytes at $32\,°C$ has been used to evaluate the hyperinflammatory responses to mild cold exposure in patients of Familial cold autoinflammatory syndrome (FACS) (Rosengren et al, 2007). Therefore, for cold stimulation assays, two million PBMC-derived monocytes or THP-1 monocytes were incubated in complete Roswell Park Memorial Institute (RPMI) medium (RPMI 1640 medium supplemented with 10% fetal bovine serum and 1% penicillin–streptomycin) overnight at $37\,°C$ under 5% $CO_2$; and subsequently, the temperature of the incubator was adjusted to $32\,°C$ for 24 h to stimulate PBMC-derived monocytes. For the cold stimulation of THP-1 monocytes, different time points were used, ranging from 1 to 24 h.

## THP-1 cell culture and inhibitor pretreatment assays

The human monocytic THP-1 cell line harboring ZNF334 wild-type or truncation mutation were maintained in complete RPMI medium at $37\,°C$ in a humidified atmosphere under 5% $CO_2$. For the evaluation of the effect of inhibitor against TNF-α(adalimumab, AbbVie Inc.), Hsp90/eHsp90 (geldanamycin, Enzo, Cat#BML-EI280), or TRPM8 (N-(3-Aminopropyl)−2-[(3-methylphenyl) methoxy]-N-(2-thienylmethyl) benzamide hydrochloride, AMTB, Sigma-Aldrich, Cat#SML0103) on the attenuation of cold-induced upregulation of inflammatory genes in THP-1 monocytes, and for the evaluation of the effects of inhibitors on the attenuation of cold-induced upregulation of inflammatory genes in THP-1 monocytes harboring wild-type ZNF334 or having heterozygous ZNF334 truncation mutation (ZNF334$^{+/-}$), THP-1 monocytes were cultured at $1 \times 10^6$ cells per ml in complete RPMI medium at $37\,°C$ with or without the presence of the respective inhibitor according to previous protocols (Hornsby et al, 2022b; Hsu et al, 2007; Lin et al, 2017; Wax et al, 2003), and then being cold-stimulated at $32\,°C$. Briefly, in TNF-α inhibitor assay, THP-1 monocytes were pretreated with 10 µg/ml adalimumab, 10 µg/ml heat-inactivated ($95\,°C$, 10 min) adalimumab control (Ctrl), or medium only for 2 h, and then placed in the $37\,°C$ or $32\,°C$ incubator for another 2 h. Relative fold change of *TNF, NLRP3,* or *IL-6* mRNA expression was calculated as compared with the mean expression level in cells cultured at $37\,°C$ with medium only. In Hsp90 inhibitor assay, THP-1 monocytes were pretreated with 2 µM geldanamycin or DMSO only for 1 h, and then placed in the $37\,°C$ or $32\,°C$ incubator for another 24 h. Relative fold change of *TNF, NLRP3* or *IL-6* mRNA expression was calculated as compared with the mean expression level in cells cultured at $37\,°C$ with DMSO only. In TRPM8 inhibitor assay, THP-1 monocytes were pretreated with 10 µM AMTB or DMSO only for 24 h, and then placed in the $37\,°C$

or 32 °C incubator for another 24 h. Relative fold change of *TNF, NLRP3* or *IL-6* mRNA expression was calculated as compared with the mean expression level in cells cultured at 37 °C with DMSO only. In the ruxolitinib inhibitor assay, THP-1 monocytes were pretreated with 1 μM ruxolitinib or DMSO only for 1 h, and then placed in the 37 °C or 32 °C incubator for another 24 h (Prutsch et al, 2024). Relative fold change of *TNF, NLRP3*, or *IL-6* mRNA expression was calculated as compared with the mean expression level in cells cultured at 37 °C with DMSO only.

## CRISPR/Cas9 gene editing

CRISPR/Cas9 ribonucleoprotein–mediated genome editing was performed to establish heterozygous and homozygous *ZNF334* loss-of-function THP-1 cell lines. THP-1 monocytes harboring the heterozygous *ZNF334* truncation mutation were designated as ZNF334$^{+/-}$ THP-1 cells, whereas THP-1 monocytes harboring the homozygous *ZNF334* truncation mutation were designated as ZNF334$^{-/-}$ THP-1 cells. In brief, an efficient gRNA was designed to remove an exon region present in all the coding transcripts of ZNF334, thus creating a truncation mutation (Fig. EV2A). The synthetic, chemically modified gRNA was obtained from Dharmacon (CO, USA). THP-1 cell lines were synchronized with 200 ng/mL nocodazole for 17 h; this was followed by Cas9/gRNA delivery through electroporation. On the day of electroporation, a gRNA complex solution (150 pmol) containing complementary RNA and transactivating CRISPR RNA (tracrRNA) was mixed with Cas9 protein (30 pmol) in Opti-MEM Reduced Serum Medium (Thermo Fisher, Cat#31985062) at room temperature for 15 min to form Cas9/gRNA ribonucleoproteins. Simultaneously, $1 \times 10^5$ synchronized cells were collected through centrifugation, washed with Opti-MEM twice, and resuspended in optimal volumes of Opti-MEM; this was followed by the addition of Cas9/gRNA ribonucleoproteins. Then, the ribonucleoproteins were delivered into the cell by using the NEPA21 electroporator under optimized pulse voltage conditions. The electroporated cells were immediately transferred to 24-well plates containing 1 mL of complete RPMI medium and then incubated at 37 °C under 5% $CO_2$. After a 48-h recovery period, we performed a T7E1 digestion assay and high-resolution melting analysis to evaluate the efficiency of gene editing. THP-1 cells containing the gene edit were seeded, and single-cell clones were subjected to high-resolution melt-qPCR analysis to verify mutations in the target gene (Fig. EV2B).

## RNA-sequencing and differential gene expression analysis

RNAs were extracted from wild-type, ZNF334$^{+/-}$, and ZNF334$^{-/-}$ THP-1 cells at 37 °C or after 8 h of incubation at 32°C using QIAGEN RNeasy Micro kit (cat# 74004, QIAGEN GmbH, Hilden, Germany). Total RNA was subjected to high-throughput sequencing using the Illumina NextSeq 2000 at the Cancer and Immunology Research Center at National Yang Ming Chiao Tung University. Raw reads were processed using CLC Genomics Workbench v25.0.1. Filtered reads were aligned to the human reference genome (GRCh38), and rRNA reads were identified and filtered using the SILVA rRNA database (v138.1). Processed RNA sequencing count data (Gene expression matrices) were imported

into RStudio running R (v4.5.0) packages. To reduce background noise, genes with low expression (fewer than 10 counts across all samples) were removed, except for *ZNF334*, which was retained due to prior biological relevance. Filtered counts were normalized using the median-of-ratios method implemented in the DESeq2 package (v1.48.1), with sample-specific size factors estimated from the distribution of counts. Multiple pairwise comparisons of gene expression values were made between ZNF334$^{+/-}$ or ZNF334$^{-/-}$ with ZNF-wild type at 37 °C or at 32 °C. Genes with an adjusted *P* value (FDR) < 0.05 and |log2 fold change| ≥0.8 were considered significantly differentially expressed. Adjusted *P* values were calculated by the Benjamini–Hochberg procedure. String v.12.0. (string-db.org) was used to predict potential interactions among these differentially expressed genes and to analyze enriched pathways in the network.

## Synthesis and transfection of in vitro-transcribed mRNA (IVT-mRNA) expressing ZNF334-WT or ZNF334-mutant

Briefly, mRNA synthesis was carried out using the HiScribe T7 mRNA Kit with CleanCap Reagent AG (New England Biolab, USA, Cat#E2080) according to the manufacturer's instructions. The transfection of IVT-mRNAs in primary monocytes was performed using Lipofectamine MessengerMAX (Thermo Scientific, Cat#LMRNA008). Monocytes from healthy controls were seeded in a 96-well plate, and the IVT-mRNAs in Opti-MEM medium were subsequently added to Lipofectamine MessengerMAX solution at a 1:1.5 ratio and mixed well. The Lipomessenger Max-mRNA complexes were then added to each well. Cells were incubated at 37 °C for 2 h, with or without subsequent cold stimulation at 32 °C for 6 h. RNAs were extracted from cells afterward for RT-PCR.

## Gene expression analysis through real-time PCR

Total RNA was extracted from monocytes or THP-1 cells by using TRIzol (Cyrusbioscience, New Taipei City, Taiwan, Cat#2001) and reverse-transcribed by using the cDNA Reverse Transcriptase Kit (Takara Bio USA, San Jose, CA, USA, Cat#RR037B). Real-time PCR (RT-PCR) was performed using the SYBR FAST qPCR Master Mix (KAPA Biosystems, Cat#KK4600) or the IQ2 TagMan Probe qPCR Master Mix (Bio-Genesis, Cat#BB-DBU-010). The cycling conditions were as follows: one preincubation cycle at 95 °C for 30 s; 50 amplification cycles at 95 °C for 10 s, 60 °C for 30 s, and 72 °C for 10 s; and a final cooling step at 40 °C for 30 s. The peptidylprolyl isomerase B (*PPIB*) gene was used as the housekeeping gene. The comparative threshold cycle (Ct) method was used to calculate the expression levels of the target genes: $\Delta Ct = Ct^{target\ gene} - Ct^{PPIB\ (housekeeping\ gene)}$. PCR primers are listed in Appendix Table S3.

## ROS detection

THP-1 monocytes were incubated with 20 μM DCFDA (Abcam, Cat#ab113851) in phenol red-free RPMI medium at 37 °C. For cold stimulation assays, cells were exposed to 32 °C for 2 h in the incubator, and were harvested and centrifuged to remove the DCFDA-containing medium. The cells were then analyzed in the FITC channel of FACSCanto II flow cytometer.

## Cell death and cell cycle analyses in cold-stimulated THP-1 cells

THP-1 cell lines (ZNF334-wild-type, ZNF334$^{+/-}$, and ZNF334$^{-/-}$) were seeded onto 96-well imaging plates (Greiner); and stained with Hoechst 33258 (Sigma, Cat# 23491-45-4), Annexin V (BD Biosciences, Cat# 51-65874X), propidium iodide (PI; BD Biosciences, Cat#51-66211E), and the cell membrane stain CellVue (Merck, Cat#CB_000730). Then, the plates were placed in an environmentally controlled live cell imaging system; the obtained images were analyzed using MetaXpress (Molecular Devices). The application module using multiwavelength cell scoring settings was used to analyze membrane integrity per cell and the rate of apoptotic and necroptotic cells per image field. Because a higher PI intensity can differentiate necroptotic cells from apoptotic cells (Pietkiewicz et al, 2015), we defined the necroptotic gating threshold as triple-positivity for Hoechst, Annexin V, and higher PI intensity (>10,000 gray levels of intensity above local background; Fig. EV4) and the early apoptotic gating threshold as double-positivity for Hoechst and Annexin V. Cell cycle analysis was performed using the Cell Cycle Module of MetaXpress to determine the cold stimulation–induced changes in the proportions of cells in G1, S, and G2 phases.

## Immunofluorescence and confocal microscopy

A monolayer of THP-1 cells was attached to coverslips by using the Cytospin centrifugation method. The cells were fixed with 100% methanol for 15 min and blocked with 1% bovine serum albumin (Gibco, Cat#30063481) for 30 min on a shaker. Then, the cells were stained with primary antibodies against ZNF334 (Invitrogen, PA5-31843; 1:100 dilution), HSP90α/β (Santa Cruz, F-8, SC-13119; 1:100 dilution), TRPM8 (ECM, M571, TM5711, 1:100 dilution; Invitrogen, PA1-46239, 1:100 dilution), STAT3 (Invitrogen, 9D8, MA1-13042; 1:100 dilution), phospho-NF-κB p65 (Ser536) (Cell Signaling, 93H1, 3033, 1:100 dilution; Santa Cruz, 27.Ser 536, sc-136548, 1:100 dilution), phospho-STAT3 (Tyr705) (Cell Signaling, D3A7, 9145, 1:100 dilution), and Calreticulin (Invitrogen, 1G6A7, MA5-15382, 1:100 dilution) overnight at 4 °C. Next, the cells were washed and stained with fluorochrome-conjugated secondary antibodies, including anti-rabbit Alexa Fluor 488 (Invitrogen, A-11008; 1:200 dilution) and anti-mouse Alexa Fluor 594 (Invitrogen, A-11005; 1:200 dilution). For multicolor (>3 colors) staining, an Alexa Fluor 647 conjugation kit (Abcam, Cat#ab269823) was used for labeling. Coverslips were mounted with an antifade mounting medium containing DAPI (Abcam, Cat#ab104139). The slides were analyzed under a confocal laser scanning microscope (ZEISS, LSM 800 microscope). ImageJ (Fiji, v.1.53.t) was used to quantify the intensity of positive staining. For the detection of mitochondria-containing EVs, cells were labeled with 100 nM Mitotracker (Invitrogen, Cat#M22426) at 37 °C for 30 min before Cytospin centrifugation and further staining.

## Isolation of cold-induced extracellular microvesicles

Cold-induced extracellular microvesicles ranging in size from 0.45 to 1 μm were isolated from cold-stimulated (32 °C 8 h) ZNF334 wild-type, ZNF334$^{+/-}$, and ZNF334$^{-/-}$ THP-1 monocytes via qEV size-exclusion chromatography columns (Izon Science, cat#IC1-70)

following the manufacturer's protocol. The isolated fraction containing microvesicles ranging in size from 70 nm to 1 μm were collected and further filtered through a 0.45-μm membrane filter (catalog number: PALL® MAPM45C68), to obtain microvesicles ranging in size from 0.45 to 1 μm. These microvesicles derived from ZNF334-wild-type or ZNF334 knockout THP-1 cells were transferred to ZNF334 wild-type THP-1 monocytes, and the supernatants were collected 48 h after incubation at 37 °C for cytokine analysis using Bio-Plex Pro Human Cytokine 27-plex Assay (BioRad, Cat#M500KCAF0Y).

## Immunoblotting and immunoprecipitation

For immunoblotting, monocytes derived from 2 million PBMCs or THP-1 cells were incubated with a lysis buffer (RIPA buffer containing a 1× phosphatase inhibitor cocktail and 1 mM phenylmethylsulfonyl fluoride) for 30 min on ice. Proteins in cell lysates were resolved through sodium dodecyl sulfate–polyacrylamide gel electrophoresis (SDS–PAGE) and transferred to polyvinylidene difluoride membranes. Then, the membranes were probed with appropriate primary antibodies, washed, and incubated with relevant horseradish peroxidase–conjugated secondary antibodies (Thermo Scientific). After washing, the membranes were incubated in SuperSignal West Pico Plus substrate (Thermo Scientific, Cat#34580) to visualize the proteins of interest. The No-Stain Protein Labeling Reagent (Invitrogen, Cat#A44449) was used as a loading control. Quantification of the blotted band relative to the loading control was performed using iBright Analysis software (version 5.1.0).

For immunoprecipitation assays, cell lysates were first mixed with specific primary antibodies and Complete Co-IP/Wash Buffer (Universal Magnetic Co-IP kit, Active Motif, CA, USA, Cat#54002) at 4 °C on a rotator for 1.5 h, and then incubated with Protein G Magnetic Beads (Active Motif, Cat#54002) at 4 °C on a rotator overnight. Afterward, the precipitants were washed and eluted in accordance with the manufacturer's instructions (Active Motif, Cat#54002), and the immunoprecipitated proteins were separated through SDS–PAGE. Antibodies used for immunoblotting and immunoprecipitation were ZNF334 (Invitrogen, PA5-31843), HSP90α/β (Santa Cruz, sc-13119), and TRPM8 (Invitrogen, PA1-46239), and IKKα (Cell Signaling, 3G12, 11930) at 1:1000 dilution for immunoblotting and 1:20 dilution for immunoprecipitation.

### Amplicon-based deep sequencing

To detect other low-frequency acquired genetic variants potentially related to cold-induced autoinflammation, genomic DNAs extracted from PBMCs of the patient and her *ZNF334*-wild type sister were further deep-sequenced using an amplicon-based deep sequencing (ADS) method. Briefly, amplicons covering the candidate autoinflammation-related genes *NLRP3* (exon 2, 3, 4, and 5), *NLRP12* (exon 1, 3, 4, 5, and 9), *NLRC4* (exon 3, and 4), *PLCG2* (exon 19, 20, 21, 22, 24, 27 and 30), *TNFRSF1A* (exon 2, 3, 4, 6, and 10), and *ZNF334* (exon 4, and 5) were designed and obtained by using PCR amplification (primers are shown in Appendix Table S5). This set included both common single-nucleotide polymorphisms and pathogenic/likely pathogenic mutations reported in gnomAD v4.1.0 and the registry of Hereditary Auto-inflammatory Disorders Mutations (https://infevers.umai-montpellier.fr/). All exons were first amplified by 2× KAPA2G

**The paper explained**

**Problem**
Dysregulation of monocyte stress response to environmental coldness can lead to autoinflammatory diseases (AIDs), however, the direct link between cold exposure and autoinflammation is still unclear. Understanding the regulatory mechanism in the human innate immune system to prevent cold stress-induced excessive cell death and hyper-inflammation is crucial to develop novel treatments for cold-induced or cold-aggravated diseases.

**Results**
In this study, we found that in human monocytes, ZNF334 maintains endogenous intracellular expressions of heat shock protein 90 (Hsp90), a major stress regulator, and transient receptor potential melastatin 8 (TRPM8), a mild cold sensor and regulator of ROS, and negatively regulates the nuclear translocations of the proinflammatory molecules p-p65 and p-STAT3 upon cold exposure. A truncation mutation in zinc finger protein 334 (ZNF334) impairs its regulatory function and results in a previously undescribed late-onset AID induced by repeated exposure to mild cold air.

**Impact**
This manuscript identifies ZNF334 as a regulatory hub of cold stress response in monocytes, and shows that in addition to cold avoidance, geldanamycin (an inhibitor of Hsp90) might be a potential treatment to alleviate cold-stimulated hyperinflammation in ZNF334-associated AID.

Fast Multiplex Mix (KK5801) and then applied to library construction by using KAPA EvoPrep Kits (10154039001) according to the standard instruction for use. The resulting libraries were assessed to qualification, and quantification by Qsep100 (Bioptic) and Qubit Fluorometer (Thermo Fisher Scientific), respectively. The sequence read data were generated using iSeq100 sequencing system (Illumina, San Diego, Calif) with 2*150 bp paired-end reads, with a minimal average depth 1000× for each library. Raw sequencing reads were subjected to quality assessment and adapter sequence removal utilizing Trimmomatic (Bolger et al, 2014) and Cutadapt software (Martin, 2011) packages. High-quality paired-end reads were subsequently aligned to the human reference genome (GRCh38/hg38) using the Burrows–Wheeler Aligner MEM algorithm. Amplicon-specific sequence demultiplexing was conducted prior to somatic variant detection using the Genome Analysis Toolkit Mutect2 caller (GATK, Broad Institute). Identified variants underwent stringent filtering based on sequencing depth and variant allele frequency criteria. Complex insertion-deletion events were consolidated, and variant calls were normalized using BCFtools (Danecek et al, 2021). Variant authenticity was validated through independent read alignment analysis.

## Statistics

Statistical analyses were performed using GraphPad Prism software, version 10.2.3 (Graphpad Software). The non-parametric Kruskal–Wallis test was performed for multiple comparisons; and Wilcoxon signed-rank test was used for paired comparisons. Significant differences were reported if $P < 0.05$. No blinding was applied to our data analysis.

## Study approval

The study protocols were approved by the Institutional Research Ethics Committee of China Medical University Hospital (protocol numbers: CMUH109-REC3-165, CMUH110-REC3-015, and CMUH111-REC2-228). Written informed consent was obtained from each participant in accordance with the Helsinki Declaration. Written informed consent was received from the patient for the use of the photographs, and that the record of informed consent has been retained. In addition to the principles set out in the WMA Declaration of Helsinki the experiments with patient samples also conformed to the Department of Health and Human Services Belmont Report.

## Data availability

The datasets produced in this study are available in the following databases: RNA-seq data for ZNF334-edited THP-1 monocytes: ArrayExpress-Functional Genomic Data, accession id E-MTAB-15463. The source data of this paper are collected in the following database record: Biostudies, accession id S-BSST2106 (https://doi.org/10.6019/S-BSST2106).

The source data of this paper are collected in the following database record: biostudies:S-SCDT-10_1038-S44321-025-00328-x.

## Peer review information

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

## Acknowledgements

We thank the patient and her family for their contribution to this study. We thank Chia-Hsun Chang, Wei-Ting Cheng, and Cheng-Hung Wang for assisting in sample collection. We thank Vignesh Kumar Balasubramanian for technical assistance. This study was supported by the National Science and Technology Council, Taiwan (grant numbers: 110-2314-B-039-017, 112-2314-B-039-030-MY3, 112-2314-B-182A-155-MY3), China Medical University Hsinchu Hospital, Taiwan (grant numbers: CMUHCH-DMR-111-001, CMUHCH-DMR-111-029, and CMUHCH-DMR-112-012), and China Medical University, Taiwan (CMU105-BC-1-3), and Chang Gung Memorial Hospital, Linkou (NMRPG3N6252). We acknowledge the technical services provided by the Bioimaging Core Facility of the National Core Facility for Biopharmaceuticals, National Science and Technology Council, Taiwan, and the Instrumentation Center at National Tsing Hua University, Taiwan. The authors also acknowledge the RNA-sequencing technical services provided by the "National Genomics Center for Clinical and Biotechnological Applications" of the Cancer and Immunology Research Center at National Yang Ming Chiao Tung University, and the National Core Facility for Biopharmaceuticals (NCFB), National Science and Technology Council. For cell sorting, the authors would like to acknowledge the technical support of the Maintenance Project of the Advanced Immunology Laboratory at Chang Gung Memorial Hospital.

## Author contributions

**Joung-Liang Lan**: Conceptualization; Resources; Formal analysis; Funding acquisition; Investigation; Project administration; Writing—review and editing. **Shih-Hsin Chang**: Investigation. **Yi-Hua Lai**: Investigation. **Ju-Pi Li**: Data curation. **Guan-Jun Chen**: Formal analysis; Validation; Investigation. **Yen-Ju Lin**: Data curation; Methodology. **Ya-Ling Huang**: Investigation. **Bor-Luen Chiang**: Conceptualization. **Jan-Gowth Chang**: Methodology. **Guan-Yun Lu**: Validation; Investigation. **Tung-Lin Tsai**: Validation; Investigation; Visualization. **Chien-Yu Lin**: Validation. **John Wang**: Methodology. **Yi-Chuan Li**: Formal analysis; Investigation. **Mien-Chie Hung**: Supervision; Writing—review and editing. **Chin-An Yang**: Conceptualization; Data curation; Formal analysis; Supervision; Funding acquisition; Investigation; Visualization; Methodology; Writing—original draft; Project administration; Writing—review and editing.

Source data underlying figure panels in this paper may have individual authorship assigned. Where available, figure panel/source data authorship is listed in the following database record: biostudies:S-SCDT-10_1038-S44321-025-00328-x.

## Disclosure and competing interests statement

The authors declare no competing interests.

# Expanded View Figures

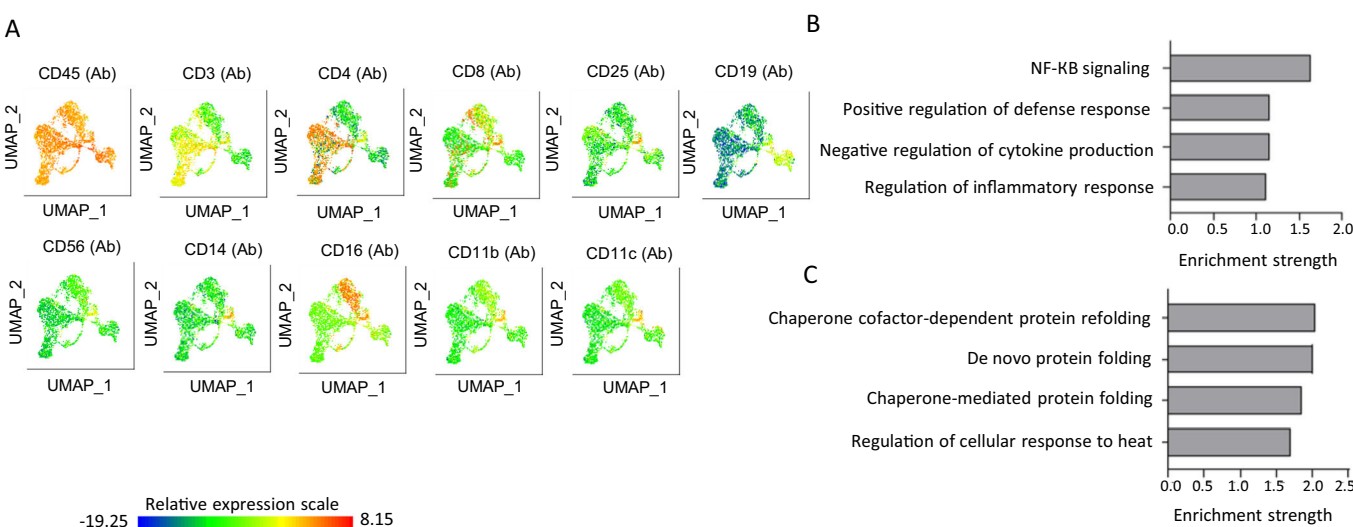

**Figure EV1.  Single cell analysis of PBMCs.**

(A) UMAPs of PBMCs annotated by the expression of each oligonucleotide-conjugated antibody targeting a cell surface marker. (B) Top four enriched pathways of the top 30 upregulated genes in the cluster annotated as "leukocyte regulating NF-κB signaling." (C) Top four enriched pathways of the top 30 upregulated genes in the cluster annotated as "myeloid cell regulating stress response."

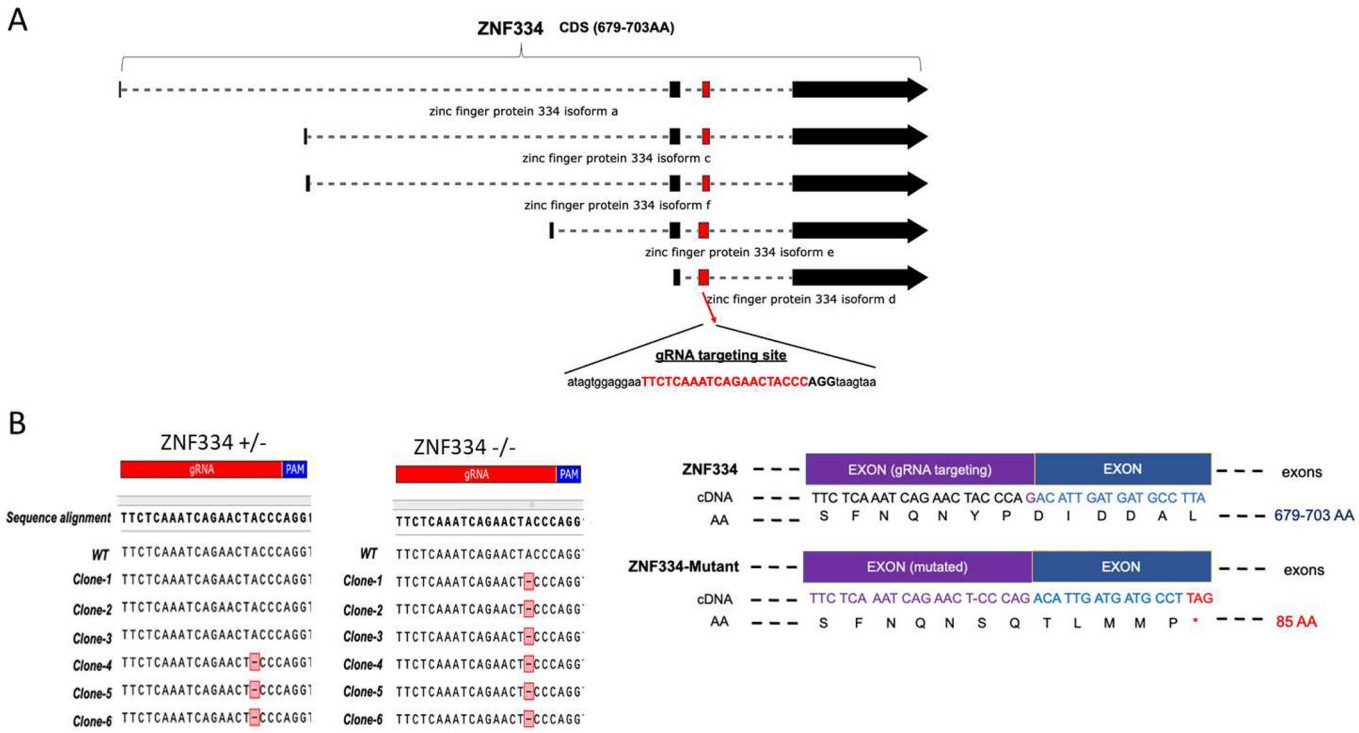

**Figure EV2. CRISPR/Cas9-mediated gene knockout of *ZNF334* in the THP-1 cell line.**

(A) Schematic of the coding transcripts of ZNF344 and the gRNA targeting site in the exon, which is present in all transcripts. (B) *ZNF334*-mutant clone has an "A" base deletion in the gRNA targeting exon of ZNF334, which produces a premature stop codon (PTC) on the mutant allele. The mutant allele is predicted to create a short, truncated protein (85 amino acids).

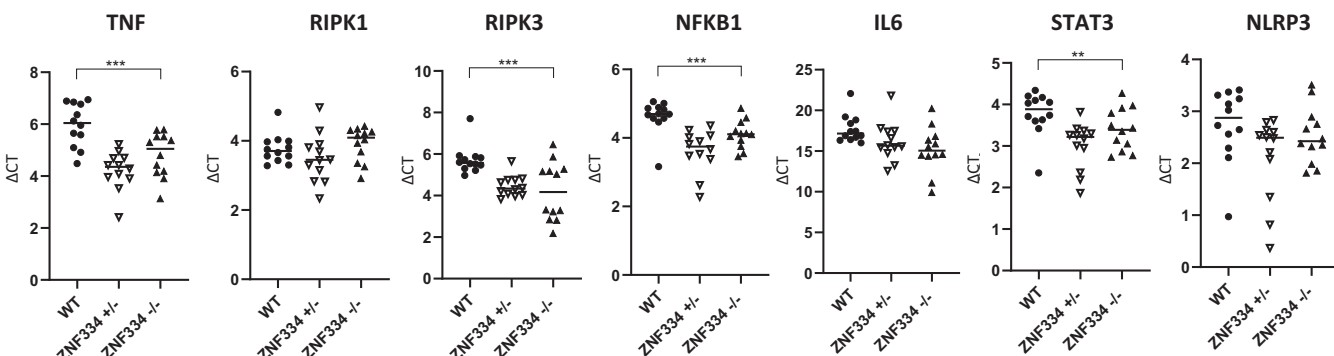

**Figure EV3.  Basal RNA expression of genes associated with inflammatory/stress response pathways in ZNF334 wild-type, ZNF334$^{+/-}$, and ZNF334$^{-/-}$ THP-1 cells.**

Relative RNA expression levels are presented as the ΔCt values of *TNF, RIPK1, RIPK3, NFKB1, IL-6, STAT3*, and *NLRP3*. THP-1 ZNF334 wild-type, $n = 12$; ZNF334 $+/-$, $n = 12$; ZNF334 $-/-$, $n = 12$. ΔCt = Ct value of the target gene $-$ Ct value of *PPIP* (housekeeping gene). **$P < 0.01$ and ***$P < 0.001$: Kruskal–Wallis test. Lines represent medians.

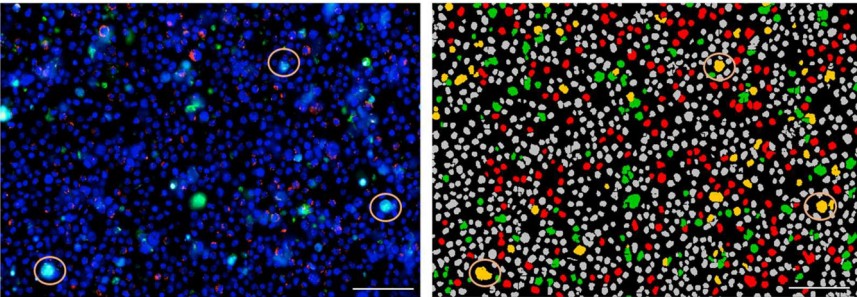

**Figure EV4. Representative image of ZNF334$^{+/-}$ THP-1 cells after 4 h of cold stimulation at 32 °C.**

Representative necroptotic cells are circled. Left: Annexin V (green) and PI (red). Right: yellow masks indicate necroptotic cells positive for nuclei stain (gray) and Annexin V (green) and PI with an intensity of >10,000 gray levels above the background (red). Images were acquired using the Celldiscoverer7 Micro wide-field fluorescence microscope (ZEISS) and analyzed using the MetaXpress software. Scale bar, 100 µm.

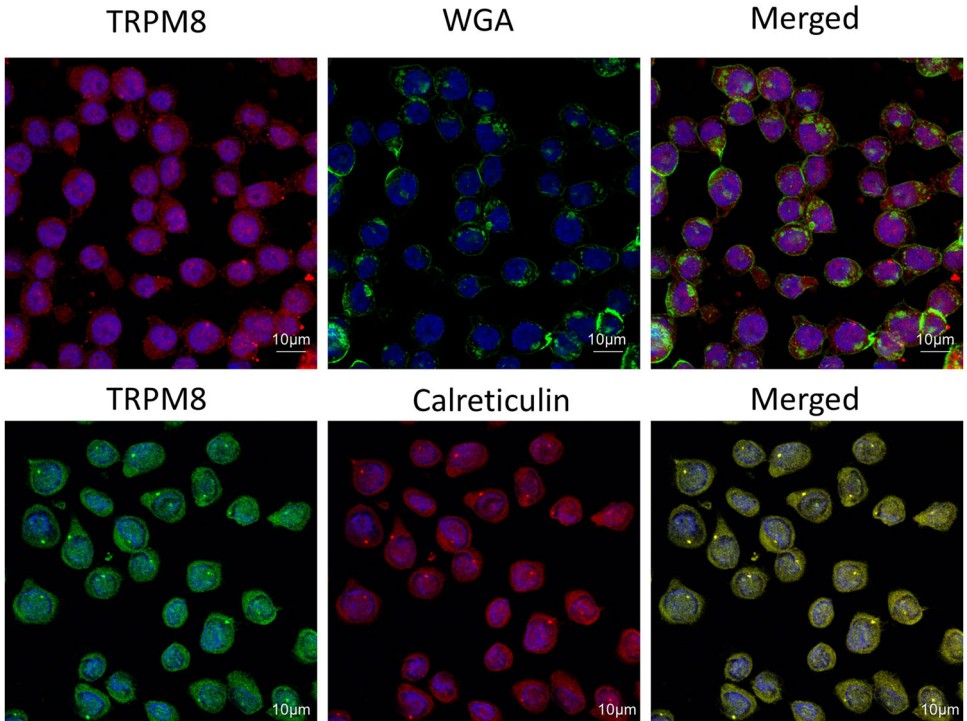

**Figure EV5.  Colocalization of TRPM8 with WGA and Calreticulin.**

THP-1 monocytes (ZNF334 wild-type, at baseline condition) were stained with TRPM8, WGA (plasma membrane stain), calreticulin (ER stain), and DAPI (nuclear stain, blue). Upper graphs: Representative confocal microscopic images showing colocalization of TRPM8 and WGA. Lower graphs: representative confocal microscopic images showing colocalization of TRPM8 and calreticulin.

