## [Peer Review File · EMBO Molecular Medicine]

ZNF334 truncation mutation drives cold-induced autoinflammation

Joung-Liang Lan, Shih-Hsin Chang, Yi-Hua Lai, Ju-Pi Li, Guan-Jun Chen, Yen-Ju Lin, Ya-Ling Huang, Bor-Luen Chiang, Jan-Gowth Chang, Guan-Yun Lu, Tung-Lin Tsai, Chien-Yu Lin, John Wang, Yi-Chuan Li, Mien-Chie Hung, and Chin-An Yang

Corresponding author: Chin-An Yang (yangca@cgmh.org.tw)

Review Timeline:

Submission Date:	21st Dec 24
Editorial Decision:	14th Feb 25
Revision Received:	1st Jul 25
Editorial Decision:	14th Jul 25
Revision Received:	26th Sep 25
Accepted:	9th Oct 25

Editor: Zeljko Durdevic

Transaction Report:

14th Feb 2025

Dear Prof. Yang,

Thank you for the submission of your manuscript to EMBO Molecular Medicine. We have now received feedback from the three reviewers who agreed to evaluate your manuscript. All three referees recognize interest of the study but also raise serious and partially overlapping concerns that should be addressed in a major revision. Focus of the revision should be in providing stronger genetic evidence for ZNF334 mutation as a cause of the disease. If you would like to discuss further the points raised by the referees, I am available to do so via email or video. Let me know if you are interested in this option.

We would welcome the submission of a revised version within three months for further consideration. Please let us know if you require longer to complete the revision.

I look forward to receiving your revised manuscript.

Yours sincerely,

Zeljko Durdevic

We require:

- 1) A .docx formatted version of the manuscript text (including legends for main figures, EV figures and tables). Please make sure that the changes are highlighted to be clearly visible.
- 2) Individual production quality figure files as .eps, .tif, .jpg (one file per figure). For guidance, download the 'Figure Guide PDF': (<https://www.embopress.org/page/journal/17574684/authorguide#figureformat>).
- 3) A .docx formatted letter INCLUDING the reviewers' reports and your detailed point-by-point responses to their comments. As part of the EMBO Press transparent editorial process, the point-by-point response is part of the Review Process File (RPF), which will be published alongside your paper.
- 4) A complete author checklist, which you can download from our author guidelines (<https://www.embopress.org/page/journal/17574684/authorguide#submissionofrevisions>). Please insert information in the checklist that is also reflected in the manuscript. The completed author checklist will also be part of the RPF.
- 5) Please note that all corresponding authors are required to supply an ORCID ID for their name upon submission of a revised manuscript.
- 6) It is mandatory to include a 'Data Availability' section after the Materials and Methods. Before submitting your revision, primary

datasets produced in this study need to be deposited in an appropriate public database, and the accession numbers and database listed under 'Data Availability'. Please remember to provide a reviewer password if the datasets are not yet public (see <https://www.embopress.org/page/journal/17574684/authorguide#dataavailability>).

12) Author contributions: You will be asked to provide CRediT (Contributor Role Taxonomy) terms in the submission system. These replace a narrative author contribution section in the manuscript.

13) A Conflict of Interest statement should be provided in the main text.

14) Every published paper now includes a 'Synopsis' to further enhance discoverability. Synopses are displayed on the journal webpage and are freely accessible to all readers. They include a short stand first (maximum of 300 characters, including space) as well as 2-5 one-sentences bullet points that summarizes the paper. Please write the bullet points to summarize the key NEW findings. They should be designed to be complementary to the abstract - i.e. not repeat the same text. We encourage inclusion

of key acronyms and quantitative information (maximum of 30 words / bullet point). Please use the passive voice. Please attach these in a separate file or send them by email, we will incorporate them accordingly.

15) Include a Reagents and Tools Table as part of the Methods section, which can be downloaded from our author guidelines (<https://www.embopress.org/page/journal/17574684/authorguide#structuredmethods>)

**** Reviewer's comments ****

Referee #1 (Comments on Novelty/Model System for Author):

This is a very nice study, describing and defining a novel inflammatory entity. Of course, the model is essentially cellular, and I would have preferred a KO mouse model reproducing the exact or a similar mutation discovered in the patient. But I also acknowledge the difficulty and the cost of building such animal model. An alternative possibility would have been to test knock out model in zebrafish with morpholinos, and evaluate the input on macrophages activation. But again, this requires a specific facility that may not be available to the authors. Anyway, these options deserve to be discussed in the manuscript.

Referee #1 (Remarks for Author):

The paper is very well written and the experiments of high quality. There are, in my opinion, some information which is missing.

- 1) the frequency of the mutation in both the general population and in Han Chinese needs to be indicated.
- 2) the conservation of the protein, with a focus on the affected domain and mutated amino acid also needs to be shown to appreciate the evolutionary impact of the mutation; this would also help discussing some appropriate animal models to be tested in future experiments.
- 3) the KO mutation in THP-1 cells is interesting, but I wonder why the authors did not perform a full transcriptomic analysis (RNAseq) followed by GO term searches in order to have an extended and non-biased view of the impact of ZNF334 deletion. This would greatly ameliorate the overall understanding of the role of this protein in macrophages.
- 4) Figure 5A is not clear to me. First, I feel that the data of ZNF334 KO follow a bimodal distribution. Explain this. Second, I don't see a difference between +/- and -/- cells, whether placed at 32 or at 38 {degree sign}C.... please, explain.
- 5) Figure 6 describes cells transfected with in vitro generated wild type or mutated transcripts. This is an unusual way to evaluate the impact of a mutation. I would have preferred a Crisp-generated KI cell line....
- 6) Figure 7 describes the impact of potential treatments in cells. Again, the in vitro model shows here its inherent limitations. Of course, an in vivo model would have been highly relevant. The impact of anti-IL1b antibodies would have been interesting. Was it even considered as a therapeutic option in this patient? What about Jak inhibitors??

Referee #2 (Comments on Novelty/Model System for Author):

It's a single patient.

Referee #2 (Remarks for Author):

In their manuscript, "ZNF334 truncation mutation drives cold induced autoinflammation and sensorineural hearing loss.", Lan et al. take a methodical and comprehensive approach to describe the phenotype of a unique patient, characterize the functional significance of a variant in a novel disease gene and propose a disease mechanism which could be relevant for normal physiologic responses. They study patient and control blood derived monocytes and recombinant THP1 cells using a variety of cutting edge technology including, ChIP-seq and CRISPR/Cas9 editing as well as standard techniques such as ELISA, WB/IP, real time PCR, cell death and cell cycle analysis on primary cell and a recombinant THP1 cell line

The strengths of this paper are

1. Novel disease gene with unique combination of clinical features
2. Detailed structural modeling
3. demonstration of functional effect of this frameshift mutation.
4. Use of unaffected sibling as control

5. extravasicle data, and

The primary weaknesses

1. Single patient with mutation without parental sequencing and late onset presentation
2. Only one related control sample studied which is inadequate to establish level of intersubject variability
3. Inadequate clinical description, skin pathology
4. Inadequate explanation for why specific studies were performed
5. Complicated model that doesn't come together
6. Manuscript would benefit from editing by an English speaker

Comments:

Title: I'm not sure the hearing loss should be in the title. While it is an important clinical feature it is not studied and only discussed briefly in discussion

Abstract: Concise

Introduction

Appropriate review of current state of knowledge and summary of findings of this paper

Specific comments

Paragraph 1 sentence 3 doesn't make sense

Paragraph 2 first sentence is inaccurate -FCAS is prototypical disease due to cold exposure

Results

Not enough detail of clinical presentation for a clinician to recognize this disease

Improper use of word anamnesis

It appears that this patient has 2 different types of rashes

Were the 2 biopsies taken from different types of lesions

Why was p65 and stat3 stained for. Were there other antibodies that were not observed

Figure legends are adequately detailed although n's are not always listed and not always clear if the data points are technical or experimental replicates

Figure 1

A. Its not clear if there is any pathology here

B. Staining should be labeled and HE of similar region should be shown to see which cells are staining

C. Color coding is difficult to grasp here - clearly the patient has chronic inflammation but it is unusual that ESR went up when CRP went down in 2015

D. Poor quality pictures

E. Useful domain model

Figure 2

A. Nomenclature of cold +++ and is + is difficult to follow

B. Authors need to briefly describe cite-seq in manuscript, not just in the methods

C. Clear differences in UMAP, but authors should walk the reader through the significance

D. What is the significance of this

E. Statistical differences are difficult to see

Figure 3

A. Clear

B. I agree with conclusions Loading controls do not appear to be loaded evenly

Figure 4

Significance of NLRP3 expression data is difficult to interpret as this does not demonstrate NLRP3 activation. Connection of this disease to NLRP3 is the weakest part of the paper.

Figure 5

Clear

Figure 6

A. domain model is clear

B. The significant of some of these graphs are problematic

C. molecular modeling Clear
D through H clear

Figure 7
Hard to follow mechanism

Figure 8
Was this data used to decide how to treat the patient or did she just stay on steroids and methotrexate

Discussion
A little long but conclusions and relevance are valid

Methods
Adequate detail of techniques

References - Mostly adequate

Paper describing Monocyte cold stimulation assay is not referenced

Referee #3 (Comments on Novelty/Model System for Author):

To my perspective, there is not enough genetic data to claim the association between the frameshift in ZNF334 and the patient's phenotype. Several experiments might benefit to be better controlled of potential artifact. However, if true, the identification of ZNF234 as a NLRP3 regulator would be highly novel.

Referee #3 (Remarks for Author):

In this study, Lan et al. report a patient with acquired cold-induced inflammation harboring a heterozygous frameshift mutation in ZNF334. Cells from the patient exhibit increased cold-induced inflammation compared to a non-carrier sister. This heterozygous mutation is associated with elevated ZNF334 transcript levels but an almost complete absence of protein expression in the patient's monocytes following cold exposure. Knockout ZNF334 in THP-1 cells results in elevated levels of inflammatory transcripts and increased oxidative stress. Overexpression of the truncated ZNF334 in human monocytes leads to excessive production of TNF transcripts after cold exposure.

I found major shortcomings in this manuscript, and some observations seem largely overinterpreted or only superficially addressed:

- The genetic results are not strong enough to claim that the observed frameshift mutation in ZNF334 is the cause of the disease.
- Since the disease was acquired, deep sequencing (deep-seq) seems more appropriate than classical whole-exome sequencing (WES) to identify acquired somatic mutations, which are common in CAPS.
- The authors do not discuss the presence of numerous frameshift mutations in ZNF334 within healthy individuals. For example, p.Arg284Ter is found 983 times in GnomAD, p.Glu402LysfsTer4 is found 330 times, p.Lys347AlafsTer31 is found 134 times, p.Glu409Ter is found 115 times, and so on.
- Although partial penetrance should not be excluded, the association between the patient's phenotype and the ZNF334 frameshift should be strengthened by both identifying other patients with ZNF334 mutations and a similar phenotype and excluding somatic mutations in genes driving autoinflammatory diseases using deep sequencing.
- Other genes identified through WES analysis should be provided in a table.

Figure 1:

- Only one control was used for these experiments. Although it is useful to have a healthy relative as a control, given the inter-individual variability of immune responses, additional healthy donors are needed to draw robust conclusions.

Figure 3:

- It is unclear how the authors explain that a heterozygous frameshift mutation results in an almost complete absence of protein expression in the patient's monocytes.
- Why are no classical loading controls provided?
- The authors should provide information on the purity of the studied monocytes after the overnight adhesion process, as it might be affected by the disease.

Figure 4:

- Were the same clones used per genotype in these experiments, or were multiple clones tested? A minimum of two clones per genotype is likely required to avoid potential artifacts.

- Is it expected for THP-1 cells to be almost 100% in G0/G1 at T0?
- For panels A to H: Present individual values instead of mean {plus minus} SEM. It should be clear whether the observed trend was consistent across replicates.
- I cannot find the figure comparing IL-1 β production in ZNF334 WT, +/- , and -/- at 37{degree sign}C.

Figure 5:

5A and C: it is unclear what each dot/triangle represent. An individual well? Independent experiment's?

Figure 6:

- It is unclear how the authors explain the discrepancy between ZNF IP and HSP IP in terms of HSP/ZNF interaction.

Response to reviewers

Referee #1 (Comments on Novelty/Model System for Author):

This is a very nice study, describing and defining a novel inflammatory entity. Of course, the model is essentially cellular, and I would have preferred a KO mouse model reproducing the exact or a similar mutation discovered in the patient. But I also acknowledge the difficulty and the cost of building such animal model. An alternative possibility would have been to test knock out model in zebrafish with morpholinos, and evaluate the input on macrophages activation. But again, this requires a specific facility that may not be available to the authors. Anyway, these options deserve to be discussed in the manuscript.

Our reply:

Thank you for your kind suggestions. We have addressed the need of further mouse model development in the revised discussion, the last sentence of the second last paragraph (p.32): “To further investigate the impacts of different dosage of ZNF334 truncation mutation in various immune cell types on systemic cold-induced or cold-aggravated autoinflammatory diseases, and to better evaluate the in vivo treatment responses of inhibitors, development of ZNF334 gene-edited mouse model is needed.” We have examined the conservation of the protein across species, including mouse and zebra fish in the revised results and Fig. 1G, which shows that the sequence homology was more similar between human and mouse, suggesting that mouse would be a better animal model as compared with zebra fish. Please also refer to our reply to bullet point 2.

Referee #1 (Remarks for Author):

The paper is very well written and the experiments of high quality. There are, in my opinion, some information which is missing.

1) the frequency of the mutation in both the general population and in Han chinese needs to be indicated.

Our reply:

Thank you for the suggestion. We have listed the general population frequency (ExAC) of 0.000255, East Asian frequency (gnomAD v2.1.1 controls) of 0.0006014, and Taiwan Biobank frequency of 0.000670241 in the revised Appendix Table S1,

and mentioned about the population frequency of the ZNF334 mutation in the revised Results, section 1, paragraph 2 (p.8).

2) the conservation of the protein, with a focus on the affected domain and mutated amino acid also needs to be shown to appreciate the evolutionary impact of the mutation; this would also help discussing some appropriate animal models to be tested in future experiments.

Our reply:

Please refer to the revised Results, section 1, paragraph 2 and the revised Fig. 1G: “This heterozygous frameshift mutation (c.1197_1198del, encoding p.Thr399fs) was predicted to affect multiple C2H2-type ZnF domains of ZNF334. Two nucleotides were discovered to be deleted in ZNF334, that correspond to the mRNA sequence encoding threonine (399T) and glycine (400G). This deletion resulted in the conversion of glutamine (401E) to lysine, and led to the early termination of the protein, which in turn resulted in the loss of subsequent C2H2-type ZnF domains containing several highly conserved amino acids (Fig. 1G). As presented in Fig. 1G, the amino acids 401E, 406C, and 409C immediately following the frameshift mutation were conserved, whereas the flanking amino acids demonstrated variability across species.” (p.8)

3) the KO mutation in THP-1 cells is interesting, but I wonder why the authors did not perform a full transcriptomic analysis (RNAseq) followed by GO term searches in order to have an extended and non biased view of the impact of ZNF334 deletion. This would greatly ameliorate the overall understanding of the role of this protein in macrophages.

Our reply:

Thank you. According to the suggestion, we have performed RNA-seq analysis on ZNF334 wild-type, ZNF334^{+/-}, and ZNF334^{-/-} THP-1 monocytes before and after 8 h of cold stimulation at 32°C in the revised manuscript, please refer to the revised Results, section 5 “RNA-seq of ZNF334-edited THP-1 monocytes further revealed involvement of ZNF334 in maintaining the regulatory protein folding machinery and suppressing the cold-induced inflammatory response”. In this revised paragraph, we added the results of subsequent GO term searches: “We identified the top 3 enriched downregulated biological pathways (Gene Ontology) were chaperone-mediated protein folding, response to topologically incorrect protein, and response to unfolded protein; whereas the top 3 enriched upregulated

biological pathways were defense response, immune response, and immune system process (Fig. 4R)” (p.15,16). Please also refer to the revised Fig.4R-T. Indeed, RNA-seq analysis helped us to better understand the role of ZNF334 in monocytes/macrophages. We summarized the conclusion of RNA-seq at the end of this section: “These data suggest that wild-type ZNF334 is required to maintain the ER protein folding response necessary to regulate redox homeostasis, and plays a role in suppressing cold-induced NF- κ B activation in monocytes” (p.16).

4) Figure 5A is not clear to me. first, I feel that the data of ZNF334 KO follow a bimodal distribution. Explain this. Second, I don't see a difference between +/- and -/- cells, whether placed at 32 or at 38 {degree sign}C.... please, explain.

Our reply:

Thank you for pointing this out. The Fig. 5A in the original manuscript described the change (delta, Δ) mean fluorescence intensity (MFI) of DCFDA staining (representing intracellular ROS) by calculating the [MFI of DCFDA detected at 32°C 2h – MFI of DCFDA detected at 37°C], as written in the previous figure legend. However, although all the cells were cultured according to the same protocol, variation of the baseline DCFDA MFI level was still detected in the same cell line between experiments. The increase in DCFDA MFI might be slightly lower in the experiments presenting higher baseline MFI values, contributing to the bimodal distribution. Therefore, in the revised Fig. 5A, we present the raw geometric MFI of DCFDA without subtraction of baseline level. Indeed, as shown in the revised Fig. 5A, significant differences of intracellular ROS levels were detected between WT vs. ZNF334+/-, and between WT vs. ZNF334-/- cells at baseline; while the levels of ROS were similar between ZNF334+/- and ZNF334-/- THP-1 monocytes. Of note, the percentage of early apoptotic cells in ZNF334-/- monocytes at 2h of cold stimulation was higher than that of ZNF334+/- monocytes (Fig. 4M), which might affect the mean level of ROS detected inside live cells via flow cytometry.

5) Figure 6 describes cells transfected with in vitro generated wild type or mutated transcripts. This is an usual way to evaluate the impact of a mutation. I would have preferred and Crisp-generated KI cell line....

Our reply:

We agree that a CRISPR gene knock-in cell line would be more suitable for ZNF334 mutation overexpression experiments. However, we encountered technical

challenges targeting the exact mutation locating in repeated sequences encoding the repeated C2H2 zinc finger motifs, therefore we could only generate the CRISPR/Cas9 edited ZNF334 knock-out (deleting all the C2H2 zinc finger domains) THP-1 monocytes as shown in Fig. EV2, and alternatively synthesized IVT-mRNA (as shown in Fig.6A) carrying the patient's mutation for primary monocyte transfection to further validate the functional effects of ZNF334 p.Thr399fs. We have added the statement of this limitation in the second last paragraph of the revised discussion: "our model of the pathogenic role of ZNF334 p.Thr399fs truncation mutation in ZACAS was strengthened by the hyperinflammatory cellular phenotype detected in primary monocytes transiently transfected with *in vitro* transcribed mRNA overexpressing this exact mutation. ...we were unable to "knock-in" our patient's ZNF334 frameshift mutation in THP-1 monocytes by using CRISPR/Cas9 gene editing technology without off-target effects. Therefore, we were only able to comprehensively examine molecular changes in ZNF334^{+/-} and ZNF334^{-/-} truncation "knockout" THP-1 monocytes." (p.31)

6) Figure 7 describes the impact of potential treatments in cells. Again, the *in vitro* model shows here its inherent limitations. of course, an *in vivo* model would have been highly relevant. The impact of anti-IL1b antibodies would have been interesting. Was it even considered as a therapeutic option in this patient? What about Jakinhibs??

Our reply:

Thank you for the comment. We have addressed this limitation in the revised Discussion, second last paragraph: "To further investigate the impacts of different dosage of ZNF334 truncation mutation in various immune cell types on systemic cold-induced or cold-aggravated autoinflammatory diseases, and to better evaluate the *in vivo* treatment responses of inhibitors, development of ZNF334 gene-edited mouse model is needed."(p.32) We did not consider anti- IL-1B antibody treatment in this patient, because of the reasons below, written in the revised Results: "Furthermore, considering that the patient's serum IL-1B level could be markedly decreased using immunosuppressants and avoiding cold exposures (Fig. 2A), and the high cost of anti-IL-1B antibody treatment which created barriers to accessibility, we did not test the effect of IL-1B inhibitor, either"(p.21). According to the suggestion, we added the JAK inhibitor results in the same section of the revised Results and the revised Fig.8D: "Since Janus kinase (JAK) inhibitors have been reported to reduce IL-6-induced inflammation.. we further performed pretreatment of THP-1 monocytes with 1µM ruxolitinib, a

JAK1/2 inhibitor, which however, did not significantly attenuate cold-induced TNF, NLRP3 and IL6 (Fig. 8D).”(p.22)

Referee #2 (Comments on Novelty/Model System for Author):

It's a single patient.

Our reply:

Indeed, this is the limitation of our study: since this is a late-onset rare disease induced by specific environmental factor (repeated exposures to cold), it is hard to find other patients with the same genotype and phenotypes. We have addressed this limitation in the revised Discussion (p.31), and replied to the bullet points of the related “primary weakness” of the study mentioned below. Please refer to our reply to specific points of primary weakness.

Referee #2 (Remarks for Author):

In their manuscript, "ZNF334 truncation mutation drives cold induced autoinflammation and sensorineural hearing loss.", Lan et al. take a methodical and comprehensive approach to describe the phenotype of a unique patient, characterize the functional significance of a variant in a novel disease gene and propose a disease mechanism which could be relevant for normal physiologic responses. They study patient and control blood derived monocytes and recombinant THP1 cells using a variety of cutting edge technology including, cite seq and crispr cas9 editing as well as standard techniques such as ELISA, WB/IP, real time PCR, cell death and cell cycle analysis on primary cell and a recombinant thp1 cell line

The strengths of this paper are

- 1. Novel disease gene with unique combination of clinical features**
- 2. Detailed structural modeling**
- 3. demonstration of functional effect of this frameshift mutation.**
- 4. Use of unaffected sibling as control**
- 5. extravesicle data, and**

The primary weaknesses

- 1. Single patient with mutation without parental sequencing and late onset**

presentation

Our reply:

In the revised manuscript, we further genotyped one of the healthy daughters, and added in the first section of the revised Results: “In addition to the patient’s healthy siblings, one of her daughters was also tested as wild-type in *ZNF334*, with this confirmed using Sanger sequencing. However, no data were available on the patient’s other child, who was working abroad, or on her parents, who were deceased (Fig. 1H)” (p.8). Furthermore, we addressed the possibility of somatic mutation in this patient in the revised Results: “Although the sequencing depth and ratio of the *ZNF334* frameshift variant detected in the patient’s blood cells were 42/104 and 40.38%, respectively, suggesting a heterozygous germline mutation pattern, the likelihood of this variant originating from acquired post-zygotic mosaicism could not be ruled out because of the late-onset nature of the disease. (p.8-9)” Therefore, we further sequenced *ZNF334* in monocytes (CD14+) and T cells (CD3+) isolated from the patient’s PBMCs, and the two FFPE blocks of skin lesional biopsies (please refer to the revised results section 1 and revised Fig. 1H) and our data suggest “that the *ZNF334* frameshift mutation occurs during a postzygotic event, and that both lymphoid and myeloid blood cells carry the heterozygous *ZNF334* mutation, potentially contributing to disease pathogenesis (p.9)”. With the evidence provided in the revised results, we proposed a model of pathogenesis of postzygotic monocyte *ZNF334* truncation mutation in the revised Discussion: “we proposed a model in which repeated cold exposure increases the levels of inflammatory cytokines and promotes the activation of *ZNF334*-mutated monocytes (Fig. 7). These monocytes release TNF- and chemokine-containing micro-vesicles that attract additional proinflammatory monocytes and lymphocytes to perivascular areas in cold-stimulated skin, as observed in our patient’s lesional skin biopsy. (p.24)”

2. Only one related control sample studied which is inadequate to establish level of intersubject variability

Our reply:

Thanks for the comment. We have added samples from other wild type healthy family members as described in the second section of the revised Results: “To investigate the effect of the *ZNF334* mutation on cold stress-associated inflammatory cytokine production, we analyzed the serum levels of TNF- α , IL-6, IL-1 β , and extracellular Hsp90 (eHsp90) in samples obtained from the patient’s

sister (who had wild-type ZNF334, with samples collected at four time points when the patient's sister accompanied her to the hospital), other wild-type family members (brothers and daughter, with samples collected at a single time point), and the patient herself (samples collected under different cold exposure conditions) (p.9-10)."

3. Inadequate clinical description, skin pathology

Our reply:

We have added detailed description of skin pathologies, and provided more information on the change of treatment and the treatment response in the first section of the revised Results: "Skin biopsy of the patient's urticaria-like rash, performed at the age of 47 years, revealed mild perivascular noncuffing lymphohistiocytic cell infiltration involving the superficial vascular plexus and small periadnexal capillaries (Fig. 1A).(p.6)" "...To determine the likelihood of urticarial vasculitis, another skin biopsy of the same type of urticaria-like rash was performed at the age of 51 years. It revealed neutrophilic and lymphocytic infiltrates in the walls of the dermal vessels (Fig. 1B), without the immune deposits of IgG, IgM, IgA, or C3. Masson's trichrome staining of the skin biopsy revealed no intravascular fibrin deposition (Fig. 1C). (p. 6)" "In June 2015, steroid and methotrexate (immunosuppressants) were discontinued for a short period as a trial. In addition, colchicine was added, and the dosage of celecoxib was increased, which resulted in a mild reduction in the patient's CRP level, but an abrupt increase in her ESR ("is-" period; Fig. 1E) (p.7)."

4. Inadequate explanation for why specific studies were performed

Our reply:

Please refer to the Results of the revised manuscript, thank you.

5. Complicated model that doesn't come together

Our reply:

We performed RNA-seq of ZNF334-edited THP-1 monocytes in the revised manuscript to gain further unbiased insight of the molecular mechanism of ZNF334 truncation pathogenesis, please refer to the revised Results section "RNA-seq of ZNF334-edited THP-1 monocytes revealed the involvement of ZNF334 in maintaining the regulatory protein folding machinery and suppressing

cold-induced inflammatory response (p.15)”. By adding the results of RNA-seq, reanalysis of CITE-seq, further Sanger sequencing of different cell types, and application of sorted cold-induced exophers to THP-1 monocytes, we have strengthened our model in the revised Fig.7, and summed up our model in the second paragraph of the revised Discussion (p.24).

6. Manuscript would benefit from editing by an English speaker

Our reply:

Thanks for your suggestion. We have sent our revised manuscript for English editing again.

Comments:

Title: I'm not sure the hearing loss should be in the title. While it is an important clinical feature it is not studied and only discussed briefly in discussion

Our reply:

We have deleted the hearing loss in the title accordingly. The revised Title is “ZNF334 truncation mutation drives cold induced autoinflammation”.

Abstract: Concise

Introduction

Appropriate review of current state of knowledge and summary of findings of this paper

Specific comments

Paragraph 1 sentence 3 doesn't make sense

Our reply:

Thank you. We have checked again the reference paper and modified the sentence in the revised Introduction: “It is reported that repeated cold water immersions increased the percentages of monocytes, activated lymphocytes, and plasma TNF- α levels in athletic young men”(p.3).

Paragraph 2 first sentence is inaccurate -FCAS is prototypical disease due to cold exposure

Our reply:

Thanks for the comment. We have corrected the sentence, please refer to the revised Introduction: “Cryopyrin-associated periodic syndromes (CAPS) encompass a spectrum of autoinflammatory disorders which result from dominant mutations in *NLRP3*, leading to hyperactivation of the *NLRP3* inflammasome in monocytes and macrophages... Familial cold autoinflammatory syndrome (FCAS) in the spectrum of CAPS is the prototypical disease due to cold exposure,...(P.3)”.

Results

Not enough detail of clinical presentation for a clinician to recognize this disease

Improper use of word anamnesis

It appears that this patient has 2 different types of rashes

Were the 2 biopsies taken from different types of lesions

Why was p65 and stat3 stained for. Were there other antibodies that were not observed

Our reply:

Thanks for the suggestions. The clinical details were added in the revised results as described in our reply to point 3 of primary weakness. We have deleted the word anamnesis. In our originally submitted manuscript, the pictures of the patient's arm before and after ice cube test might be misleading that the light of the camera was focused on the site where the ice cube bag was placed, making it brighter and therefore dampening the shape and color of the mildly raised urticarial rash at that area. Therefore, we have adjusted the saturation of all 3 photos to the same level in the revised Fig. 1F, which showed that the urticarial rash was already present before ice cube test, and the extent and redness of the same type of urticarial rash increased after ice cube test. The 2 biopsies were taken from different skin regions and at different time points, but with the same type of urticarial rash, as described in the revised Results, section 1: “another skin biopsy of the same type of urticaria-like rash was performed at the age of 51 years.(p.6)” We stained p-p65 and STAT3 because we have observed perivascular leukocyte infiltration in the HE staining, and would like to study if these leukocytes or other cell component in the

skin biopsy could be also stained positive for inflammatory markers. To explain this, we added the skin biopsy slides of HE staining with magnification of the infiltrated perivascular lymphocytes/neutrophils (revised Fig. 1A-B), and changed the confocal slide placing the dermal vessel in the middle of the photo (revised Fig. 1D). We did not stain for other fluorescent antibodies, but the skin biopsy had been stained for the immune deposits of IgG, IgM, IgA, or C3, which showed negative results (please refer to Results section 1) (p.6).

Figure legends are adequately detailed although n's are not always listed and not always clear if the data points are technical or experimental replicates

Our reply:

Please see the revised Figure legends: Fig 4O,P,Q; Fig.5A,B,C

Figure 1

A. Its not clear if there is any pathology here

B. Staining should be labeled and HE of similar region should be shown to see which cells are staining

C. Color coding is difficult to grasp here - clearly the patient has chronic inflammation but it is unusual that ESR went up when CRP went down in 2015

D. Poor quality pictures

E. Useful domain model

Our reply:

Please refer to the revised Fig.1: We added the skin biopsy slides of HE staining with magnification of the infiltrated perivascular lymphocytes/neutrophils (revised Fig. 1A-B), and Masson's trichrome staining of the skin biopsy showed absence of intravascular fibrin deposition (revised Fig. 1C). Furthermore, we selected also the perivascular view of confocal microscopy (revised Fig. 1D), which is similar to the area showing in (revised Fig. 1A-1B). The color coding in the figure of serial plasma inflammatory markers represents periods of cold exposures, there were two colors emphasizing the periods with or without the use of immunosuppressants, we simplified the color coding to one color in the revised Fig. 1E. We have improved the quality of the pictures.

Figure 2

A. Nomenclature of cold +++ and is + is difficult to follow

B. Authors need to briefly describe cite-seq in manuscript, not just in the

methods

C. Clear differences in UMAP, but authors should walk the reader through the significance

D. What is the significance of this

E. Statistical differences are difficult to see

Our reply:

Please refer to the revised Fig.2.: In the revised Fig.2A, we renamed these conditions as S1, S2, S3. We have added the brief description about CITE-seq in the revised Results, section of CITE-seq, second paragraph. Also, we provide stacked bar plots (revised Fig. 2D) and violin plots (revised Fig.2F-G) to further explain the significant differences of differential expression of clusters or cell subsets presented in the UMAP. We have modified the presentation of Fig.2D to a stacked bar plot which is easier to see the different proportions of each cluster in every sample. Differential top gene expressions in each cluster are presented in the heatmap of the revised Fig.2E. And according to these results, we presented in the revised Fig.2F-G, the significant differences in gene expression of monocytes (2F) and lymphocytes (2G). We have added the box plots in the revised violin plots to make the statistical significance more visible.

Figure 3

A. Clear

B. I agree with conclusions Loading controls do not appear to be loaded evenly

Figure 4

Significance of NLRP3 expression data is difficult to interpret as this does not demonstrate NLRP3 activation. Connection of this disease to NLRP3 is the weakest part of the paper.

Our reply:

We agree that *NLRP3* expression could be activated by NF- κ B, but does not demonstrate NLRP3 inflammasome activation. Therefore, we further looked at level of IL-1 β in the cell culture supernatant, which could hint the activation of inflammasomes. Please refer to Fig.4H and the Results “Furthermore, ZNF334^{+/-} and ZNF334^{-/-} THP-1 monocytes produced higher levels of IL-1 β detected in the culture supernatant than did ZNF334 wild-type cells both at 37°C (P = 0.0007) and after 24 h of 32°C cold stimulation (P = 0.0032; Fig. 4H).”(p.13)

Figure 5

Clear

Figure 6

A. domain model is clear

B. The significant of some of these graphs are problematic

C. molecular modeling Clear

D through H clear

Our reply:

Fig.6B: We used one-tailed Wilcoxon signed rank tests in the original figures. In the revised Fig.6B, we used two-tailed Wilcoxon signed rank tests, please see the modified statistical significance.

Figure 7

Hard to follow mechanism

Our reply:

Please refer to the revised Fig.7 and the revised Discussion, paragraph 2 (p.24).

Figure 8

Was this data used to decide how to treat the patient or did she just stay on steroids and methotrexate

Our reply:

She just stayed on steroids and methotrexate and avoided from cold exposures.

Discussion

A little long but conclusions and relevance are valid

Methods

Adequate detail of techniques

References - Mostly adequate

Paper describing Monocyte cold stimulation assay is not referenced

Our reply:

Thank you, we have added the reference in the revised Methods, section “Monocyte cold stimulation”. “Incubation of monocytes at 32°C has been used to evaluate the hyperinflammatory responses to mild cold exposure in patients of Familial cold autoinflammatory syndrome (FACS) (Rosengren et al, 2007).”(p.40)

Referee #3 (Comments on Novelty/Model System for Author):

To my perspective, there is not enough genetic data to claim the association between the frameshift in ZNF334 and the patients phenotype. Several experiments might benefit to be better controlled of potential artifact. However, if true, the identification of ZNF234 as a NLRP3 regulator would be highly novel.

Our reply:

Thanks for the comment. We have performed more analyses on the association between *ZNF334* truncation mutations in monocytes and cold-induced inflammatory cellular and clinical phenotypes, and added data of controls in the experiments. Please see our replies to the points below and the revised manuscript.

Referee #3 (Remarks for Author):

In this study, Lan et al. report a patient with acquired cold-induced inflammation harboring a heterozygous frameshift mutation in ZNF334. Cells from the patient exhibit increased cold-induced inflammation compared to a non-carrier sister. This heterozygous mutation is associated with elevated ZNF334 transcript levels but an almost complete absence of protein expression in the patient's monocytes following cold exposure. Knockout ZNF334 in THP-1 cells results in elevated levels of inflammatory transcripts and increased oxidative stress. Overexpression of the truncated ZNF334 in human monocytes leads to excessive production of TNF transcripts after cold exposure.

I found major shortcomings in this manuscript, and some observations seem largely overinterpreted or only superficially addressed:

- **The genetic results are not strong enough to claim that the observed frameshift mutation in ZNF334 is the cause of the disease.**

Our reply:

In the revised manuscript, we strengthened the impact of the *ZNF334* frameshift mutation by adding the conservation study of the mutation (revised Fig. 1G), and characterizing the mutation in monocytes and lymphocytes of the patient in the revised Results, section 1 (p.9). We further elaborated the CITE-seq analysis (p.9-11) and added RNA-seq analysis of *ZNF334*-edited THP-1 monocytes (p.15-16) to elucidate the molecular impact of *ZNF334* truncation mutation on monocytes, thus strengthening the pathogenic role of this mutation. Moreover, we added the section "Analyses of public genetic and RNA-seq datasets revealed an association between downregulated monocyte *ZNF334* expression and other rheumatic disease (p.22-23)" at the end of the revised Results, and described the limitations of genotype and phenotype association of this study in the revised Discussion (p.31-32).

- **Since the disease was acquired, deep sequencing (deep-seq) seems more appropriate than classical whole-exome sequencing (WES) to identify acquired somatic mutations, which are common in CAPS.**

Our reply:

Thanks for the suggestion. Due to the shortage of our research budget, we did not perform deep-seq in the revised manuscript. However, in the revised manuscript, we ran the mosaic pipeline in WES analysis which we did not identify pathogenic variants in *NLRP3*, please see the revised Discussion, second last paragraph (p.31): "Although deep sequencing is required to rule out known pathogenic somatic variants responsible for AIDs, our whole exome sequencing analysis with the available mosaic pipeline did not reveal any of these variants in our patient." We also sequenced monocytes and T cells and two FFPE blocks of skin biopsies for *ZNF334*, and identified that the *ZNF334* frameshift mutation in our patient is a post-zygotic somatic mutation. Please see the revised Results, section 1, paragraph 2 (p.9).

- **The authors do not discuss the presence of numerous frameshift mutations in *ZNF334* within healthy individuals. For example, p.Arg284Ter is found 983 times in GnomAD, p.Glu402LysfsTer4 is found 330 times, p.Lys347AlafsTer31 is found 134 times, p.Glu409Ter is found 115 times, and so on.**

Our reply:

Thank you for the comment. We have added the list of these frameshift mutations

in the revised Appendix Table S2, and the description in the last section of the revised Results: “We searched gnomAD v4.1, a large population-based genomics database that contains the genotypic data of more than 1.6 million alleles, for other ZNF334 truncation mutations nearby (p.22)”. Of note, the ZNF334 p.Thr399fs in our patient was identified by sequence alignment to GRCh37 (the reference gnomAD version was v.2.1.1), the same variant (location 20-45130776-CCT-C) has been lifted over to GRCh38 location 20-46502137-CCT-C, reported in gnomAD v4.1.0, with the same protein consequence, but designated as p.Glu402LysfsTer4 in the gnomAD database. Although the reported allele count was 330 in gnomAD v4.1.0, the total allele number in the database was 1614146, making an allele frequency of 0.0002044, which is similar to the rare frequency in the Taiwan Biobank population of 0.000670241. Both gnomAD and Taiwan Biobank did not exclude individuals with rheumatic disease whom might share similar inflammatory phenotypes as our patient. Furthermore, there are limitations in the interpretation of these truncation mutations as mentioned in our revised Results: “However, these variants were not reported in ClinVar, and data of Sanger sequencing validations, monocyte functional assays and environmental cold exposures were unavailable, making it difficult to interpretate their clinical significance (p.22-23)” In the revised Discussion, second last paragraph, we also mentioned about potential artifacts in sequencing ZNF334 p.Arg284Ter, since there were significant discrepant frequencies observed between whole exome and whole genome data of the ZNF334 p.Arg284Ter variant reported in gnomAD v4.1.0, and it is known that short-read sequencing is prone to mapping errors of ZNF variants due to the repetitive nature of ZNF genes. “Although other ZNF334 truncation mutations are reported in the gnomAD database, the genetic variants detected in ZNF proteins have been reported to be prone to inaccuracies in short-read sequencing because of potential alignment errors in sequences encoding repeated ZnF domains (Field *et al*, 2019). For example, significantly different frequencies are observed between the whole exome and whole genome data of ZNF334 p.Arg284Ter mutation in gnomAD v4.1, suggesting potential artifacts (Atkinson *et al*, 2023). Furthermore, the general population contributing to the total allele frequencies calculated in gnomAD does not exclude those with rheumatic diseases.”(p.31-32)

- **Although partial penetrance should not be excluded, the association between the patient's phenotype and the ZNF334 frameshift should be strengthened by both identifying other patients with ZNF334 mutations and a**

similar phenotype and excluding somatic mutations in genes driving autoinflammatory diseases using deep sequencing.

Our reply:

Thanks for the comment. Since the frequency of the ZNF334 frameshift variant is rare and the phenotype was induced by repeated cold exposures for years, it is very hard to find other patients with ZNF334 mutations and a similar phenotype. Therefore, we have addressed this limitation in the revised Discussion (p.31). Alternatively, we investigated whether the ZNF334 loss-of-function expression signature identified in our RNA-seq could be also observed in monocytes of other chronic inflammatory disease, such as juvenile idiopathic arthritis (JIA) in the last section of the revised Results (p.23). We have identified potential involvements of reduced *ZNF334* expression and decreased oxidative protein folding signaling in chronic monocyte inflammation in JIA as well. Please refer to Appendix Fig. S2.

• Other genes identified through WES analysis should be provided in a table.

Our reply:

Please refer to the revised Appendix Table S1.

Figure 1:

• Only one control was used for these experiments. Although it is useful to have a healthy relative as a control, given the inter-individual variability of immune responses, additional healthy donors are needed to draw robust conclusions.

Our reply:

Thanks for the comment. We have added additional wild-type healthy family members as controls in the revised Results.

Figure 3:

• It is unclear how the authors explain that a heterozygous frameshift mutation results in an almost complete absence of protein expression in the patient's monocytes.

Our reply:

It seems like the amount of ZNF334 protein isoforms and the degree of protein degradation in the patient's monocyte varied under different environmental

exposure conditions. As described in the revised Results, section 3 “Consistent with the results regarding mRNA levels, the protein expression level of ZNF334 was lowest in the patient’s monocytes under the “cold+ is+” condition (Fig. 3B) (p.11)” And as shown in the total protein blots, the “cold+ is+” lysate appeared to have more degradations than the other conditions. Both above factors might contribute to the faint to almost absent band seen in the lane of the “cold+ is+” lysate.

- **Why are no classical loading controls provided?**

Our reply:

Please refer to the revised Results, section 3: “In addition to Hsp90, the expression of several abundant proteins commonly used as Western blot loading controls, such as β -actin and GAPDH, have been reported to be altered in proteotoxic/oxidative stress conditions (Eaton et al, 2013; Nakajima et al, 2017). Therefore, in this study, we used stain-free total protein as a loading control (Fig. 3C). (p.12).”

- **The authors should provide information on the purity of the studied monocytes after the overnight adhesion process, as it might be affected by the disease.**

Our reply:

Please refer to the revised Methods, section “PBMC and monocyte isolation”: For RNA extraction and Western blotting, monocytes were further isolated from PBMCs by using the overnight adhesion method (monocyte enriched to 80~90%). (p.36)

Figure 4:

- **Were the same clones used per genotype in these experiments, or were multiple clones tested? A minimum of two clones per genotype is likely required to avoid potential artifacts.**

Our reply:

The same clones were used per genotype in these experiments. In the revised manuscript we further tested another clone of ZNF334^{+/-} THP-1 monocyte. Please refer to the revised Results, section “CRISPR/Cas9-mediated gene knockout revealed the role of ZNF334 in regulating cold-induced inflammation and cell

death in THP-1 monocytes“: “To evaluate potential off-target effects in CRISPR/Cas9 editing, we tested another ZNF334^{+/-} THP-1 clone, and observed similar effects on NF-κB and STAT3 signaling (Appendix Fig. S1). (p.13)”.

- **Is it expected for THP-1 cells to be almost 100% in G0/G1 at T0?**

Our reply:

As shown in Fig. 4I, ZNF334 wild-type THP-1 monocytes were almost 100% in G0/G1 at T0. However, 85~90% of the ZNF334-knockout THP-1 monocytes were in G0/G1 at T0, this could be due to increased ROS at baseline (shown in the revised Fig.5A), and increased inflammation (shown in Fig. 4A~G) at baseline.

- **For panels A to H: Present individual values instead of mean {plus minus} SEM. It should be clear whether the observed trend was consistent across replicates.**

Our reply:

Thanks for the suggestion. We have shown individual values in the revised Fig.4.

- **I cannot find the figure comparing IL-1β production in ZNF334 WT, +/- , and -/- at 37{degree sign}C.**

Our reply:

Please refer to Fig. 4H, 0 hour.

Figure 5:

5A and C: it is unclear what each dot/triangle represent. An individual well? Independent experiment's?

Our reply:

Fig. 5A and 5C data points were collected from individual experiments. Please refer to the revised Figure legend for Fig.5.

Figure 6:

• **It is unclear how the authors explain the discrepancy between ZNF IP and HSP IP in terms of HSP/ZNF interaction.**

Our reply:

Please refer to the 4th paragraph of the revised Discussion: “In our IP-western analyses, the binding of mutant ZNF334 to Hsp90 was almost absent in ZNF334 pull-down assays but present in Hsp90α/β antibody-immunoprecipitated samples. These findings may be attributable to the predicted weakening, rather than complete loss, in the interaction between mutant ZNF334 and Hsp90 and the potential masking of reduced binding affinity during ZNF334 blotting in immunoprecipitants of Hsp90, which is an abundant cellular protein. Upon cold exposure, the interaction between ZNF334 and Hsp90 was further weakened in the ZNF334^{+/-} monocytes (p.27).”

14th Jul 2025

Dear Prof. Yang,

Thank you for the submission of your revised manuscript to EMBO Molecular Medicine. We have now heard back from the two referees who agreed to re-evaluate your manuscript. As you will see from the reports below, referee #1 is supporting publication of the manuscript, while referee #3 remains critical regarding the lack of deep sequencing to exclude acquired mutation in genes involved in cold induced auto inflammation. After a consultation with my colleagues here, we concluded that raised concerns are justified and should be addressed in an additional and final round of major revision. We agreed that performing deep sequencing analysis as suggested by the referee #3 is essential for further consideration of the manuscript.

Please also amend following points:

1) Author Checklist: It seems that the Author Checklist is only partially completed. Please select the appropriate entries in column D

2) Figures: During our routine image checks, we noticed that the microscopy panels across the figure set appear pixelated. This is a common result of converting original 16-bit TIFF images to RGB format for publication, and while not a cause for concern, it can sometimes give the impression of image alteration to critical readers. To avoid any misunderstanding and to meet EMBO Press standards, we kindly ask that you:

- Resubmit the complete figure set at its original data resolution.

3) In the main manuscript file, please do the following:

- Please address all comments suggested by our data editors listed below:

- o Figure legends:

1. Please note that the box plots need to be defined in terms of minima, maxima, centre, bounds of box and whiskers, and percentile in the legends of figures 2F, G.

2. Please note that information related to n is missing in the legends of figures 2F, G; 8A-D; EV3.

- In Methods, please include statement that in addition to the principles set out in the WMA Declaration of Helsinki the experiments with patient samples also conformed the Department of Health and Human Services Belmont Report.

- Indicate in legends exact n and exact p values, not a range, along with the statistical test used. To keep the figures "clear" some authors found providing an Appendix table Sx with all exact p-values preferable. You are welcome to do this if you want to.

- Please remove Reagents and Tools Table and uploaded it as a separate file. Structured Methods section includes Reagents and Tools Table followed by a Methods and Protocols section. More information on how to adhere to this format as well as downloadable templates (.docx) for the Reagents and Tools Table can be found in our author guidelines:

<https://www.embopress.org/page/journal/17574684/authorguide#structuredmethods>

An example of a paper with Structured Methods can be found here:

<https://www.embopress.org/doi/full/10.1038/s44320-024-00037-6#sec-4>

- Author contributions: Please remove it from the manuscript. CRediT has replaced the traditional author contributions section because it offers a systematic machine-readable author contributions format that allows for more effective research assessment. You are encouraged to use the free text boxes beneath each contributing author's name to add specific details on the author's contribution. More information is available in our guide to authors:

<https://www.embopress.org/page/journal/17574684/authorguide#authorshipguidelines>

- Data availability: Please use the following format to report the accession number of your data:

[data type]: [full name of the resource] [accession number/identifier] ([doi or URL or identifiers.org/DATABASE:ACCESSION])

Please check "Author Guidelines" for more information.

<https://www.embopress.org/page/journal/17574684/authorguide#availabilityofpublishedmaterial>

4) Appendix: Please combine Appendix Figures S1-2 and Appendix Tables S1-4 in one file, uploaded in PDF format and labelled "Appendix". Please add a table of contents to the first page of the appendix, including page numbers.

5) Synopsis:

- Synopsis text: Please remove it from the manuscript and upload it as a separate .doc file.

- Synopsis figure: Please resize the image to 550 px-wide x 300-600 pixels high and upload it as a high-resolution jpeg file. Also, please simplify the image and adjust font size in the image, so that all components are visible and the text is readable in the new format.

- Please check your synopsis text and image, revise them if necessary and submit the final versions with your revised manuscript. Please be aware that in the proof stage minor corrections only are allowed (e.g., typos). Please submit synopsis text as a separate .doc file.

6) Source data: Please arrange the source data deposited in BioStudies as one file per figure.

Further consideration of a revision that addresses reviewer's concerns in full will entail an additional round of review.

Acceptance or rejection of the manuscript will depend on the completeness of your responses included in the next, final version

of the manuscript. For this reason, and to save you from any frustrations in the end, I would strongly advise against returning an incomplete revision.

We would welcome the submission of a revised version within three months for further consideration. Please let us know if you require longer to complete the revision.

I look forward to receiving your revised manuscript.

Yours sincerely,

Zeljko Durdevic

Zeljko Durdevic
Senior Editor
EMBO Molecular Medicine

We require:

- 1) A .docx formatted version of the manuscript text (including legends for main figures, EV figures and tables). Please make sure that the changes are highlighted to be clearly visible.
- 2) Individual production quality figure files as .eps, .tif, .jpg (one file per figure). For guidance, download the 'Figure Guide PDF': (<https://www.embopress.org/page/journal/17574684/authorguide#figureformat>).
- 3) A .docx formatted letter INCLUDING the reviewers' reports and your detailed point-by-point responses to their comments. As part of the EMBO Press transparent editorial process, the point-by-point response is part of the Review Process File (RPF), which will be published alongside your paper.
- 4) A complete author checklist, which you can download from our author guidelines (<https://www.embopress.org/page/journal/17574684/authorguide#submissionofrevisions>). Please insert information in the checklist that is also reflected in the manuscript. The completed author checklist will also be part of the RPF.
- 5) Please note that all corresponding authors are required to supply an ORCID ID for their name upon submission of a revised manuscript.
- 6) It is mandatory to include a 'Data Availability' section after the Materials and Methods. Before submitting your revision, primary datasets produced in this study need to be deposited in an appropriate public database, and the accession numbers and database listed under 'Data Availability'. Please remember to provide a reviewer password if the datasets are not yet public (see <https://www.embopress.org/page/journal/17574684/authorguide#dataavailability>).

- 8) At EMBO Press we ask authors to provide source data for the main manuscript figures. You will receive a separate email with

instructions for providing source data with your revised manuscript, including how to upload and organize the files.

12) Author contributions: You will be asked to provide CRediT (Contributor Role Taxonomy) terms in the submission system. These replace a narrative author contribution section in the manuscript.

13) A Conflict of Interest statement should be provided in the main text.

14) Every published paper now includes a 'Synopsis' to further enhance discoverability. Synopses are displayed on the journal webpage and are freely accessible to all readers. They include a short stand first (maximum of 300 characters, including space) as well as 2-5 one-sentences bullet points that summarizes the paper. Please write the bullet points to summarize the key NEW findings. They should be designed to be complementary to the abstract - i.e. not repeat the same text. We encourage inclusion of key acronyms and quantitative information (maximum of 30 words / bullet point). Please use the passive voice. Please attach these in a separate file or send them by email, we will incorporate them accordingly.

15) Include a Reagents and Tools Table as part of the Methods section, which can be downloaded from our author guidelines (<https://www.embopress.org/page/journal/17574684/authorguide#structuredmethods>)

***** Reviewer's comments *****

Referee #1 (Remarks for Author):

all my concerns have been addressed in this revised version

Referee #3 (Comments on Novelty/Model System for Author):

deep sequencing in patients blood cells is needed to exclude acquired mutation in genes involved in cold induced auto inflammation.

Referee #3 (Remarks for Author):

The authors have not responded to my 2 main criticisms: Given the frequency of ZNF334 LOF variants in public data bases, in the absence of deep sequencing to exclude acquired mutation in genes involved in cold induced auto inflammation (NLRP3, PLCG2) and in the absence of other cases, there is not enough evidence to claim that "ZNF334 truncation mutation in ZNF334 via whole exome sequencing monocytes as the novel genetic cause of this previously unreported case of a cold-induced AID with SNHL".

Manuscript number; EEM-2024-21141-V3

Point-by-point reply

<Comments from Referee #3>

1. Deep sequencing in patients blood cells is needed to exclude acquired mutation in genes involved in cold induced auto inflammation.

Our reply:

Thanks for the comment. We have deep sequenced the DNAs extracted from PBMCs of the patient and her *ZNF334*-wild type sister using amplicon-based deep sequencing covering the candidate autoinflammation-related genes *NLRP3* (exon 2, 3, 4, and 5), *NLRP12* (exon 1, 3, 4, 5, and 9), *NLRC4* (exon 3, and 4), *PLCG2* (exon 19, 20, 21, 22, 24, 27 and 30), *TNFRSF1A* (exon 2, 3, 4, 6, and 10), and *ZNF334* (exon 4, and 5) in the revised manuscript. Please refer to the revised Methods, section "Amplicon-based deep sequencing", and the revised Results, section "Deep sequencing revealed co-existence of a low-level *NLRP3* mutant in PBMCs derived from the patient".

Although we detected a 14% co-existing *NLRP3* p.Thr347Ile mutation in the patient's PBMCs, analyses of the patient's single-cell data and the plasma cytokine level during cold avoidance showed no evidence of spontaneous *NLRP3* inflammasome activation or excessive IL-1 β production in the monocytes, which is different from the reported functional impact of *NLRP3* p.Thr347Ile. Furthermore, there is a lack of cold-induced hyperinflammatory cellular and clinical phenotype in the literatures reporting the *NLRP3* p.Thr347Ile variant. Therefore, it is less likely that the acquired 14% *NLRP3* p.Thr347Ile mutation is involved in cold-induced autoinflammation in our patient. Please refer to our detailed reply to next point below.

2. The authors have not responded to my 2 main criticisms: Given the frequency of *ZNF334* LOF variants in public data bases, in the absence of deep sequencing to exclude acquired mutation in genes involved in cold induced auto inflammation (*NLRP3*, *PLCG2*) and in the absence of other cases, there is not enough evidence to claim that "ZNF334 truncation mutation in *ZNF334* via whole exome sequencing monocytes as the novel genetic cause of this previously unreported case of a cold-induced AID with SNHL".

Our reply:

Thanks for the comment. We have performed deep sequencing to detect potentially co-existing acquired mutations in genes involved in AID, including NLRP3 and PLCG2, as described in our reply to comment 1. We have added these findings in the revised Results (last section) and the revised Discussion (paragraph 8). Specifically, please refer to the Results, last section: “Similar to the results of whole exome sequencing, ZNF334 p.Thr399fs with variant allele frequency of 53.1% was detected in the patient, but not detected in the sister. Moreover, NLRP3 p.Thr347Ile with variant allele frequency of 14% (sequencing depth 16544) was also detected only in the patient”.

We also evaluated the potential impact of the low-level NLRP3 p.Thr347Ile on inflammatory cytokine production in monocytes by reanalyzing the CITE-seq single cell data gating on CD14+monocytes derived from PBMCs of the healthy sister (HC) and from PBMCs of the patient collected at two conditions: after a period of frequent cold exposure (S2) and after a period of cold avoidance (S3), “A total of 592 monocytes were analyzed in each sample: the number of *TNF*+ monocytes were 7, 25, 17 in HC, S2, and S3, respectively; the number of *NLRP3*+ monocytes were 2, 12, 5 in HC, S2, and S3, respectively; the number of *IL-1B*+ monocytes were 9, 8, 9 in HC, S2, and S3, respectively”. Please also refer to the last sentence of the revised Result: “Taken together, different levels of somatic mutations (53.1% ZNF334 p.Thr399fs and 14% NLRP3 p.Thr347Ile) were detected in patient’s blood cells. The numbers of *TNF*+ monocytes and *NLRP3*+ monocytes were highest in S2, while the proportions of *IL-1B*+ monocytes were similar among the three samples.” Since NLRP3 gain-of-function variant-carrying monocytes-induced CAPS phenotype requires number of *IL-1B*+ monocytes to increase, and previous functional assay reported spontaneous ASC speck formation and constitutive NLRP3 activation in NLRP3 p.Thr347Ile-expressing cells leading to excessive IL-1 β production (Feng *et al*, 2025, Nat Immunol), but the 14% NLRP3 p.Thr347Ile mutation in our patient’s blood cells did not show that, therefore it is unlikely the 14% NLRP3 p.Thr347Ile mutation causes the phenotype that was observed in the current study. Thus the most reasonable interpretation is the cold phenotype is caused by the 53.1% ZNF334 p.Thr399fs.

In the revised Discussion, paragraph 8, line 8, we discussed about the consensus that “the pathogenicity of low-level somatic mutations requires rigorous functional studies and extensive genetic investigations of co-existing germline or somatic variants that might be the actual driver mutation or exerting synergistic effects

(Schmitz *et al*, 2025, J Exp Med)”. According to our CITE-seq data (as mentioned above) and the plasma cytokine levels of our patient at the stage of cold avoidance, there is no evidence of baseline monocyte IL-1 β overproduction, which is different from the cellular phenotype of NLRP3 p.Thr347Ile reported using *in vitro* assays (Feng *et al*, 2025, Nat Immunol), please refer to the revised Discussion, paragraph 8, line 16-25, “.... These data suggest that unlike the reported phenotype of cells carrying NLRP3 p.Thr347Ile that showed constitutive NLRP3 inflammasome activation in *in vitro* assay (Feng *et al*, 2025), the majority of our patient’s monocytes did not secrete excessive IL-1 β .

Furthermore, given the strong association of the inflammatory phenotypes with cold exposures in our patient, we examined the details of the NLRP3 p.Thr347Ile listed in the INFEVERS registry website, including the reference of functional study and the reference for initial reported clinical phenotypes. We found that “...NLRP3 p.Thr347Ile was not listed in the category of disease-associated variants conferring cold exposure-triggered NLRP3 hyperactivation (Feng *et al*, 2025). The percentage of NLRP3 p.Thr347Ile variant detected in the patient was not reported in the reference publication listed in the INFEVERS registry (Suri *et al*, 2021). Moreover, unlike our patient, all the CAPS/MWS/NOMID cases described in this reference had very young age-of-onset, with severe consequences or accompanied with amyloidosis, and none of these patients had the record of cold exposure-induced symptoms (Suri *et al*, 2021).” Therefore, we made the assumption that “... it is less likely that the low-level NLRP3 somatic variant alone contributed to the systemic AID seen in this patient. Instead, our data derived from cold stimulation assays using CRISPR/Cas9-edited ZNF334-mutant THP-1 monocytes and *in vitro* transcribed mRNA overexpressing ZNF334 p.Thr399fs suggest that the heterozygous ZNF334 p.Thr399fs variant in monocytes is the main driver contributing to the adult-onset cold-induced AID in our case.” We also made the discussion that “Interestingly, while we observed that ZNF334 p.Thr399fs decreased baseline Hsp90 levels in monocytes, it has been reported that NLRP3 gain-of function variant-mediated NLRP3 autoactivations were hampered by Hsp90 deficiency (Spel *et al*, 2024)”, and “Whether the co-existing low-level NLRP3 p.Thr347Ile has minor impact on enhancing the autoinflammatory phenotype needs further investigations.” Please refer to the revised Discussion, paragraph 8, line 33-41.

As for the reviewer’s second criticism about “the absence of other cases”, we agree that it would strengthen our findings if we could identify other cases of exact ZNF334 mutation, with the same duration of cold exposures and the unique cellular

and clinical phenotypes. However, since we are reporting the first case of ZNF334 mutation-associated adult-onset cold-induced AID, the chance is very small to find other cases without publishing our first case. We have addressed this limitation in the Discussion in the previously revised version of our manuscript, and provided an analysis of RNA-seq derived from patients with juvenile idiopathic arthritis (JIA), given that abrupt reductions in temperature have been reported to trigger arthralgia. During last revision, we addressed that “We identified the potential involvement of downregulated *ZNF334* expression and decreased oxidative protein folding signaling in chronic monocyte inflammation in JIA”. Please refer to paragraph 9 of Discussion.

9th Oct 2025

Dear Prof. Yang,

Please find enclosed the final reports on your manuscript. We are pleased to inform you that your manuscript is accepted for publication and is now being sent to our publisher to be included in the next available issue of EMBO Molecular Medicine.

Zeljko Durdevic
Senior Editor
EMBO Molecular Medicine

Referee #1 (Comments on Novelty/Model System for Author):

the quality of the paper has been enhanced by all the controls made to exclude possibilities of somatic mutations in inflammatory genes; however, the clinical significance remains uncertain at this stage given the very low frequency of the syndrom which is described here.

As far as model system is concerned, the construction of a mutated mouse engineered to mimic the varuiant identified by the authors would take several months; but this would certainly close the debate regarding the causal relationship between the variant end the patients symptoms....

Referee #1 (Remarks for Author):

The addiction made in the manuscript clearly show that the authors have made a lot of efforts in order to exclude the potential involvement of somatic mutations. In the end, the causal relationship between a rare variant and the phenotype of a patient requires a finely adapted animal model (KI mouse) and/or additionnal human cases...
